# Is Pure Exploitation Sufficient in Exogenous MDPs with Linear Function Approximation?

**Hao Liang[1]**,* **Jiayu Cheng[2], Sean R. Sinclair[3], Yali Du[1,4]**

[1]Department of Informatics, King's College London    [2]Independent Researcher
[3]Department of Industrial Engineering and Management Sciences, Northwestern University
[4]The Alan Turing Institute
`haoliang.research@gmail.com,jiayucheng31@gmail.com,`
`sean.sinclair@northwestern.edu,yali.du@kcl.ac.uk`

## Abstract

Exogenous MDPs (Exo-MDPs) capture sequential decision-making where uncertainty comes solely from exogenous inputs that evolve independently of the learner's actions. This structure is especially common in operations research applications such as inventory control, energy storage, and resource allocation, where exogenous randomness (e.g., demand, arrivals, or prices) drives system behavior. Despite decades of empirical evidence that greedy, exploitation-only methods work remarkably well in these settings, theory has lagged behind: all existing regret guarantees for Exo-MDPs rely on explicit exploration or tabular assumptions. We show that exploration is unnecessary. We propose Pure Exploitation Learning (`PEL`) and prove the first general finite-sample regret bounds for exploitation-only algorithms in Exo-MDPs. In the tabular case, PEL achieves $\widetilde{O}(H^2|\Xi|\sqrt{K})$, where $H$ is the horizon, $\Xi$ the exogenous state space, and $K$ the number of episodes. For large, continuous endogenous state spaces, we introduce LSVI-PE, a simple linear-approximation method whose regret is polynomial in the feature dimension, exogenous state space, and horizon, independent of the endogenous state and action spaces. Our analysis introduces two new tools: counterfactual trajectories and Bellman-closed feature transport, which together allow greedy policies to have accurate value estimates without optimism. Experiments on synthetic and resource-management tasks show `PEL` consistently outperforming baselines. Overall, our results overturn the conventional wisdom that exploration is required, demonstrating that in Exo-MDPs, pure exploitation is enough.

## 1 Introduction

Sequential decision-making under uncertainty is central to a wide range of domains, from inventory control and energy storage to cloud resource management and supply chains (Madeka et al., 2022; Yu et al., 2021; Sinclair et al., 2023b; Oroojlooyjadid et al., 2022). In these applications the system dynamics are shaped by controllable endogenous states and exogenous inputs that evolve independently of the agent's actions. Exogenous Markov Decision Processes (Exo-MDPs) formalize this setting by partitioning states into endogenous and exogenous components, where actions only affect the former (Mao et al., 2018; Sinclair et al., 2023b). This separation models many practical settings where randomness is *external* (e.g. demands, arrivals, or prices) yet crucial for optimal control.

A striking empirical observation across these domains is that *pure exploitation works extremely well*. Classical approximate dynamic programming (ADP) and OR techniques repeatedly solve, act, and update from observed trajectories *without deliberate exploration*, and are deployed at industrial scale. Existing results show these schemes can converge in structured settings. For example, Nascimento & Powell (2009) prove convergence for lagged asset acquisition under concavity, demonstrating that pure exploitation can even outperform $\epsilon$-greedy exploration. More broadly, Powell (2022) highlights post-decision states and trajectory-based evaluation as foundational principles enabling effective

---

*Corresponding author.

exploitation-driven learning in practice. However, the theoretical guarantees in this line rely heavily on structure such as concavity or piecewise linearity.

In contrast, the reinforcement learning (RL) literature provides strong statistical guarantees for Exo-MDPs but almost always through *explicit exploration*. Sinclair et al. (2023b) develop hindsight- and replay-based methods that reuse exogenous traces, and Wan et al. (2024) establish a connection to linear-mixture models with regret bounds depending only on the exogenous cardinality. While these results underscore the power of exogenous structure, they either require exploration, assume tabular endogenous spaces, or rely on optimistic planning. This creates a fundamental mismatch with practice in operations research, where exploitation-heavy methods dominate. Hence a central question remains:

*Can pure exploitation strategies achieve near-optimal regret in Exo-MDPs under linear function approximation at scale?*

OUR CONTRIBUTIONS.

**Pure exploitation learning paradigm.** We introduce `PEL` (Pure Exploitation Learning), a unified exploitation-only framework for Exo-MDPs in which the learner repeatedly fits value approximations from observed trajectories and then acts greedily with respect to them. Prior ADP results are largely asymptotic or depend on problem-specific concavity, while existing RL guarantees for Exo-MDPs typically assume tabular structure, impose optimism, or reduce to linear mixtures, none of which address simple greedy methods under function approximation. A key structural observation is that in Exo-MDPs the exogenous process evolves independently of the agent's actions, so every trajectory provides unbiased information about *all* policies. This enables powerful data reuse. A single exogenous trace can be replayed to evaluate any policy's performance, eliminating the need for deliberate exploration. We resolve this gap by leveraging this philosophy and giving the first general finite-sample regret guarantees for `PEL` in Exo-MDPs with linear function approximation (LFA).

**Exo-bandits and tabular Exo-MDP.** To illustrate the core philosophy of `PEL`, we first analyze multi-armed bandits with exogenous information and tabular Exo-MDPs. In both settings we establish regret guarantees for pure exploitation, complementing and simplifying prior exploration-based approaches. These results form the basis for our extension to Exo-MDPs with LFA. Classical optimism-based analysis fails for `PEL`, and we propose a new regret decomposition and counterfactual analysis to derive a sublinear regret independent of endogenous space and action space size.

**Extension to LFA.** We then propose and analyze `LSVI-PE` (**L**east-**S**quares **V**alue **I**teration with **P**ure **E**xploitation), a backward value-iteration procedure that (i) builds empirical models of the exogenous process, (ii) constructs regression targets using post-decision states that disentangle action choice from exogenous randomness, and (iii) fits linear value approximations using data gathered entirely from greedy trajectories. Two technical ideas drive our analysis: (a) a counterfactual trajectory construction that enables reasoning about the value estimates produced under alternative endogenous traces, and (b) an anchor-closed Bellman-transport condition that controls how approximate Bellman updates propagate through the fitted linear representation. The resulting regret bounds are polynomial in the feature dimension, exogenous state cardinality, and horizon, and critically demonstrate that explicit exploration is unnecessary because exogenous data reuse suffices.

**Necessity of Exo-MDP structure for `PEL`.** The Exo-MDP assumptions are not only sufficient and realistic, but also necessary for `PEL` to work. If either the endogenous transition or the reward function is unknown, the problem no longer fits the Exo-MDP class, and exogenous traces cannot be reused for counterfactual evaluation. In such settings, any `PEL` algorithm necessarily incurs linear regret, showing that pure exploitation succeeds only under Exo-MDP structure.

**Paper organization.** Section 2 reviews related work and Section 3 formalizes the Exo-MDP model. Section 4 analyzes pure exploitation in the tabular setting, and Section 5 introduces `LSVI-PE` with its regret analysis under linear function approximation. Section 6 reports empirical results. Section 7 concludes the paper. Proofs are deferred to the appendix for space considerations.

## 2 RELATED WORK

We briefly review the most salient related works here and refer to Appendix B for more details.

**Exo-MDPs.** Exogenous MDPs, a sub-class of structured MDPs, were introduced by Powell (2022) and further studied in an evolving line of work (Dieterich et al., 2018; Efroni et al., 2022; Sinclair et al., 2023b; Powell, 2022). For instance, Dieterich et al. (2018); Efroni et al. (2022) considered factorizations that filter out the exogenous process, simplifying algorithms but yielding suboptimal policies since ignoring exogenous states may discard useful information. Sinclair et al. (2023b) analyzed hindsight optimization, showing that its regret can be bounded by the hindsight bias, a problem-dependent quantity. Most closely related to our work, Wan et al. (2024) establish statistical connections between Exo-MDPs and linear mixture models and design exploration-based algorithms with regret guarantees in fully discrete Exo-MDPs, assuming finite endogenous and exogenous state spaces and primarily i.i.d. exogenous inputs, with a Markovian extension. Overall, existing results largely focus on discrete endogenous dynamics and i.i.d. (or simplified) exogenous processes and typically rely on explicit exploration or optimism. In contrast, we study Exo-MDPs with *continuous endogenous states* and *Markovian exogenous processes*, and we provide the first near-optimal finite-sample regret guarantees for *pure exploitation* strategies in this more general setting.

**Exploitation-based ADP.** A parallel line of work in ADP shows that greedy or exploitation-oriented strategies can succeed under strong structural assumptions. Nascimento & Powell (2009) propose a pure-exploitation ADP method for the lagged asset acquisition model, leveraging concavity of the value function to guarantee convergence without explicit exploration. Nascimento & Powell (2013) extend this to vector-valued controls in storage problems under similar conditions. More broadly, Jiang & Powell (2015) and Powell (2022) highlight methods such as Monotone-ADP and post-decision state exploitation schemes that reduce the need for exploration by exploiting monotonicity or other structural regularities. However, these methods either assume discrete state and action spaces, rely on asymptotic convergence, or require structural conditions like convexity, monotonicity, or piecewise-linearity. In contrast, we provide finite-sample regret guarantees for pure exploitation in *general* Exo-MDPs without any explicit structural assumptions.

**MDPs with LFA.** Recent work on RL with LFA has studied various linear structures, including MDPs with low Bellman rank (Jiang et al., 2017; Dann et al., 2018), linear MDPs (Yang & Wang, 2019; Jin et al., 2020), low inherent Bellman error (Zanette et al., 2020), and linear mixture MDPs (Jia et al., 2020; Ayoub et al., 2020; Zhou et al., 2021). Our results contribute to this literature by establishing near-optimal regret guarantees for Exo-MDPs with LFA under pure exploitation.

## 3 PRELIMINARIES AND PROBLEM SETTING

**Notation.** We write $[N] := \{1, 2, \cdots, N\}$ for any positive integers $N$. For a matrix $A$, we use $\|A\|$ to denote its operator norm. We use $\mathbb{I}\{\cdot\}$ to denote the indicator function. For any $x \in \mathbb{R}$, we define $[x]^+ := \max\{x, 0\}$. We use $\tilde{\mathcal{O}}(\cdot)$ to denote $\mathcal{O}(\cdot)$ omitting logarithmic factors. A table of notation is provided in Appendix A.

**MDPs with exogenous states.** We consider Exogenous Markovian Decision Processes (Exo-MDPs) with Markovian exogenous dynamics, a subclass of MDPs that explicitly separates the state into *endogenous* and *exogenous* components (Dieterich et al., 2018; Efroni et al., 2022; Sinclair et al., 2023b; Powell, 2022). Here, a state $s = (x, \xi)$ factorizes into an endogenous (system) state $x \in \mathcal{X}$ and exogenous input $\xi \in \Xi$. Intuitively, the exogenous state $\xi_h$ captures all randomness (e.g., demand, arrivals, or prices), while the endogenous state $x_h$ captures the system's internal configuration. Because actions cannot influence $\xi_h$, the agent cannot manipulate future randomness, which is central to our pure-exploitation results. Formally, an Exo-MDP is defined by the tuple $\mathcal{M}(\mathbb{P}, f, r) = (\mathcal{X} \times \Xi, \mathcal{A}, \mathbb{P}, r, H)$. At each stage $h$, the agent selects an action $a_h = \pi_h(s_h) \in \mathcal{A}$ given the current state $s_h = (x_h, \xi_h)$ under their policy $\pi = (\pi_h)_{h \in [H]} \in \Pi$ where $\Pi = \{(\pi_h)_{h \in [H]} : \pi_h : \mathcal{X} \times \Xi \to \mathcal{A}\}$. The exogenous state evolves as a Markov process $\xi_{h+1} \sim \mathbb{P}_h(\cdot|\xi_h)$, independent of $x_h$ and $a_h$.[1] Throughout we assume the exogenous state space is discrete, which is well-aligned in operations research where the exogenous randomness corresponds to discrete demand levels in inventory control (Besbes & Muharremoglu, 2013; Cheung et al., 2023) or job types in cloud computing systems (Balseiro et al., 2020; Sinclair et al., 2023b).

---

[1]We discuss the general $m$-Markovian setting in Appendix C.

Conditional on $(x_h, a_h, \xi_h)$, the endogenous process evolves and the reward function is specified by deterministic functions:

$$x_{h+1} = f(x_h, a_h, \xi_{h+1}), \quad r_h = r(x_h, a_h, \xi_h) \in [0, 1].$$

The endogenous dynamics are still stochastic, only deterministic as a function of the exogenous state distribution through $f$. The full transition kernel from a state can be written as $P(s_{h+1}|s_h, a_h) = \mathbb{I}\{f(x_h, a_h, \xi_{h+1}) = x_{h+1}\}\mathbb{P}_h(\xi_{h+1} \mid \xi_h)$.

*Remark* 1. These modeling assumptions are well-motivated in many operation research applications, especially resource management. For example, in inventory control, the endogenous state $x_h$ is the on-hand inventory level, while the exogenous state $\xi_h$ is the demand realization at time $h$ (Madeka et al., 2022). Actions $a_h$ correspond to order quantities. The system transition function are deterministic given demand, e.g. the newsvendor dynamics $x_{h+1} = f(x_h, a_h, \xi_{h+1}) = \max\{x_h + a_h - \xi_{h+1}, 0\}$. The reward depends on sales revenue and holding or stockout costs, $r(x_h, a_h, \xi_h)$. The only randomness arises from the exogenous demand process. We give more examples of Exo-MDPs in Appendix C.1.3.

**Value functions and Bellman equations.** For a policy $\pi$, the action-value functions and state-value functions at step $h$ are defined as:

$$Q_h^\pi(s, a) := \mathbb{E}\left[\sum_{\tau=h}^H r(x_\tau, a_\tau, \xi_\tau) \mid (s_h, a_h) = (s, a), \pi\right], \quad V_h^\pi(s) := Q_h^\pi(s, \pi_h(s)).$$

We also define *hindsight value functions* for a fixed exogenous trace $\boldsymbol{\xi}_{>h} = (\xi_{h+1}, \ldots, \xi_H)$:

$$Q_h^\pi(s, a, \boldsymbol{\xi}_{>h}) := \sum_{\tau=h}^H r(s_\tau, a_\tau, \xi_\tau) \mid (s_h, a_h) = (s, a), \pi, \quad V_h^\pi(s, \boldsymbol{\xi}_{>h}) := Q_h^\pi(s, \pi_h(s), \boldsymbol{\xi}_{>h}).$$

These are deterministic once $\boldsymbol{\xi}_{>h}$ are fixed, so no Monte Carlo sampling is required under the known functions $f$ and $g$. Sinclair et al. (2023b) show that unconditional values are expectations over hindsight values, i.e. for every $h \in [H], (s, a) \in \mathcal{S} \times \mathcal{A}$, and policy $\pi$,

$$Q_h^\pi(s, a) = \mathbb{E}_{\boldsymbol{\xi}_{>h}}[Q_h^\pi(s, a, \boldsymbol{\xi}_{>h})], \quad V_h^\pi(s) = \mathbb{E}_{\boldsymbol{\xi}_{>h}}[V_h^\pi(s, \boldsymbol{\xi}_{>h})],$$

where the expectation is taken over the conditional distribution $\boldsymbol{\xi}_{>h} \sim \mathbb{P}_h(\cdot \mid \xi_h)$.

**Online learning.** We consider an agent interacting with the Exo-MDP over $K$ episodes. At the beginning of episode $k$, the agent starts from an initial state $s_1^k$ and commits to a policy $\hat{\pi}^k \in \Pi$. At each step $h$, the agent observes $s_h^k = (x_h^k, \xi_h^k)$, takes action $a_h^k = \hat{\pi}_h^k(s_h^k)$, receives reward $r(x_h^k, a_h^k, \xi_h^k)$, observes $\xi_{h+1}^k$, and transitions to $x_{h+1}^k = f(x_h^k, a_h^k, \xi_{h+1}^k)$. Each episode has $H$ steps. The performance of an algorithm is measured by its cumulative simple regret over $K$ episodes:

$$\mathrm{SR}(\texttt{alg}, k) := V_1^{\pi^\star}(s_1) - V_1^{\hat{\pi}^k}, \quad \mathrm{CR}(\texttt{alg}, K) := \sum_{k=1}^K \left[V_1^{\pi^\star}(s_1) - V_1^{\hat{\pi}^k}(s_1)\right],$$

where $\pi^\star = \arg\max_{\pi \in \Pi} V_1^\pi(s)$ is the optimal policy and $\hat{\pi}^k$ is the policy employed in episode $k$. The initial exogenous state $\xi_1^k$ in each episode can be arbitrarily chosen. For each $(k, h) \in [K] \times [H]$, we denote by $\mathcal{H}_h^k \triangleq \left(s_1^1, a_1^1, s_2^1, a_2^1, \ldots, s_H^1, a_H^1, \ldots, s_h^k, a_h^k\right)$ the (random) history up to step $h$ of episode $k$. We define $\mathcal{F}_k \triangleq \mathcal{H}_H^{k-1}$ as the history up to episode $k - 1$. We use $\boldsymbol{\xi}^k := (\boldsymbol{\xi}^l)_{l \in [k]}$ to denote the exogenous trace up to the end of episode $k$.

## 4 PURE EXPLOITATION LEARNING IN TABULAR EXO-MDPS

We now illustrate the philosophy of *Pure Exploitation Learning* (PEL). In Exo-MDPs, the only unknown component is the *exogenous* process, which evolves according to a Markov chain independent of the agent's actions. As a result, trajectories collected under *any policy* provide unbiased information about this process, so explicit exploration is *not required*. PEL builds on this observation: instead of adding optimism or randomization, PEL algorithms repeatedly *fit* empirical models or value functions from observed exogenous traces and then acts *greedily* with respect to these estimates. To summarize we define PEL algorithms as:

**Definition 1** (Informal). PEL denotes the family of algorithms that, at each round or episode, construct an empirical value function from previously observed exogenous traces and act by greedily maximizing this function, with no optimism or forced exploration.

We next make PEL concrete in two simple settings: (i) an Exo-bandit warm-up ($H = 1$) and (ii) the tabular Exo-MDP. After presenting regret guarantees and computational remarks, we conclude with an impossibility example showing that PEL can fail in general MDPs without exogenous structure. We then move onto the linear function approximation case.

## 4.1 Warm-up: Exo-bandits

We start with multi-armed bandits with exogenous information (coinciding with bandits with full feedback), an Exo-MDP with no states and $H = 1$. At each round $k$ the agent selects arm $a_k$, an exogenous input $\xi_k$ is realized, and because the reward map $r(a, \xi)$ is known the agent can evaluate the reward $r(a, \xi_k)$ of *all* arms. Following Wan et al. (2024) we call this setting an Exo-Bandit.

Here, the `PEL` strategy reduces to the classic Follow-The-Leader (`FTL`) strategy: at round $k$ simply choose the arm with the largest empirical mean reward $a_k \in \arg\max_{a \in \mathcal{A}} \hat{\mu}_a(k) := \frac{1}{k-1} \sum_{s=1}^{k-1} r(a, \xi_s)$. This procedure is entirely exploration free, unlike in classical bandits where exploration schemes such as UCB or Thompson Sampling are essential for learning (Auer et al., 2002; Russo et al., 2018). This contrast illustrates how exogenous information fundamentally changes the role of exploration since the algorithm can use the *counterfactual* information inferred from the observed exogenous information.

**Proposition 1.** Assume rewards are $\sigma^2$-sub-Gaussian. Then the expected per-round simple regret of `FTL` satisfies $\mathrm{SR}(\texttt{FTL}, k) \leq \sqrt{\frac{2\sigma^2 \log A}{k-1}}$, and consequently the cumulative regret obeys $\mathrm{CR}(\texttt{FTL}, K) \leq 2\sigma \sqrt{(K-1)\log A}$.

The proof is provided in Appendix F.1. These regret bounds recover standard full-information or experts-type guarantees and are are minimax-optimal (Cesa-Bianchi & Lugosi, 2006; Shalev-Shwartz et al., 2012). The point here is not novelty but an illustration: when full feedback is available via the exogenous feedback, simple `PEL` suffices, and additional exploration is no longer necessary.

## 4.2 Tabular Exo-MDPs

We now extend `PEL` to finite-horizon Exo-MDPs with finite state and action spaces. Since exogenous traces can be reused across policies, one can form unbiased value estimates and apply Follow-the-Leader (`FTL`) at the policy level. This yields near-optimal regret bounds, consistent with Sinclair et al. (2023b), but evaluating all policies amounts to empirical risk minimization (ERM) over $\Pi$. This is computationally infeasible in general since $|\Pi| \leq |\mathcal{A}|^{H|\mathcal{X}||\Xi|}$. See Appendix D for a discussion of this algorithm and the result.

To address this, we consider a *more practical* `PEL` instance, Predict-Then-Optimize (`PTO`). `PTO` first estimates the exogenous transition kernels $\widehat{\mathbb{P}}_h^k(\cdot \mid \xi_h)$ (e.g., via empirical counts or MLE), and then plugs them into standard dynamic programming to compute greedy policies:

$$\widehat{Q}_h^k(s_h, a_h) := r(x_h, a_h, \xi_h) + \mathbb{E}_{\xi_{h+1}|\xi_h}\left[\widehat{V}_{h+1}^k(f(x_h, a_h, \xi_{h+1}), \xi_{h+1}); \widehat{\mathbb{P}}^k\right],$$

$$\widehat{\pi}_h^k(s_h) \in \arg\max_{a_h} \widehat{Q}_h^k(s_h, a_h), \quad \widehat{V}_h^k(s_h) := \widehat{Q}_h^k(s_h, \widehat{\pi}_h^k(s_h)).$$

The following theorem bounds the cumulative regret of `PTO` under Markovian exogenous noise by reducing model error to *exogenous-row* errors, yielding rates independent of $|\mathcal{X}|$ and $|\mathcal{A}|$.

**Theorem 1** (Regret of `PTO` under Markovian exogenous process). With high probability, the cumulative regret of `PTO` after $K$ episodes satisfies

$$\mathrm{CR}(\texttt{PTO}, K) \leq \widetilde{\mathcal{O}}\big(H^2|\Xi|\sqrt{K}\big).$$

The proof is provided in Appendix F.3. A key challenge is that classical optimism-based analysis fails for pure exploitation in Exo-MDPs. Even though the only unknown is the exogenous kernel $P_\Xi$, the optimistic inequality $V_1^{\pi^\star} - \widehat{V}_1^{k,\pi^k} \leq 0$ does not hold, so the usual simulation-lemma telescoping breaks. We instead introduce a *new regret decomposition* with two *double value gaps*, separating model errors for $\pi^\star$ and $\pi^k$. While the on-policy term can be bounded as in classical analyses, the term for $\pi^\star$ cannot—its trajectory is never observed. This motivates our use of *counterfactual trajectories* that follow $\pi^\star$ but share the same exogenous realization $\boldsymbol{\xi}^k$. Conditioning on the exogenous filtration allows a simulation-lemma bound without requiring visitations under $\pi^\star$. This rewrites the value gaps purely in terms of exogenous-kernel errors, replacing state-action counts by policy-independent exogenous counts $C_h^k(\xi_h)$, resolving the policy-misalignment issue and yielding sublinear regret independent of the endogenous state and action spaces.

Unlike the exhaustive ERM/FTL approach, which is statistically sound but computationally infeasible, `PTO` provides a practical and efficient `PEL` implementation. It runs in time polynomial in $|\mathcal{X}|, |\mathcal{A}|$, and $H$, while preserving regret guarantees that depend only mildly on the exogenous cardinality $|\Xi|$.

### 4.3 IMPOSSIBILITY: PURE EXPLOITATION CAN FAIL IN GENERAL MDPS

To understand why the Exo-MDP structure is essential for `PEL`, we examine what happens when its key assumptions are violated. Specifically, we consider `PEL` when either the endogenous transition function $f$ or the reward function $r$ is unknown.

**Proposition 2.** For any pure-exploitation algorithm, there exists a tabular MDP with unknown transition function $f$ or unknown reward function $r$ on which the algorithm suffers linear regret.

The proof is provided in Section C.1.2. This result shows that once $f$ or $r$ is unknown, the class of otherwise tabular Exo-MDPs already contains instances where every `PEL` algorithm incurs linear regret. In such settings, pure exploitation is not minimax-sufficient. Hence, the Exo-MDP structure is not just sufficient, it is the critical condition that makes it possible for `PEL` to achieve sublinear regret, in sharp contrast to the general tabular MDP setting with unknown $f$ or $r$.

**Discussion.** While this section considered simple tabular Exo-MDPs, we showed that pure exploitation suffices: exploration is unnecessary because exogenous randomness is decoupled from the agent's actions, and Exo-MDP structure is necessary for `PEL` to work. With the right implementation (e.g., `PTO`), `PEL` is both statistically and computationally efficient. However, these results hinge on tabular representations, limiting scalability. In the next section, we extend these ideas to continuous state and action spaces under linear function approximation.

## 5 LINEAR FUNCTION APPROXIMATION

The previous section established that `PEL` suffices in tabular Exo-MDPs. However, in order to make this useful for more realistic and high-dimensional problems, we need to move beyond finite endogenous state spaces. This section develops `LSVI-PE`, a simple and efficient pure-exploitation algorithm under linear function approximation. Our algorithm leverages two structural ideas: (i) post-decision states, which remove the confounding between actions and exogenous noise; and (ii) counterfactual trajectories, the same principle that underpinned our tabular analysis.

**Continuous Exo-MDPs.** We now consider Exo-MDPs with continuous endogenous states $x_h \in \mathcal{X}$, continuous actions $a_h \in \mathcal{A}$, and finite exogenous states $\xi_h \in \Xi$ over horizon $H$. This extension is essential for modeling realistic operations research and control applications, where system states (e.g., inventory levels, resource capacities, storage levels) and actions are naturally continuous. Following the ADP literature (Nascimento & Powell, 2009; 2013; Powell, 2022), we assume that the dynamics decompose into two steps:

$$x_h^a = f^a(x_h, a_h) \in \mathcal{X}^a \subset \mathcal{X} \text{ (post--decision state)}, \quad x_{h+1} = g(x_h^a, \xi_{h+1}) \in \mathcal{X} \text{ (next state)},$$

with $\xi_{h+1} \sim \mathbb{P}_h(\cdot \mid \xi_h)$. For any policy $\pi$, we define the *post--decision value function*

$$V_h^{\pi,a}(x^a, \xi) = \mathbb{E}_{\xi' \sim \mathbb{P}_h(\cdot|\xi)}\Big[ V_{h+1}^\pi(g(x^a, \xi'), \xi') \Big],$$

which represents the expected downstream value after committing to action $a_h$ but before the next exogenous state is revealed. The pre-decision value function then decomposes as

$$V_h^\pi(x, \xi) = r\big(x, \pi(x, \xi), \xi\big) + V_h^{\pi,a}\big(f^a(x, \pi(x, \xi)), \xi\big).$$

The optimal policy also obeys

$$V_h^\star(x, \xi) = \max_{a \in \mathcal{A}} \Big\{ r(x, a, \xi) + V_h^{\star,a}\big(f^a(x, a), \xi\big) \Big\}, V_h^{\star,a}(x^a, \xi) = \mathbb{E}_{\xi' \sim \mathbb{P}_h(\cdot|\xi)}\Big[ V_{h+1}^\star\big(g(x^a, \xi'), \xi'\big) \Big].$$

We now formalize the definition of Exo-MDP with linear function approximation (LFA):

**Definition 2.** An Exo-MDP is said to satisfy **post--decision LFA** with respect to a known feature mapping $\phi : \mathcal{X} \to \mathbb{R}^d$ if, for every policy $\pi$, step $h$, and state $(x^a, \xi) \in \mathcal{X} \times \Xi$,

$$V_h^{\pi,a}(x^a, \xi) = \phi(x^a)^\top w_h^\pi(\xi),$$

where $\sup_{x^a} \|\phi(x^a)\|_2 \leq 1$, and the weight vectors satisfy $\sup_{\pi,h,\xi} \|w_h^\pi(\xi)\|_2 \leq \sqrt{d}$.

Thus, post-decision LFA can be viewed as an Exo-MDP analogue of the linear MDP assumption, tailored to exploit the separation between endogenous dynamics and exogenous randomness. We denote the optimal weights by $w_h^\star(\xi) := w_h^{\pi^\star}(\xi)$ so that $V_h^{\star,a}(x^a, \xi) = \phi(x^a)^\top w_h^\star(\xi)$.

Since the endogenous state space $\mathcal{X}$ may be continuous, we cannot directly regress on all post-decision states. To make the LFA identifiable and the least-squares updates well defined, we introduce a finite collection of representative post-decision states whose feature vectors span the feature space.

**Assumption 1** (Anchor set). For each step $h$, there exist $N \geq d$ fixed post–decision states $\{x_h^a(n)\}_{n=1}^N$ such that the feature matrix $\Phi_h := \left[\phi(x_h^a(1)), \ldots, \phi(x_h^a(N))\right] \in \mathbb{R}^{d \times N}$ has full row rank, i.e., $\mathrm{rank}(\Phi_h) = d$.

Together, the LFA assumption and anchor condition provide a tractable representation that supports efficient algorithms while keeping regret bounds polynomial in the feature dimension $d$ rather than the size of the underlying endogenous state or action spaces. We also emphasize that Assumption 1 is standard in the ADP literature (Nascimento & Powell, 2009; 2013).

## 5.1 ALGORITHM

In this section, we present our algorithm **L**east-**S**quares **V**alue **I**teration with **P**ure **E**xploitation (LSVI-PE) for Exo-MDPs with LFA. See Algorithm 1 for pseudo-code.

**High-level intuition.** Our algorithm LSVI-PE alternates between two phases:

1. **Policy evaluation (backward pass):** At each stage $h$, we construct Bellman regression targets using the empirical exogenous model $\hat{\mathbb{P}}_h$ (Line 10). Then we run least-squares regression on the anchor states to produce weight vectors $w_h^k(\xi)$ for each exogenous state $\xi$ and stage $h$, defining a linear approximation for the value function as $V_h^{k,a}(x^a, \xi) = \phi(x^a)^\top w_h^k(\xi) \approx V_h^\star(x^a, \xi)$.

2. **Policy execution (forward pass):** In episode $k$, the agent acts greedily with respect to these value estimates (Line 19). The observed exogenous trajectory is used to refine the empirical estimate $\hat{\mathbb{P}}$.

Before moving onto the regret analysis we briefly comment on several aspects of the algorithm.

**Role of anchor states.** Anchor states $\{x_h^a(n)\}_{n=1}^N$ are chosen to guarantee that the feature matrix $\Phi_h$ has full row rank (Assumption 1). This ensures that the regression weights $w_h^k(\xi)$ are unique. Intuitively, anchors serve as representative endogenous states: they provide just enough coverage of the feature space to propagate accurate value estimates without requiring samples from the entire (possibly continuous) state space.

Anchor states are not unknown structural assumptions but design choices made by the learner. Assumption 1 simply requires fixing a finite set of post–decision states whose feature vectors span the space. These serve as the representative grid underlying the feature map (e.g., hat bases, spline knots, or tile centers (Sutton et al., 1998)). This mirrors standard practice in RL with LFA, where choosing $\phi$ implicitly specifies the underlying basis points. Such anchor constructions are routine in ADP and operations-research applications, including inventory control and storage systems, where practitioners exploit domain structure (e.g., piecewise linearity or convexity) to select natural breakpoints as anchors (Nascimento & Powell, 2009; 2013; Powell, 2022).

**Exploration-free design.** Conventional RL algorithms with LFA rely on explicit exploration mechanisms. For instance, LSVI-UCB (Jin et al., 2020) enforces optimism in the value estimates, while RLSVI (Osband et al., 2016) injects random perturbations into regression targets. In contrast, LSVI-PE is a *pure exploitation* algorithm. All updates come directly from empirical exogenous trajectories observed along greedy play. The independence of the exogenous process makes this design both natural and theoretically justified, and we later show it achieves near-optimal regret.

**Computational efficiency.** In LSVI-PE, regression targets are computed only at the anchor states, and updates decompose stage by stage. This structure makes the algorithm scalable when the endogenous state and action spaces are continuous. Compared to FTL-style policy search, which requires evaluating every policy, LSVI-PE is implementable in polynomial time.

---

**Algorithm 1** `LSVI-PE`

---

**Require:** Anchor states $\{x_h^a(n)\}_{h=1,n=1}^{H,N}$; feature map $\phi : \mathcal{X} \to \mathbb{R}^d$

1: **Precompute:** For each $h$, set $\Phi_h \leftarrow [\phi(x_h^a(1)), \ldots, \phi(x_h^a(N))] \in \mathbb{R}^{d \times N}$ and $\Sigma_h \leftarrow \Phi_h \Phi_h^\top$

2: **Initialize:** For each $h$ and $\xi, \xi' \in \Xi$, set counts $C_h(\xi, \xi') \leftarrow 0$ and $\hat{\mathbb{P}}_h^0(\xi'|\xi) \leftarrow 1/|\Xi|$; set $w_h^0(\xi) \leftarrow \mathbf{0}$ for all $h \in [H+1], \xi$

3: **for** $k = 1$ to $K$ **do**                                                    // Episode loop

4:    // Policy computation using data up to $k-1$ //

5:    **for** $h = H$ down to 1 **do**

6:       **for** each $\xi \in \Xi$ **do**

7:          $b_h^k(\xi) \leftarrow \mathbf{0} \in \mathbb{R}^d$

8:          **for** $n = 1$ to $N$ **do**

9:             Define $x_n'(\xi') \leftarrow g(x_h^a(n), \xi')$ for each $\xi' \in \Xi$

10:             $y_h^k(n; \xi) \leftarrow \sum_{\xi' \in \Xi} \hat{\mathbb{P}}_h^{k-1}(\xi'|\xi) \max_{a' \in \mathcal{A}} \left\{ r(x_n'(\xi'), a', \xi') + \phi(f^a(x_n'(\xi'), a'))^\top w_{h+1}^k(\xi') \right\}$

11:             $b_h^k(\xi) \leftarrow b_h^k(\xi) + \phi(x_h^a(n)) \, y_h^k(n; \xi)$

12:          **end for**

13:          $w_h^k(\xi) \leftarrow \Sigma_h^{-1} b_h^k(\xi)$                         // Least squares on anchors

14:       **end for**

15:    **end for**

16:    // Act in episode $k$ with $\{w_h^k\}$ and collect data $\boldsymbol{\xi}^k$ //

17:    Receive $x_1^k$; observe $\xi_1^k$

18:    **for** $h = 1$ to $H$ **do**

19:       $a_h^k \in \arg\max_{a \in \mathcal{A}} \left\{ r(x_h^k, a, \xi_h^k) + \phi(f^a(x_h^k, a))^\top w_h^k(\xi_h^k) \right\}$

20:       $x_h^{k,a} \leftarrow f^a(x_h^k, a_h^k)$; observe $\xi_{h+1}^k$; set $x_{h+1}^k \leftarrow g(x_h^{k,a}, \xi_{h+1}^k)$

21:       Update counts: $N_h^k(\xi_h, \xi_{h+1}) \leftarrow N_h^{k-1}(\xi_h, \xi_{h+1}) + \mathbb{I}\{(\xi_h, \xi_{h+1}) = (\xi_h^k, \xi_{h+1}^k)\}$;

22:    **end for**

23:    **Update empirical model:** For all $h, \xi, \xi'$, $\hat{\mathbb{P}}_h^k(\xi'|\xi) \leftarrow \frac{N_h^k(\xi, \xi')}{\sum_{\zeta \in \Xi} N_h^k(\xi, \zeta)}$.

24: **end for**

25: **Output:** $w_h^k(\xi)$ for each $h$ and $\xi$

---

## 5.2 REGRET ANALYSIS

Before presenting our main result we introduce some additional notation. Let $\phi_h(n) := \phi(x_h^a(n))$ and define the anchor feature matrix $\Phi_h := [\phi_h(1), \ldots, \phi_h(N)] \in \mathbb{R}^{d \times N}$. We also define $\lambda_0 := \min_{h \in [H]} \lambda_{\min}(\Sigma_h) > 0$, where $\Sigma_h = \Phi_h \Phi_h^\top$ is the anchor covariance. Fix $h$, $\pi$, and $\xi' \in \Xi$. We define the *post-decision transition operator* as $\mathcal{T}_h^\pi(\xi') : \mathcal{X}^a \to \mathcal{X}^a$ as

$$\mathcal{T}_h^\pi(\xi')(x^a) := f^a\Big(g(x^a, \xi'), \, \pi\big(g(x^a, \xi'), \xi'\big)\Big).$$

This represents one step of evolution:

$$x^a \xrightarrow{\xi'} x' \xrightarrow{\pi} a' \xrightarrow{f^a} (x^a)' \quad \text{as the compressed arrow} \quad x^a \xRightarrow[\pi]{\xi'} (x^a)' = \mathcal{T}_h^\pi(\xi')(x^a).$$

We introduce two additional assumptions to establish our regret guarantees. We begin with a weaker requirement, that the anchor states are *closed under the Bellman operator*. Intuitively, this condition ensures that when an anchor state undergoes one step of post-decision transition, its image remains in the span of the anchor feature representation.

**Assumption 2** (Anchor-closed Bellman transport (weaker))**.** For any $\pi$, $h \in [H]$, and $\xi' \in \Xi$, there exists a matrix $M_h^\pi(\xi') \in \mathbb{R}^{d \times d}$ with $\sup_{\pi, \xi', h} \|M_h^\pi(\xi')\|_2 \leq 1$ such that for every anchor $x_h^a(n)$,

$$\phi\big(\mathcal{T}_h^\pi(\xi')(x_h^a(n))\big) = M_h^\pi(\xi') \, \phi(x_h^a(n)).$$

Note that this establishes the one-step image of any anchor under the post-decision transition lies in the same feature span and is linearly transported by $M_h^\pi(\xi')$.

**Assumption 3.** For any $x^a$, $\phi(x^a)$ is in the nonnegative cone of $\Phi$.

Assumption 3 ensures the pointwise policy-improvement, the greedy update makes all anchor residuals nonnegative, and thus guarantees improvement at arbitrary post-decision states.

**Theorem 2.** Under Assumption 2-3, the regret of `LSVI-PE` after $K$ episodes satisfies

$$\mathrm{CR}\,(\texttt{LSVI-PE}, K) \;\leq\; \tilde{\mathcal{O}}\Big(\big(\sqrt{N/\lambda_0} + \sqrt{d}\,\big)\,|\Xi|\,H\,\sqrt{K}\,\Big).$$

`PEL` achieves standard sublinear regret under Assumption 2, with dependence on the feature dimension $d$, the number of anchors and their conditioning via $\sqrt{N/\lambda_0}$, and the exogenous state size $|\Xi|$, while remaining independent of the size of the endogenous state and action spaces. In well-conditioned designs (e.g., $\lambda_0 = \Theta(1)$ and $N \approx d$), the bound simplifies to $\tilde{\mathcal{O}}(|\Xi|H\sqrt{dK})$.

The intuition behind the proof parallels the tabular setting. We adopt a new decomposition that requires controlling two value gaps. The key step is to bound the optimal-policy gap using a simulation lemma applied not to the realized trajectory of $\pi^k$, but to a *counterfactual trajectory* generated under $\pi^\star$ while sharing the same realized exogenous sequence. Assumption 2 ensures that all Bellman regression targets remain in the anchor span, so each stage-$h$ update reduces to a well-conditioned least-squares problem governed by $\lambda_0$. Along the counterfactual process, the proof decomposes the error into (i) Bellman regression errors at the anchors and (ii) exogenous-model errors, and then couples both components to the observed exogenous trajectory through martingale concentration on the estimated exogenous rows. This contrasts sharply with standard linear-MDP optimism analyses, which rely on confidence sets and self-normalized concentration in parameter space; here the central analytic objects are the counterfactual trajectories and the exogenous martingales that enable a stage-wise telescoping of Bellman errors and yield the $\tilde{\mathcal{O}}(\sqrt{K})$ regret bound without optimism. Full proofs are in Appendix G.

Our next assumption strengthens Assumption 2 to hold for all $x^a$ instead of just the anchors:

**Assumption 4** (Global Bellman-closed transport (stronger)). For any $\pi$, $h \in [H]$, and $\xi' \in \Xi$, there exists $M_h^\pi(\xi')$ with $\sup_{\pi,\xi',h} \|M_h^\pi(\xi')\|_2 \leq 1$ such that for all $x^a$, $\phi\big(\mathcal{T}_h^\pi(\xi')(x^a)\big) = M_h^\pi(\xi')\,\phi(x^a)$.

Under this we can establish the following regret guarantee:

**Theorem 3.** Under Assumption 4, the regret of `LSVI-PE` after $K$ episodes satisfies

$$\mathrm{CR}\,(\texttt{LSVI-PE}, K) \;\leq\; \tilde{\mathcal{O}}\Big(\big(H + \sqrt{N/\lambda_0}\,\big)\,|\Xi|\,H\,\sqrt{K}\,\Big).$$

While both theorems share the same dependence on $K$, this refinement tightens the guarantees when $H < d$. Although Assumption 4 is stricter than what `LSVI-PE` requires, we show it yields sharper propagation bounds when exact closure is plausible (or enforced by feature design).

We provide a detailed discussion of the anchor-set assumptions in Appendix C.2, including the role and selection of anchor states, connections to coreset and Frank-Wolfe methods, the invertibility and conditioning of $\Sigma_h$, the reasonableness of Assumptions 2-4, and several weakenings and relaxations. Below we briefly highlight the intuition behind Assumptions 2- 4.

**Discussion on Assumptions 2 to 4.** Many Exo-MDPs such as storage problems or linearizable post-decision dynamics naturally induce linear transport within common LFA classes (linear splines, tile coding, localized RBFs, etc). Moreover, the constraint $\|M_h^\pi(\xi')\|_2 \leq 1$ ensures that one-step feature transport is non-expansive, a standard stability condition in ADP/LFA analyses. Additional discussion of Assumptions 2 to 4 is provided in Appendix E.

**`LSVI-PE` with misspecification (approximation) error.** When Assumptions 2 to 4 fails and the true value functions do not lie exactly in the linear span, or the function class is misspecified, Theorem 4 shows that the regret bounds match the earlier ones with an additive $O(K\varepsilon_{\mathrm{BE}})$ where $\varepsilon_{\mathrm{BE}}$ measures the measures the inherent Bellman error[2] (approximation gap between the true Bellman updates and the best function in the linear class). This bias term is unavoidable in general, since even an oracle learner suffers an $O(K\varepsilon_{\mathrm{BE}})$ cumulative bias (Zanette et al., 2020).

**Theorem 4.** Assume Assumption 1 holds. Fix $\delta \in (0,1)$. Then with probability at least $1 - \delta$,

$$\mathrm{CR}\,(\texttt{LSVI-PE}, K) \;\leq\; \tilde{\mathcal{O}}\left(\Big(H + \sqrt{\tfrac{N}{\lambda_0}}\Big)|\Xi|H\sqrt{K} \;+\; \frac{H}{\sqrt{\lambda_0}}\,K\,\varepsilon_{\mathrm{BE}}\right).$$

---

[2]Formal definition is provided in Appendix E.1

## 6 NUMERICAL EXPERIMENTS

### 6.1 TABULAR EXO-MDP

**Setup.** We evaluate on synthetic tabular Exo-MDPs with endogenous state space $\mathcal{X} = [5]$, exogenous state space $\Xi = [5]$, and action set $\mathcal{A} = [3]$ and horizon $T = 5$, and $K = 250$ episodes. Rewards are drawn i.i.d. as $r(x, a, \xi) \sim$ Unif$(0, 1)$. Endogenous dynamics are deterministic, $x_{h+1} = f(x_h, a_h, \xi_{h+1}) = (x_h + a_h + \xi_{h+1}) \bmod X$, while the exogenous process is a Markov chain with transition matrix $P_y$ sampled row-wise from a Dirichlet prior.

**Comparisons.** We compare PTO with its optimistic counterpart PTO-Opt (using optimistic model $\tilde{\mathbb{P}}^k$) and PTO-Lite (lightweight estimate $\tilde{\mathbb{P}}^k$ using sub-sampling).

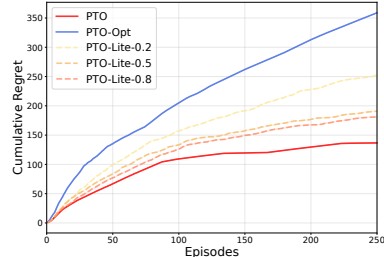

Figure 1: Comparison of PTO, PTO-Opt and PTO-Lite.

Figure 1 illustrates the benefit of PEL. Despite no explicit exploration, PTO outperforms PTO-Lite and the exploration-heavy baseline PTO-Opt in cumulative regret.

### 6.2 STORAGE CONTROL

**Setup.** We consider a storage control setting where $x_h \in \mathcal{X} = [0, C]$ denotes the current storage level. After taking action $a_h \in \mathcal{A} = [-a_{\max}, a_{\max}]$, the system transitions to the post-decision state $x_h^a = f^a(x_h, a) = \text{clip}_{[0,C]}\left(x_h + \eta^+ a^+ - \frac{1}{\eta^-} a^-\right)$. The exogenous component is the discrete price. The storage level is modeled as $x_{h+1} = g(x_h^a, \xi_{h+1}) = \alpha\, x_h^a, \alpha \in (0, 1]$, with default $\alpha = 1$. The reward function is $r(x_h, a_h, \xi_h) = -\xi_h a_h - \alpha_c |a_h| - \beta_h x_h$, capturing the market transaction, transaction cost, and holding penalty respectively.

**Features and anchors.** We discretize $\mathcal{X}$ using anchors $\rho_n = \frac{n-1}{N-1} C$ for $n \in [N]$. A one-dimensional hat basis is employed: for any $x^a$, the feature vector $\phi(x^a) \in \mathbb{R}^N$ has at most two nonzero entries. Let $\Delta = \rho_{j+1} - \rho_j$. If $x^a \in [\rho_j, \rho_{j+1}]$, then $\phi_j(x^a) = \frac{\rho_{j+1}-x^a}{\Delta}, \phi_{j+1}(x^a) = \frac{x^a-\rho_j}{\Delta}$, with all other coordinates zero. At anchor points, the basis reduces to canonical vectors, $\phi(\rho_n) = e_n$, so that $\Phi_h = I_N$ and $\Sigma_h = \Phi_h \Phi_h^\top = I_N$.

**Comparisons.** In Figure 2 we compare LSVI-PE with optimism-based exploration LSVI-Opt. Across all instances, LSVI-PE consistently outperforms LSVI-Opt, emphasizing that in Exo-MDPs exploitation strategies dominate optimism-based ones.

We provide scaled-up experimental setup and comprehensive comparisons in Appendix H.

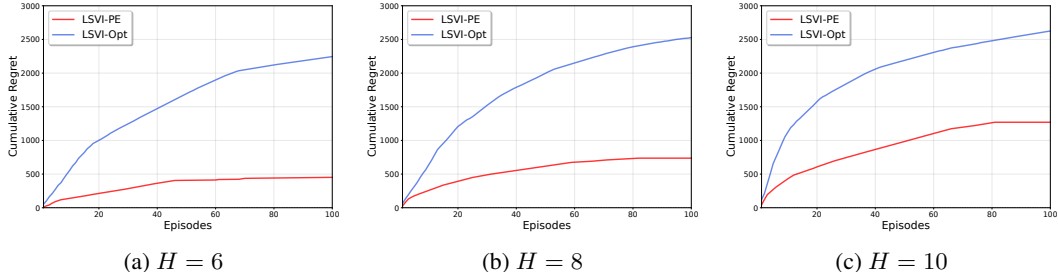

(a) $H = 6$      (b) $H = 8$      (c) $H = 10$

Figure 2: Comparison of LSVI-PE and LSVI-Opt across three different time horizon lengths.

## 7 CONCLUSION

We show that exploitation is sufficient in Exo-MDP. By introducing PEL, we give the first finite-sample regret bounds for PEL under tabular and LFA, and demonstrate PEL outperforms optimism-based baselines on synthetic and resource-management benchmarks. Future work include relax structural assumptions (richer function classes, continuous or partially observed exogenous processes) while preserving exploitation's sample efficiency.

ACKNOWLEDGMENTS

We would like to thank all the anonymous reviewers for their careful proofreading and constructive feedback, which have greatly improved the quality of this work. This work was supported by the Engineering and Physical Sciences Research Council [grant number EP/Y003187/1 and UKRI849].

ETHICS STATEMENT

This research is foundational and develops theoretical results on reinforcement learning in Exo-MDPs with linear function approximation. As such, it does not raise any direct ethical concerns. However, applications of our algorithms to specific domains (e.g., inventory control, pricing, or resource allocation) may influence real-world decision-making that affects people and organizations. We therefore encourage practitioners to carefully consider ethical implications such as fairness, accessibility, and potential unintended consequences when deploying these methods in practice.

REPRODUCIBILITY STATEMENT

All proofs of theorems and lemmas are included in the appendix, and we clearly specify all assumptions used in our analysis. Algorithmic details (see Algorithms 1 and 2) are provided to ensure transparency. Our empirical results are based on synthetic Exo-MDP benchmarks and resource-management tasks, both of which we describe in Section 6 and Appendix H.

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

## A  TABLE OF NOTATION

Table 1: List of common notations.

| Symbol | Definition |
|---|---|
| *Exo-MDP specification* | |
| $\mathcal{X}$ | Endogenous (system) state space |
| $\Xi$ | Exogenous state space |
| $\mathcal{A}$ | Action space |
| $H$ | Planning horizon |
| $K$ | Number of episodes |
| $x_t \in \mathcal{X}$ | Endogenous state at time $t$ |
| $\xi_t \in \Xi$ | Exogenous input at time $t$ |
| $a_t \in \mathcal{A}$ | Action at time $t$ |
| $f : \mathcal{X} \times \mathcal{A} \times \Xi \to \mathcal{X}$ | Endogenous transition function, $x_{t+1} = f(x_t, a_t, \xi_t)$ |
| $\mathbb{P}(\xi' \mid \xi)$ | Exogenous transition kernel |
| $r : \mathcal{X} \times \mathcal{A} \times \Xi \to [0, 1]$ | Reward function |
| $\pi : \mathcal{X} \times \Xi \to \mathcal{A}$ | Policy mapping state to action |
| $V_h^\pi(x, \xi)$ | Value function of policy $\pi$ at stage $h$ |
| $Q_h^\pi(x, a, \xi)$ | State-action value function of policy $\pi$ at stage $h$ |
| $\text{Regret}(K)$ | Cumulative regret after $K$ episodes |
| *Pure Exploitation Framework* | |
| `PEL` | Pure Exploitation Learning framework |
| `FTL` | Pure Exploitation algorithm for Exo-bandits ($H = 1$) and tabular Exo-MDPs |
| `LSVI-PE` | Pure Exploitation algorithm for Exo-MDPs with linear function approximation |
| *LFA* | |
| $\phi(x)$ | Feature map of state $x$ |
| $d$ | Feature dimension |
| $\theta_h$ | Parameter vector at stage $h$ |
| $\hat{P}_h$ | Empirical estimate of exogenous transition at stage $h$ |
| $\hat{Q}_h, \hat{V}_h$ | Estimated $Q$- and value functions |
| $\iota$ | Logarithmic factor $\log(2KH|\Xi|/\delta)$ in regret bounds |
| *Storage Control Example* | |
| $C$ | Storage capacity |
| $x_h \in [0, C]$ | Storage level at stage $h$ |
| $\xi_h \in \Xi$ | Price at stage $h$ |
| $a_h = (a_h^+, a_h^-)$ | Charge ($a^+$) / discharge ($a^-$) actions |
| $\eta^+, \eta^-$ | Charging/discharging efficiencies |
| $x_h^a$ | Post-decision state after action $a_h$ |
| $\hat{\mathbb{P}}(\xi'|\xi)$ | Estimated price transition kernel |
| *Theoretical Analysis* | |
| $\delta$ | Confidence parameter in high-probability bounds |
| $N$ | number of anchor points |
| $\mathcal{O}(\cdot), \tilde{\mathcal{O}}(\cdot)$ | Standard big-$O$ and log-suppressed complexity notation |

## B  DETAILED RELATED WORK

**Exo-MDPs.** Exogenous MDPs, a structured sub-class of MDPs, have been introduced and studied in a growing line of work (Powell, 2022; Dieterich et al., 2018; Efroni et al., 2022; Sinclair et al., 2023b; Feng et al., 2021; Alvo et al., 2023; Chen et al., 2024). Early approaches (e.g., Dieterich et al. (2018); Efroni et al. (2022)) exploit factorizations that filter out the exogenous process, simplifying learning but potentially yielding suboptimal policies since policies agnostic to the exogenous states need not be optimal. Other work leverages hindsight optimization, bounding regret by the hindsight bias, a problem-dependent quantity (Sinclair et al., 2023b; Feng et al., 2021). Across this literature, the dominant assumptions are that endogenous states and actions are discrete and that guarantees

rely on optimism or tabular analysis. More recently, Wan et al. (2024) connect Exo-MDPs to linear mixture MDPs, proving regret bounds that are independent from the size of the endogenous state and action spaces, but their results apply only to discrete endogenous states. In contrast, we study Exo-MDPs with *continuous endogenous states* and *Markovian exogenous processes*, and establish the first near-optimal regret guarantees for *pure exploitation* under linear function approximation.

**Exploitation-based ADP.** A parallel line of research in ADP has shown that greedy or exploitation only strategies can succeed under strong structural assumptions. Nascimento & Powell (2009) analyze a pure-exploitation ADP method for the lagged asset acquisition model, where the concavity of the value function guarantees convergence without explicit exploration. Nascimento & Powell (2013) extend this approach to storage problems with vector-valued controls under similar conditions. More broadly, Jiang & Powell (2015) and Powell (2022) survey methods such as Monotone-ADP and post-decision exploitation schemes which reduce the need for exploration by leveraging monotonicity or other structural regularities. Related work has also sought to mitigate exploration using Bayesian beliefs (Ryzhov et al., 2019) or by exploiting factored state representations (Guestrin et al., 2003; Kveton et al., 2006). However, these methods generally assume discrete state and action spaces, or depend on strong structural conditions (e.g. concavity or monotonicity). In contrast, we provide finite-sample regret guarantees for pure exploitation in general Exo-MDPs with continuous endogenous states and Markovian exogenous components.

**Regret analysis of pure exploitation (exploration-free) methods.** Recent work has begun characterizing when greedy policies can still achieve sublinear regret. Bastani et al. (2021) show that in contextual bandits, a fully greedy algorithm attains $O(\sqrt{T})$ regret under a covariate diversity assumption. Civitavecchia & Papini push this into RL, proving that greedy LSVI (no bonus) can yield sublinear regret under sufficient feature diversity. Jedor et al. (2021) analyze greedy strategies in multi-armed bandits and delineate regimes where pure exploitation suffices. Bayati et al. (2020) demonstrate that in many-armed regimes, greedy policies exploit a "free exploration" effect emerging from the tail structure of the prior to achieve sublinear regret. Kim & Oh (2024) gives a broader class of context distributions under which greedy linear contextual bandits enjoy poly-logarithmic regret Kim & Oh (2024). Efroni et al. (2019) show that in finite MDPs, one can match minimax regret bounds by using greedy planning on estimated models (i.e. no explicit exploration). Most recently, Hssaine et al. (2025) show that in certain structured resource management settings, greedy value-iteration–type methods can enjoy convergence guarantees without explicit exploration. These results suggest that under strong structural or distributional conditions, pure exploitation may rival exploration-based methods, albeit in narrower settings than general theory guarantees.

**MDPs with function approximation.** RL with structural assumptions has been studied under both nonparametric and parametric models. Nonparametric approaches, such as imposing Lipschitz continuity or smoothness conditions on the $Q$-function, offer flexibility but suffer from exponential dependence on state/action dimension (Shah & Xie, 2018; Sinclair et al., 2023a). Parametric approaches trade model flexibility for computational tractability, typically assuming that the MDP can be well-approximated by a linear representation. This has fueled a rich literature on RL with linear function approximation, spanning settings such as low Bellman rank (Jiang et al., 2017; Dann et al., 2018), linear MDPs (Yang & Wang, 2019; Jin et al., 2020; Hu et al., 2024), low inherent Bellman error (Zanette et al., 2020), and linear mixture MDPs (Jia et al., 2020; Ayoub et al., 2020; Zhou et al., 2021). Indeed, Exo-MDPs are closely related to linear mixture MDPs. Wan et al. (2024) establish a structural equivalence between the two, but only in the case of *discrete* endogenous and exogenous spaces. Our contribution focuses on adapting the machinery of linear function approximation to Exo-MDPs for *continuous* endogenous spaces, and show that their properties allow for pure exploitation strategies to achieve near-optimal regret.

**Exo-MDPs in practice.** A growing empirical literature has applied function approximation (typically using neural networks) to Exo-MDPs in operations research applications, particular in inventory control and resource management problems (Madeka et al., 2022; Alvo et al., 2023; Fan et al., 2024; Qin et al., 2023). These works demonstrate strong practical performance but provide limited theoretical guarantees. In contrast, our contribution simplifies the function class to *linear* function approximation, which allows us to obtain sharp regret bounds while retaining the structural advantage of Exo-MDPs. Moreover, while some prior work focused on heuristic policy classes such as base stock policies (Agrawal & Jia, 2022; Zhang et al., 2025), our algorithms converge to the *true optimal policy*, thereby avoiding the suboptimality inherent to such restricted classes. Lastly we note that RL

has been applied to various other problems in operations research (without exploiting their Exo-MDP structure) including ride-sharing systems (Feng et al., 2021), stochastic queuing networks (Dai & Gluzman, 2021), and jitter buffers (Fang et al., 2019). Applications of our method can potentially improve sample efficiency in these applications by exploiting the underlying exogenous structure.

## C DISCUSSION ON EXO-MDP MODELING ASSUMPTIONS

### C.1 KNOWN TRANSITION FUNCTIONS AND REWARD FUNCTIONS.

Our model assumes that the endogenous dynamics $f$ and the reward function $r$ are known and deterministic given the exogenous state. While this assumption is more restrictive than the fully general unknown MDP model typically studied in the RL literature, it is well-motivated in many operations research domains. Indeed, inventory control, pricing, scheduling, and resource allocation problems are often modeled with deterministic system dynamics where the only uncertainty arises from exogenous randomness (Powell, 2022). This assumption also aligns with the practice of simulator-based design, widely adopted in queueing and inventory control studies (e.g., Madeka et al. (2022); Alvo et al. (2023); Che et al. (2024)).

We highlight several aspects of the assumption that the transition and reward function $f$ and $r$ are known below.

#### C.1.1 SUFFICIENCY

These assumptions are precisely what make pure exploitation viable. Once $f$ and $r$ are known the only source of uncertainty is the exogenous distribution $\mathbb{P}_h(\cdot \mid \xi)$ which is independent of the learner's actions. This structure enables data reuse and counterfactual value estimation and is the basis for our regret guarantees. No analysis of PE in even fully tabular Exo-MDP (with Markovian exogeneous proceses). We establish the regret bound of PE in tabular Exo-MDP, and more importantly, we are the first to show the effectiveness of PE in the case of continuous endogenous state space and continuous action space.

#### C.1.2 NECESSITY

First, we show that in Proposition 2 and Theorem 5 that there exist 2-armed Bernoulli bandits such that `PEL` (FTL, greedy w.r.t. empirical means) suffers linear regret.

**Impossibility with unknown reward functions.** Because a (1-step) bandit is a special case of a tabular Exo-MDP with unknown reward function, this directly yields:

**Corollary 1.** Consider the class of tabular Exo-MDPs with horizon $H = 1$, a single state $x$, and a finite action set $\mathcal{A}$. The reward of each action $a \in \mathcal{A}$ is an unknown random variable with mean $\mu(a) := \mathbb{E}[r(a, \xi)]$. Any pure-exploitation algorithm suffers $\Omega(K)$ expected regret on some MDP in this class.

*Proof.* A 1-step, 1-state tabular Exo-MDP with unknown reward function is exactly a stochastic MAB, with each action corresponding to an arm and the (unknown) random reward $R_a = r(a, \xi)$. Theorem 5 then yields an instance on which any pure-exploitation algorithm has $\Omega(K)$ regret. $\square$

Over the class of tabular Exo-MDPs with unknown reward function $r$, pure exploitation is not minimax-optimal. No pure-exploitation algorithm can guarantee $\mathcal{O}(K)$ regret in the worst case.

**Impossibility with unknown transition functions.** Unknown transition functions and known terminal reward are equivalent (from the learner's perspective) to unknown rewards of the two actions at $h = 1$. A pure-exploitation algorithm that plans greedily from an estimated model behaves just like a pure-greedy bandit algorithm on the two "effective arms" corresponding to "go to state 1 or 2". Formaly, we let $\mathcal{A}$ be any pure-exploitation algorithm that uses its estimates of the transition function (e.g., empirical transition frequencies) to choose, in each episode, an action at $x^{(0)}$ that maximizes its current estimated value function, never choosing actions whose estimated value is strictly smaller than another action's estimated value.

**Corollary 2.** Consider tabular Exo-MDPs with horizon $H = 2$, state space $\mathcal{X} = \{x^{(0)}, x^{(1)}, x^{(2)}\}$, and action set $\mathcal{A} = \{1, 2\}$. The initial state is always $x^{(0)}$. At the first step $h = 1$, the transition from $x^{(0)}$ under action $a \in \{1, 2\}$ is random:

$$x^{(i(a))} = f(x^{(0)}, a, \xi),$$

where each action $a$ leads to stochastic state $x^{(i(a))} \in \{x^{(1)}, x^{(2)}\}$. At the second step $h = 2$, the episode terminates and a (known) deterministic reward is obtained, and the initial return for each action $a$ is given by

$$R_a = \mathbb{P}(i(a) = 1)r(x^{(1)}) + \mathbb{P}(i(a) = 2)r(x^{(2)}).$$

In particular, we choose $f$ and $r(x^{(1)}), r(x^{(2)})$ such that

$$R_1 = \tfrac{1}{2} + \Delta, \qquad R_2 = \tfrac{1}{2},$$

for some fixed $\Delta \in (0, 1/4]$, independently across episodes. Then there exists a choice of $f$ and $r(x^{(1)}), r(x^{(2)})$ such that the expected cumulative regret of $\mathcal{A}$ over $K$ episodes is $\Omega(K)$.

*Proof.* From the learner's perspective, each action $a \in \{1, 2\}$ induces an unknown expected return equal to the (known) reward of the state it deterministically reaches at $h = 2$. Thus the problem is equivalent to a two-armed bandit with unknown means $\frac{1}{2} + \Delta$ and $\frac{1}{2}$. A pure-exploitation algorithm that always selects an action with maximal estimated value behaves exactly like a pure-greedy bandit algorithm on these two arms. By the same argument as in Theorem 5, there exists an stochastic assignment of actions to terminal states such that the algorithm suffers $\Omega(K)$ expected regret. $\square$

**Necessity of known endogenous transition functions and rewards.** Corollaries 1 and 2 show that, as soon as either the reward function $r$ or the transition function $f$ is unknown, the class of tabular Exo-MDPs already contains instances where any pure-exploitation algorithm incurs *linear* regret. In particular, pure exploitation is not minimax-sufficient in these settings.

By contrast, in our Exo-MDP framework we assume that the endogenous dynamics $f$ and reward function $r$ are known, and only the exogenous kernel is unknown. This structural assumption is crucial. It allows us to design pure-exploitation algorithms that achieve sublinear regret, in sharp contrast to the tabular Exo-MDP setting with unknown $f$ or $r$.

### C.1.3   REASONABLENESS

**Example applications of Exo-MDP** We have introduced the storage control in Section 6.1. See (Powell, 2022; Sinclair et al., 2023a) for a more exhaustive list.

**Inventory control.** In classical inventory models, the endogenous state $x_h$ is the on-hand inventory level, while the exogenous state $\xi_h$ is the demand realization at time $h$ (Madeka et al., 2022). Actions $a_h$ correspond to order quantities. The system dynamics are deterministic given demand, e.g. the newsvendor dynamics $x_{h+1} = f(x_h, a_h, \xi_{h+1}) = \max\{x_h + a_h - \xi_{h+1}, 0\}$. The reward depends on sales revenue and holding or stockout costs, $r(x_h, a_h, \xi_h)$. The only randomness arises from the exogenous demand process, making this a canonical instance of an Exo-MDP.

**Cloud resource allocation.** In cloud computing and service systems, the endogenous state $x_h$ may represent the allocation of resources (e.g., virtual machines, CPU quotas, or bandwidth) across job requests (Sinclair et al., 2023b). The exogenous state $\xi_h$ captures job arrivals at time $h$, which evolve independently of the resource allocation policy. Actions $a_h$ correspond to scheduling decisions, and the reward reflects performance metrics such as throughput or delay penalties. The exogenous job-arrival process drives all stochasticity, while the system dynamics (queue updates, resource usage) are deterministic given arrivals.

### C.2   DISCUSSIONS ON ANCHOR SET

### C.2.1   KNOWLEDGE OF ANCHOR SET

While knowing the anchor set a priori appears strong from a general RL perspective, this assumption is well-motivated in our Exo-MDP setting:

Anchor states are designed by the learner. Assumption 1 requires that we fix a finite collection of post-decision states $x_h^a(n)$ such that the feature matrix $\Phi_h := [\phi(x_h^a(1)), \ldots, \phi(x_h^a(N))]$ has full row rank. These states are the representative grid that the practitioner chooses when constructing the feature map $\phi$, including hat basis, spline knots, tile centers (Sutton et al., 1998). This mirrors standard RL with LFA, where the learner chooses $\phi$ and implicitly chooses the *basis points* on which $\phi$ is built.

This is standard in ADP and matches how Exo-MDPs are implemented in practice. Anchor states are standard in ADP for control and operation research applications Nascimento & Powell (2009; 2013); Powell (2022). In applications like inventory control and storage systems Nascimento & Powell (2009; 2013), practitioners often can exploit *domain knowledge* and *problem structure*, e.g. piecewise-linearity or convexity of the value functions Powell (2022). For instance, with piecewise-linear hat features, anchors are just the breakpoints of the basis functions Nascimento & Powell (2013); Powell (2022). This is also shown in our storage control example in Section 6.2.

### C.2.2 CONNECTIONS TO CORESET AND FRANK-WOLFE PROCEDURES

The anchors in Assumption 1 act as a small, well-conditioned spanning set of feature vectors, analogous in spirit to coresets used in linear RL and ADP. We clarify the connections and differences below.

Anchor sets serve the similar purpose as coresets in linear RL and ADP. In our paper, anchor sets provide a well-conditioned spanning set of feature vectors enabling stable value regression and Bellman transport. This is similar to the concept of coresets or representative set in linear bandits and RL, e.g. Mahalanobis-distance representative sets in Yang & Wang (2019) with well-conditioned feature coverage, optimal design in Lattimore & Szepesvári (2020) with minimal set of points that well-condition the Gram matrix, coreset with well-conditioned feature coverage in Eaton et al. (2025).

Regarding the connections between coreset and Frank-Wolfe (FW) methods, which arises in convex optimization. Specifically, the coreset there represents small subset of points that approximately represents a much larger dataset for the purpose of convex optimization. FW constructs such coresets by iteratively selecting "anchor" points via the linear minimization oracle. This idea has been used in problems such as learning convex bodies Clarkson (2010), sparse convex optimization Jaggi (2013), and practical large-scale machine learning Bachem et al. (2017). However, FW-based coreset construction has largely remained within the convex optimization literature rather than RL settings involving linear function approximation.

In contrast, in our Exo-MDP setting, the anchor set is designed a priori using domain knowledge and problem structure. In many structured control problems they can be constructed a priori, without observing any data. For example, in inventory or resource-storage problems, anchors can be naturally chosen as a grid of storage levels, e.g., extreme low / high and intermediate points, which are natural design points for value approximation.

Extension from domain-driven anchors to algorithmic construction. Our currunt method leverages domain knowlege or problem struture. When such problem-dependent design choice is difficult, it is a promising direction to adapts anchor selection online. We first generate large pool of candidate post-decision states, and adaptively construct the anchors by FW-like methods.

### C.2.3 INVERTABILITY AND WELL CONDITIONEDNESS

Anchors are fully user-selected, and practitioners can exploit domain knowledge or problem structure to construct a well-spread collection of feature vectors. In principle, one can precompute the anchor feature vectors $\phi(x_n^a)$ offline, prior to learning, ensuring that $\Sigma$ is full-rank and well-conditioned. For example, in the storage control experiment in Section 6.2, a simple uniform grid yields $\Sigma = I_N$ and $\lambda_0 = 1$.

Ridge regression, however, is a promising method to improve the invertibility/numerical stability while tradeoffing bias. While we can improve the invertibility of $\Sigma$ by carefully designing the anchors and features, such process can be computationally heavy sometimes. Regularization can replace strict invertibility with a controlled bias term. It trades a small bias controlled by $\beta$ for numerical stability. We now provide a short theoretical sketch showing how a $\beta$-regularizer impacts the regret bound.

The regret bound in Theorem 2 states

$$(\sqrt{N/\lambda_0} + \sqrt{d})|\Xi|H\sqrt{K}.$$

Letting $\lambda_\beta := \lambda_0 + \beta$, then reg. reduces the conditioning part of the regret bound to $(\sqrt{N/\lambda_\beta} + \sqrt{d})|\Xi|H\sqrt{K}$. However, it introduces the additional bias as it solves a reguralized problem

$$\min_w \|\Sigma_h w - y\|_2^2 + \beta\|w\|_2^2.$$

So even if the model is perfectly realizable (no approximation error), we can show that his induces an extra Bellman error term of order $\frac{\beta}{\lambda_\beta}\sqrt{d}$, which then propagates through the horizon and across $K$ episodes. Therefore, the total regret bound is worsened as

$$(\sqrt{N/\lambda_\beta} + \sqrt{d})|\Xi|H\sqrt{K} + \frac{H}{\sqrt{\lambda_\beta}}K\frac{\beta}{\lambda_\beta}\sqrt{d}.$$

In the realizable setting our main theory takes $\beta = 0$, so no extra bias term appears. If one adds a small ridge term $\beta I$ numerical stability, the analysis can be interpreted as introducing an effective inherent Bellman error of size $\epsilon_{\text{ridge}} = \mathcal{O}(\frac{\beta}{\lambda_\beta}\sqrt{d})$, which adds an $\frac{\beta H\sqrt{d}K}{\lambda_\beta^{\frac{3}{2}}}$. Thus ridge reduces only constants in the $\mathcal{O}(\sqrt{K})$ term, but introduces an additional linear-in-$K$ contribution, which vanishes as $\beta \to 0$.

### C.2.4 EXAMPLE WHERE ASSUMPTION 4 HOLDS

**Models.** Consider an storage control Exo-MDP where the endogenous (pre-decision) storage state is $x_h \in [0, R_{\max}]$. At each stage the controller chooses an action $a_h \in \mathcal{A}(x_h, \xi_h) \subset \mathbb{R}$, which produces the post-decision storage

$$x_h^a = \Pi_{[0,R_{\max}]}(x_h + a_h),$$

where $\Pi$ denotes projection onto $[0, R_{\max}]$. After acting, the exogenous state evolves as $\xi_{h+1} \sim \mathbb{P}(\cdot \mid \xi_h)$ and the storage evolves according to

$$x_{h+1} = \Pi_{[0,R_{\max}]}(A(\xi_{h+1})\, x_h^a + b(\xi_{h+1})),$$

with efficiency/retention factor $A(\xi') \in [0, 1]$ and inflow/outflow $b(\xi') \in \mathbb{R}$. The next post-decision storage under policy $\pi$ is then

$$x_{h+1}^a = \Pi_{[0,R_{\max}]}(x_{h+1} + \pi(x_{h+1}, \xi_{h+1})).$$

The one-period reward is a bounded measurable function $r_h(x_h, a_h, \xi_h)$.

**Basis, anchors, and value representation.** Choose storage anchors $0 = \rho_0 < \rho_1 < \cdots < \rho_N = R_{\max}$. Define nonnegative, nodal, partition-of-unity piecewise-linear hat functions $\{\eta_k(\rho)\}_{n=0}^N$, and set

$$\phi(\rho) = (\eta_0(\rho), \ldots, \eta_N(\rho)), \qquad \phi(\rho_n) = e_n.$$

Thus each $\phi(\rho)$ is a convex combination of anchor vectors. The post-decision value is represented using storage-only features and information-dependent weights:

$$V_h^{\pi,a}(x^a, \xi) = \phi(x^a)^\top w_h^\pi(\xi),$$

where $w_h^\pi(\xi) \in \mathbb{R}^{N+1}$ and $[w_h^\pi(\xi)]_n = V_h^{\pi,a}(\rho_n, \xi)$. At the terminal time, weights encode salvage values, e.g. $w_H^\pi(\xi) = 0$ or $[w_H^\pi(\xi)]_n = S(\rho_n, \xi)$.

Recall that Assumption 4 holds if for each $h$, policy $\pi$, and exogenous realization $\xi'$, there exists a storage-feature transport matrix $M_h^\pi(\xi') \in \mathbb{R}^{(N+1)\times(N+1)}$ such that for all post-decision storage states $x^a \in [0, R_{\max}]$,

$$\phi\Big(\Pi(\alpha_{h,\pi}(\xi')\, x^a + \beta_{h,\pi}(\xi'))\Big) = M_h^\pi(\xi')\, \phi(x^a),$$

where $\alpha_{h,\pi}(\xi')$ and $\beta_{h,\pi}(\xi')$ are the coefficients induced by the composition of the storage dynamics and the policy's action, followed by projection. Crucially, $M_h^\pi(\xi')$ does not depend on $x^a$, so the identity holds globally. The weights evolve linearly in expectation over $\xi'$:

$$w_h^\pi(\xi) \;=\; \mathbb{E}_{\xi'\sim\mathbb{P}(\cdot|\xi)}\big[\, M_h^\pi(\xi')^\top \, w_{h+1}^\pi(\xi') \,\big].$$

This formulation is reasonable under the following conditions. First, the post-decision to next pre-decision mapping is affine in $r^a$, possibly followed by clipping. Second, the policy $\pi$ is piecewise-affine in $r$, so that the overall map to $r_{h+1}^a$ is affine with clipping. Third, the storage basis $\phi$ is translation-stable: for any affine map $r \mapsto \Pi(\alpha r + \beta)$ there exists a fixed sparse matrix $S_{\alpha,\beta}$ such that $\phi(\Pi(\alpha r + \beta)) = S_{\alpha,\beta}\phi(r)$ for all $r$. Finally, since $\phi$ forms a partition of unity and clipping corresponds to convex mixing with boundary anchors, each $M_h^\pi(\xi')$ is row-stochastic or sub-stochastic, and therefore non-expansive with $\|M_h^\pi(\xi')\|_\infty \le 1$.

### C.2.5 WHEN ASSUMPTION 3 HOLDS

Assumption 3 requires that every post-decision feature vector can be written as a nonnegative combination of a fixed set of *anchor* feature vectors. This section lists common modeling choices under which the condition is automatically satisfied and gives a simple recipe to enforce it in practice. Assumption 3 aligns with widely used feature constructions in ADP/RL (tabular, hat/spline, histogram, grid/ReLU bases).

**Tabular features.** With one-hot features, each post-decision state corresponds to a standard basis vector, which is in the conic (indeed, convex) hull of the anchor set by construction.

**Storage with piecewise-linear (hat) features.** Let $0 = \rho_0 < \rho_1 < \cdots < \rho_N = R_{\max}$ be storage anchors and define nonnegative, nodal, partition-of-unity hat functions $\{\eta_n\}_{n=0}^N$. Set $\phi(\rho) = (\eta_0(\rho), \ldots, \eta_N(\rho))$ so that $\phi(\rho_n) = e_n$ and $\sum_n \eta_n(\rho) = 1$ for all $\rho$. For any post-decision level $x^a \in [0, R_{\max}]$, we have $\phi(x^a) = \sum_n \eta_n(x^a)\,\phi(\rho_n)$ with $\eta_n(x^a) \ge 0$, so $\phi(x^a)$ lies in the conic hull of the anchor features (in fact, in their convex hull). Clipping at the bounds $0$ and $R_{\max}$ simply mixes with boundary anchors and preserves nonnegativity.

**Histogram / indicator bases.** If $\phi$ is formed by nonoverlapping (or softly overlapping) nonnegative basis functions that sum to at most one (e.g., bin indicators or triangular kernels), then $\phi(x^a)$ is a nonnegative combination of the anchor features obtained by placing anchors at the bin centers or knot points.

**B-splines and ReLU tiles.** Nonnegative partition-of-unity spline bases (e.g., linear B-splines) and grid-based ReLU "tiles" yield $\phi(x^a)$ with nonnegative entries and local support. Choosing anchors at the knots/cell corners makes $\phi(x^a)$ a nonnegative combination of anchor feature vectors.

To ensure Assumption 3: (i) include boundary anchors so that clipping/projection maps to anchors; (ii) use nonnegative, locally supported basis functions that form (approximate) partitions of unity over the post-decision domain; (iii) place anchors at the basis nodes (knots, cell corners, or representative states) so that $\phi(\text{state})$ is a sparse nonnegative combination of anchor columns. If a signed feature map is preferred (e.g., mean-centered features), a standard fix is a *nonnegative lifting* $\tilde{\phi} = [\phi_+;\ \phi_-]$ with $\phi_+ = \max\{\phi, 0\}$ and $\phi_- = \max\{-\phi, 0\}$; placing anchors on the lifted coordinates restores the cone property.

### C.2.6 WHEN THE BOUND $\sup_{\pi,\xi,t} \|M_h^\pi(\xi)\|_2 \le 1$ HOLDS

Recall under Assumption 2 or Assumption 4 that for each $(h, \pi, \xi')$ one builds a mixing matrix

$$M_h^\pi(\xi') \in \mathbb{R}^{(N+1)\times(N+1)},$$

whose $n$-th row contains the interpolation weights $\beta_{nj}(\xi', \pi) \ge 0$ (usually two nonzeros) taking the anchor $\rho_n$ to the next post-decision storage $r_{h+1}^a$ and then back onto the anchor grid. Thus each row sums to 1 (row-stochastic; sub-stochastic at the capacity boundaries when clipping pins to $\rho_0$ or $\rho_N$). We provide some sufficient conditions for $\|M_h^\pi(\xi')\|_2 \le 1$ below.

*Lipschitz-in-storage dynamics with hat basis.* If the continuous map $T(r) = \Pi(\alpha r + \beta)$ is 1-Lipschitz (i.e., $|\alpha| \le 1$) and functions of $r$ are represented on a uniform grid with nodal PLC interpolation, then the discrete composition operator interpolate $\circ\, T$ is nonexpansive on grid values under the Euclidean

norm. This operator is exactly $M_h^\pi(\xi')^\top$, hence $\|M_h^\pi(\xi')\|_2 \leq 1$. Intuitively, 1-Lipschitz maps do not increase distances between storage levels; interpolation preserves (and slightly underestimates) distances, so the induced linear map is nonexpansive.

*Doubly (sub-)stochastic mixing.* If every $M_h^\pi(\xi')$ is row-stochastic and column-sub-stochastic (all column sums $\leq 1$), then

$$\|M_h^\pi(\xi')\|_2 \ \leq \ \sqrt{\|M_h^\pi(\xi')\|_1 \|M_h^\pi(\xi')\|_\infty} \ \leq \ \sqrt{1 \cdot 1} \ = \ 1.$$

Column-sub-stochasticity holds, for example, if the one-step map in storage is monotone and nonexpansive: $r^a \mapsto \Pi(\alpha r^a + \beta)$ with $|\alpha| \leq 1$, and the basis is nodal hat (partition-of-unity) features on a uniform grid. Each anchor's "mass" spreads to at most two neighbors without duplication, and clipping removes mass near the boundaries.

*Decomposition into contractions.* If the mixing matrix can be expressed as a convex combination of contractions,

$$M_h^\pi(\xi') = \sum_\ell \gamma_\ell T_\ell, \qquad \gamma_\ell \geq 0, \ \sum_\ell \gamma_\ell = 1, \ \ \|T_\ell\|_2 \leq 1,$$

then by subadditivity and convexity of the operator norm, $\|M_h^\pi(\xi')\|_2 \leq 1$. Two useful instances are: *Permutation/shift structure:* when the map is a grid shift or clipping, each $T_\ell$ is a permutation (possibly composed with a boundary projector), hence $\|T_\ell\|_2 = 1$. *Row-weighted permutations:* if $M = \sum_\ell D_\ell \Pi_\ell$ with $\Pi_\ell$ permutations and $D_\ell$ diagonal with entries in $[0,1]$, then $\|M\|_2 \leq \sum_\ell \|D_\ell\|_2 \leq \sum_\ell \max_i (D_\ell)_{ii}$. If the row-wise weights over $\ell$ sum to $\leq 1$, the bound is $\leq 1$.

*Doubly-stochastic special case.* If columns also sum to 1 (e.g., pure permutations, or measure-preserving monotone maps without clipping on a periodic grid), then $M$ is doubly stochastic and $\|M\|_2 \leq 1$ with equality only if $M$ is a permutation.

Furthermore, we provide some methods to check or enforce the assumption. Empirically, one can draw a batch of $\xi' \sim Q(\cdot \mid \xi)$, build $M_h^\pi(\xi')$, and compute the largest singular value $\sigma_{\max}$, verifying $\max \sigma_{\max} \leq 1$ (allowing numerical tolerance). Design-wise, one can ensure nonexpansiveness by using uniform nodal hat features (partition of unity), storage dynamics with $|\alpha| \leq 1$, and capacity clipping. If some scenarios have $|\alpha| > 1$ (expansive), increase grid resolution or add a smoothing step (row-wise convex averaging) that preserves row sums, making $M$ a contraction. For non-uniform grids or unusual features, "whitening" each local two-anchor block (normalizing columns per cell) enforces contraction while preserving row sums.

Under storage-only anchors and nodal, nonnegative, partition-of-unity hat basis, and with standard storage dynamics (affine + clipping) satisfying $|\alpha| \leq 1$, the transport matrices $M_h^\pi(\xi')$ are row-stochastic and column-sub-stochastic. Hence $\sup_{\pi,\xi,h} \|M_h^\pi(\xi)\|_2 \leq 1$. This can be verified numerically, and if needed enforced by smoothing or per-cell normalization without altering the PLC interpolation semantics.

**Connections to Nascimento & Powell (2013).** Under the modeling assumptions in Nascimento & Powell (2013) the bound is justified when one implements the storage-only anchor/hat-basis scheme. Nascimento & Powell (2013) works in post-decision form and shows that, for each information state, the value function in the scalar storage is piecewise-linear concave with breakpoints. Each period's decision is obtained from a deterministic linear program with vector-valued control, and the algorithm maintains concavity of slopes via projection. This is exactly the setting where one uses storage-only anchors $\{\rho_n\}$ and nodal hat features. The storage dynamics between periods are affine plus clipping: the model introduces exogenous changes in storage in post-decision form, so that the next storage is an additive update (possibly with losses) followed by projection to capacity. This map is 1-Lipschitz in the storage variable.

With nodal, nonnegative, partition-of-unity hat functions on $\{\rho_n\}$, the push-forward and interpolation step from an anchor $\rho_n$ produces a row-stochastic mixing row (two nonzeros in one dimension). Collecting these rows defines the matrix $M_h^\pi(\xi)$. Because the underlying continuous map is 1-Lipschitz and interpolation is stable, the induced discrete operator on nodal values is nonexpansive in the Euclidean norm, hence $\|M_h^\pi(\xi)\|_2 \leq 1$. At capacity boundaries, clipping only reduces distances, so the bound continues to hold. This is consistent with the PLC/anchor structure and concavity projection used in the paper. It should be noted that the paper does not phrase its analysis in terms of an $M$ matrix or a spectral-norm bound. Instead, it proceeds via a dynamic programming operator on

slope vectors with technical conditions ensuring monotonicity, continuity, and convergence. Thus the spectral-norm assumption is an implied property of the standard discretization, rather than a stated theorem.

In summary, for Nascimento & Powell (2013), with the standard storage law (additive exogenous changes with clipping) and the PLC/anchor representation, the discretization induces row-stochastic (and nonexpansive) mixing operators. Therefore it is reasonable and consistent to assume $\sup_{\pi,\xi,h} \|M_h^\pi(\xi)\|_2 \leq 1$, even though the paper establishes convergence via slope-operator monotonicity and continuity rather than an explicit spectral-norm bound.

### C.2.7 WEAKENING AND RELAXATIONS

**Adaptive anchors selection via coreset/Frank-Wolfe style procedures.** The anchors can be viewed as a small, hand-picked coreset for the endogenous space. In principle, we can learn this coreset from data via coreset/Frank-Wolfe style procedures and then running LSVI-PE on the resulting anchors. Analyzing such a scheme requires a second layer of error control between the data-driven anchors and the optimal anchor set and is beyond the scope of the present work, but we now mention this as a promising direction in the Conclusion.

**Approximate closure assumptions via inherent Bellman error.** The realizable analysis (Theorems 2–3) uses Assumptions 2–3 (or 4) to guarantee exact closure of Bellman updates in the anchor span. However, our agnostic analysis (Theorems 4–5) only requires the basic Assumption 1 plus a finite inherent Bellman error. In other words, even if the post-decision values are only approximately representable by the anchor features and the cone/closure conditions only hold approximately, the regret bound still holds with an additional $O(K\varepsilon_{\mathrm{BE}})$ bias term. This already provides a quantitative weakening: violations of Assumptions 2–3 are absorbed into $\varepsilon_{\mathrm{BE}}$, rather than being ruled out outright.

### C.3 ADDITIONAL DISCUSSIONS

$m$**-Markovian exogenous process.** Our framework extends to exogenous processes with finite memory. Specifically, we assume that the exogenous state follows a $m$-Markov model: at time $h$, the augmented state includes the endogenous component $x_h$ together with the last $m$ exogenous states,

$$s_h = \big(x_h, \xi_{h-m}, \ldots, \xi_h\big).$$

The next exogenous state $\xi_{h+1}$ is drawn from a conditional distribution that depends only on the most recent $k$ exogenous states:

$$\xi_{h+1} \sim \mathbb{P}\big(\cdot \mid \xi_{h-m}, \ldots, \xi_h\big).$$

This formulation strictly generalizes the i.i.d. and first-order Markov settings while retaining a compact representation that captures temporal correlations in the exogenous sequence.

## D OMITTED DISCUSSION IN SECTION 4

Here we outline the application of PEL (and FTL) to the simpler tabular Exo-MDP settings.

### D.1 FTL FOR TABULAR EXO-MDPS

As discussed in Section 4, one can extend the FTL principle to finite-horizon Exo-MDPs with finite state and action spaces. For any deterministic policy $\pi$, using the exogenous traces $\{\boldsymbol{\xi}^1, \ldots, \boldsymbol{\xi}^{k-1}\}$ collected up to episode $k$, we can form the unbiased empirical value estimator:

$$\widetilde{V}_1^{k,\pi}(s_1) := \frac{1}{k-1} \sum_{l=1}^{k-1} V_1^\pi\big(s_1, \boldsymbol{\xi}_{>1}^l\big) = \frac{1}{k-1} \sum_{l=1}^{k-1} \sum_{h=1}^{H} r(x_h, \pi_h(s_h), \xi_h^l),$$

where the transitions take the form

$$s_{h+1} = (x_{h+1}, \xi_{h+1}^l), \qquad x_{h+1} = f(x_h, a_h, \xi_{h+1}^l).$$

The `FTL` algorithm then selects the greedy policy in episode $k$ with respect to these empirical value estimates:

$$\tilde{\pi}^k \in \arg\max_{\pi \in \Pi} \widetilde{V}_1^{k,\pi}(s_1).$$

This construction *crucially* leverages the fact that the exogenous trace distribution $\boldsymbol{\xi}$ is independent of the agent's actions. Hence, every exogenous trace can be reused to evaluate *all* candidate policies without bias, a property that enables policy-level `FTL` in Exo-MDPs and sharply contrasts with general MDPs where action-dependent transitions break this replay.

The following proposition is a restatement of known ERM/FTL-style guarantees in this setting. Note, however, that the computational cost of an unconstrained search over $\Pi$ can be prohibitive.

**Proposition 3.** [`FTL` guarantee, Theorem 7 in Sinclair et al. (2023b)] For any $\delta \in (0,1)$, with probability at least $1 - \delta$,

$$\text{SR}\left(\text{FTL}, K\right) \leq H\sqrt{\frac{2\log(2|\Pi|/\delta)}{K}}.$$

In the tabular case this gives the stated dependence $|\Pi| \leq A^{H|\mathcal{X}||\Xi|}$.

Motivated by the proof of regret bound of `FTL` for Exo-MAB, we also provide the expected regret bound of `FTL` for Exo-MDP.

**Proposition 4.** The expected regret of `FTL` can be bounded as

$$\mathbb{E}[\text{SR}\left(\text{FTL}, K\right)] \leq \sqrt{\frac{H^2 \log |\Pi|}{K}}.$$

In the tabular case this gives the stated dependence $|\Pi| \leq A^{H|\mathcal{X}||\Xi|}$.

Thus, while the statistical guarantees for `FTL` are strong, the algorithm is computationally infeasible in practice due to the exponential size of the policy space. This motivates more efficient implementations of `PEL` that avoid enumerating $\Pi$. In particular, one can estimate the exogenous transition model directly and then apply dynamic programming to compute greedy policies—an approach we refer to as *Predict-Then-Optimize (PTO)* in Section 4.

### D.2 PTO UNDER GENERAL $m$-MARKOVIAN CASE

For the general Markovian setting, `PTO` learns the transition model $\widehat{\mathbb{P}}\left(\xi_h \mid \boldsymbol{\xi}_{h-1}\right)$ to approximate the true distribution $\mathbb{P}\left(\xi_h \mid \boldsymbol{\xi}_{h-1}\right)$. `PTO` uses the model $\widehat{\mathbb{P}}\left(\xi_h \mid \boldsymbol{\xi}_{h-1}\right)$ to solve the Bellman equation. `PTO` uses the maximum likelihood estimator of transition model, which is the empirical distribution

$$\widehat{\mathbb{P}}\left(\xi_h \mid \boldsymbol{\xi}_{h-1}\right) := \sum_{l=1}^{k-1} \mathbb{I}\left\{\boldsymbol{\xi}_{h-1}^l = \boldsymbol{\xi}_{h-1}, \xi_h^l = \xi_h\right\} \Big/ \sum_{l=1}^{k-1} \mathbb{I}\left\{\boldsymbol{\xi}_{h-1}^l = \boldsymbol{\xi}_{h-1}\right\}$$

to solve the Bellman equation

$$\widehat{Q}_h(s_h, a_h) := \mathbb{E}_{\xi_h|\boldsymbol{\xi}_{h-1}}\left[r(x_h, a_h, \xi_h) + \widehat{V}_{h+1}\left(f(x_h, a_h, \xi_h), \boldsymbol{\xi}_h\right) \mid \widehat{\mathbb{P}}\right]$$

$$=: \hat{\mathbb{E}}_{\xi_h|\boldsymbol{\xi}_{h-1}}\left[r(x_h, a_h, \xi_h) + \widehat{V}_{h+1}\left(f(x_h, a_h, \xi_h), \boldsymbol{\xi}_h\right)\right]$$

$$\widehat{V}_h(s_h) := \max_{a_h \in \mathcal{A}} \widehat{Q}_h(s_h, a_h)$$

$$\widehat{\pi}_h(s_h) := \arg\max_{a_h \in \mathcal{A}} \widehat{Q}_h(s_h, a_h),$$

where $s_h = (x_h, \boldsymbol{\xi}_{h-1})$. Note that the size of policy set $|\Pi|$ depends on the $m$

$$|\Pi| = \prod_{h=1}^{H} |\Pi_h| = \begin{cases} \prod_{h=1}^{H} A^{|\mathcal{X}|} = A^{H|\mathcal{X}|}, m = 0, \\ \prod_{h=1}^{H} A^{|\mathcal{X}||\Xi|} = A^{H|\mathcal{X}||\Xi|}, m = 1, \\ \prod_{h=1}^{H} A^{|\mathcal{X}||\Xi|^{h-1}} = A^{|\mathcal{X}|\sum_{h=1}^{H} |\Xi|^{h-1}} = \mathcal{O}(A^{|\mathcal{X}||\Xi|^{H-1}}), m = H. \end{cases}$$

**Proposition 5** (Theorem 6 in Sinclair et al. (2023b)). Suppose that

$$\sup_{h\in[T],\boldsymbol{\xi}_{<h}\in\Xi^{[h-1]}} \left\|\widehat{\mathbb{P}}\left(\cdot\mid\boldsymbol{\xi}_{<t}\right)-\mathbb{P}\left(\cdot\mid\boldsymbol{\xi}_{<t}\right)\right\|_1 \le \epsilon.$$

Then we have that

$$\text{SR}\left(\hat{\pi},K\right)\le H^2\epsilon.$$

In addition, if each $\xi_h$ is independent from $\boldsymbol{\xi}_{<h}$, then $\forall \delta\in(0,1)$, with probability at least $1-\delta$

$$\text{SR}\left(\hat{\pi},K\right)\le H^2\sqrt{\frac{2|\Xi|\log(2H/\delta)}{K}}.$$

Therefore, the regret of `PTO` can be bounded as follows.

**Corollary 3.** Fix $\delta\in(0,1)$. with probability at least $1-\delta$,

$$\text{SR}\left(\text{FTL},K\right)\le\begin{cases}H\sqrt{\frac{2H|\mathcal{X}|\log(A/\delta)}{K}}, m=0,\\ H\sqrt{\frac{2H|\mathcal{X}||\Xi|\log(A/\delta)}{K}}, m=1,\\ H\sqrt{\frac{2H|\mathcal{X}||\Xi|^{H-1}\log(A/\delta)}{K}}, m=H.\end{cases}$$

**Corollary 4.**

$$\mathbb{E}[\text{SR}\left(\text{FTL},K\right)]\le\begin{cases}H\sqrt{\frac{H|\mathcal{X}|\log(A)}{K}}, m=0,\\ H\sqrt{\frac{H|\mathcal{X}||\Xi|\log(A)}{K}}, m=1,\\ H\sqrt{\frac{H|\mathcal{X}||\Xi|^{H-1}\log(A)}{K}}, m=H.\end{cases}$$

*Proof of Proposition 5.* $\widehat{Q}_h$ and $\widehat{V}_h$ refer to the $Q$ and $V$ values for the optimal policy in $\widehat{M}$ where the exogenous input distribution is replaced by its estimate $\widehat{\mathbb{P}}(\cdot\mid\boldsymbol{\xi}_{h-1})$. Denote by $\widehat{V}_h^\pi$ as the value function for some policy $\pi$ in the MDP $\widehat{M}$. Then $\widehat{V}_h^{\hat{\pi}}=\widehat{V}_h$ by construction.

$$\begin{aligned}\text{SR}\left(\hat{\pi},K\right)&=V_1^\star(s_1)-V_1^{\hat{\pi}}(s_1)\\&=V_1^\star(s_1)-\widehat{V}_1^{\pi^\star}(s_1)+\widehat{V}_1^{\pi^\star}(s_1)-\widehat{V}_1(s_1)+\widehat{V}_1(s_1)-V_1^{\hat{\pi}}(s_1)\\&\le 2\sup_\pi\left|V_1^\pi(s_1)-\widehat{V}_1^\pi(s_1)\right|.\end{aligned}$$

By the simulation lemma, it is bounded above by $\frac{H^2}{2}\max_{s,a,h}|P_h(s,a)-\hat{P}_h(s,a)|$. Since $P_h(\cdot|s,a)$ is the pushfoward measure of $\mathbb{P}\left(\cdot\mid\xi_{h-1}\right)$ under mapping $f$

$$P_h(s'\in\cdot|s,a)=P_h(f(x,a,\xi)\in\cdot|s,a)=\mathbb{P}\left(f^{-1}(s,a,\cdot)\mid\xi_{h-1}\right),$$

we have (since $f$ is function)

$$\left\|P_h(s,a)-\widehat{P}_h(s,a)\right\|_1\le\left\|\widehat{\mathbb{P}}\left(\cdot\mid\xi_{h-1}\right)-\mathbb{P}\left(\cdot\mid\xi_{h-1}\right)\right\|_1$$

and thus

$$\max_{s,a,h}\left\|P_h(s,a)-\widehat{P}_h(s,a)\right\|_1\le\max_{h,\xi_{h-1}}\left\|\widehat{\mathbb{P}}\left(\cdot\mid\xi_{h-1}\right)-\mathbb{P}\left(\cdot\mid\xi_{h-1}\right)\right\|_1.$$

Then the proof for the first part is finished

$$\text{SR}\left(\hat{\pi},K\right)\le H^2\max_{h,\xi_{h-1}}\left\|\widehat{\mathbb{P}}\left(\cdot\mid\xi_{h-1}\right)-\mathbb{P}\left(\cdot\mid\xi_{h-1}\right)\right\|_1.$$

Now suppose that $\boldsymbol{\xi}\sim\mathbb{P}$ has each $\xi_h$ independent from $\xi_{h-1}$ and let $\widehat{\mathbb{P}}$ be the empirical distribution. Using the $\ell_1$ concentration bound shows that the event

$$\mathcal{E}=\left\{\forall h:\left\|\widehat{\mathbb{P}}(\xi_h\in\cdot)-\mathbb{P}(\xi_h\in\cdot)\right\|_1\le\sqrt{\frac{2|\Xi|\log(H/\delta)}{K}}\right\}$$

occurs with probability at least $1-\delta$. Under $\mathcal{E}$ we then have that:

$$\max_{h\in[H],\boldsymbol{\xi}_{h-1}\in\Xi^{[h-1]}}\left\|\widehat{\mathbb{P}}\left(\cdot\mid\boldsymbol{\xi}_{h-1}\right)-\mathbb{P}\left(\cdot\mid\boldsymbol{\xi}_{h-1}\right)\right\|_1\le\sqrt{\frac{2|\Xi|\log(H/\delta)}{K}}.$$

Taking this in the previous result shows the claim. $\square$

*Remark* 2. The quadratic horizon multiplicative factor $\mathcal{O}(H^2)$ in regret is due to compounding errors in the distribution shift. In the worst case, $\epsilon$ can scale as $\mathcal{O}(|\Xi|^T)$ if each $\xi_h$ is correlated with $\boldsymbol{\xi}_{h-1}$.

*Remark* 3. Proposition 5 is not valid for the $m$-Markovian case. A straightforward extension of the proof for Exo-Bandit is not valid since

$$V^{*,\mathcal{M}} = \max_{\pi} V^{\pi,\mathcal{M}} \neq \max_{\pi} \mathbb{E}[V^{\pi,\widehat{\mathcal{M}}}] \leq \mathbb{E}[\max_{\pi} V^{\pi,\widehat{\mathcal{M}}}] = \mathbb{E}[V^{\hat{\pi},\widehat{\mathcal{M}}}].$$

The inequality is due to that the value function is nonlinear in $P$ and $\hat{P}_h \not\perp \hat{P}_{t'}$ for $t \neq t'$. In particular,

$$\mathbb{E}[\widehat{V}_h] = \mathbb{E}[r_h + \widehat{P}_h \widehat{V}_{h+1}] = r_h + \mathbb{E}[\widehat{P}_h(r_{h+1} + \widehat{P}_{h+1}\widehat{V}_{t+2})] = r_h + P_h r_{h+1} + \mathbb{E}[\widehat{P}_h \widehat{P}_{h+1} \widehat{V}_{t+2}]$$
$$\neq r_h + P_h r_{h+1} + P_h P_{h+1} V_{t+2}.$$

### D.3 REGRET BOUNDS OF OPTIMISM-BASED METHODS FOR TABULAR EXO-MDPS

#### D.3.1 REGRET BOUND OF UCB FOR EXO-MAB

**Proposition 6** (UCB for Exo-MAB). The expected cumulative regret of UCB in the full information setting with $A$ arms satisfies

$$\text{CR}\,(\text{UCB}, K) \leq \sqrt{2\sigma^2 \log(AK^2)(K-1)} + \mathcal{O}(1).$$

*Proof.* With prob. at least $1 - \delta$, the event $E$ holds

$$\forall a \in [A], \forall k \in [K], |\mu_i - \hat{\mu}_i(k)| \leq b_i(k) := \sqrt{2\sigma^2 \frac{\log(AK/\delta)}{k-1}}.$$

Conditioned on event $E$, the simple regret can be bounded as

$$\text{SR}\,(\text{UCB}, k) = \mu^\star - \mu_{a_k} \leq \bar{\mu}_1(k) - \mu_{a_k} \leq \bar{\mu}_{a_h}(k) - \mu_{a_k} \leq 2b_{a_h}(k) = 2\sqrt{2\sigma^2 \frac{\log(AK/\delta)}{k-1}}.$$

The expected simple regret is bounded as

$$\text{SR}\,(\text{UCB}, k) = \mathbb{E}[\mu^\star - \mu_{a_k}] = \mathbb{E}[\mu^\star - \mu_{a_k}|E]\mathbb{P}(E) + \mathbb{E}[\mu^\star - \mu_{a_k}|E^c]\mathbb{P}(E^c) \leq 2\sqrt{2\sigma^2 \frac{\log(AK/\delta)}{k-1}} + \delta.$$

Therefore, the expected total regret

$$\text{CR}\,(\text{UCB}, K) \leq \sum_{t=2}^{K} 2\sqrt{2\sigma^2 \frac{\log(AK/\delta)}{k-1}} + \delta \leq \sqrt{2\sigma^2 \log(AK/\delta)(K-1)} + K\delta.$$

Choosing $\delta = 1/K$ yields

$$\text{CR}\,(\text{UCB}, K) \leq \sqrt{2\sigma^2 \log(AK^2)(K-1)} + \mathcal{O}(1)$$
$$\leq \mathcal{O}(\sigma\sqrt{K\log A}) + \mathcal{O}(\sigma\sqrt{K\log K}).$$

$\square$

#### D.3.2 REGRET BOUND OF OPTIMISTIC PTO FOR TABULAR EXO-MDP

We consider `PTO-Opt`, an optimistic version of `PTO`, which replaces the exogenous transition model with its optimistic version. In episode $k$, `PTO-Opt` performs

$$\bar{Q}_h^k(s_h, a_h) := r(x_h, a_h, \xi_h) + \mathbb{E}_{\xi_{h+1}|\xi_h}\left[\bar{V}_{h+1}^k(f(x_h, a_h, \xi_{h+1}), \xi_{h+1}); \bar{\mathbb{P}}^k\right]$$
$$= r(x_h, a_h, \xi_h) + \max_{Q_h : \left\|Q_t - \hat{\mathbb{P}}_h^k(\xi)\right\|_1 \leq c_t(\xi)} \sum_{\xi'} Q_h(\xi')\bar{V}_{h+1}^k(f(x_h, a_h, \xi_{h+1}), \xi_{h+1}),$$
$$\bar{\pi}_h^k(s_h) \in \arg\max_{a_h} \bar{Q}_h^k(s_h, a_h), \quad \bar{V}_h^k(s_h) := \bar{Q}_h^k(s_h, \bar{\pi}_h^k(s_h)).$$

**Proposition 7** (High probability cumulative regret bound of `PTO-Opt`). Fix any $\delta \in (0, 1)$. With probability at least $1 - \delta$,

$$\text{CR}\,(\text{PTO-Opt}, K) \leq \mathcal{O}(H^2|\Xi|\sqrt{K \log(KH|\Xi|/\delta)}).$$

Compared with Theorem 1, `PTO-Opt` has slightly worse regret bound. This verifies that `PEL` is sufficient for tabular Exo-MDP with simple implementations.

# E   OMITTED DISCUSSION IN SECTION 5

## E.1   LSVI-PE WITH MISSPECIFICATION (APPROXIMATION) ERROR.

Here we consider the case where the function class is misspecified and the true value functions may not lie exactly in the linear span. To capture this, we introduce the notion of post–decision Bellman operators.

Write $x' := g(x^a, \xi')$. For any $U_{h+1} : \mathcal{X} \times \Xi \to \mathbb{R}$,

$$(\mathcal{T}^\pi U_{h+1})(x^a, \xi) := \mathbb{E}_{\xi' \sim P_h(\cdot|\xi)}\Big[ r\big(x', \pi(x', \xi'), \xi'\big) + U_{h+1}\big(\mathcal{T}_h^\pi(\xi')(x^a),\, \xi'\big)\Big],$$

$$(\mathcal{T} U_{h+1})(x^a, \xi) := \mathbb{E}_{\xi' \sim P_h(\cdot|\xi)}\Big[ \max_{a' \in \mathcal{A}} \big\{ r(x', a', \xi') + U_{h+1}\big(f^a(x', a'),\, \xi'\big)\big\}\Big].$$

Let $\mathcal{F}_h := \{(x^a, \xi) \mapsto \phi(x^a)^\top w_h(\xi) : w_h(\xi) \in \mathbb{R}^d\}$ be the post-decision linear class at stage $h$.

We then have the Bellman errors or approximation errors as follows:

**Definition 3** (Inherent Bellman error). Define the (post-decision) inherent Bellman errors

$$\varepsilon_{\mathrm{BE}}^\pi := \max_{h \in [H]} \sup_{\xi \in \Xi} \sup_{U_{h+1} \in \mathcal{F}_{h+1}} \inf_{W_h \in \mathcal{F}_h} \sup_{x^a} \big|(T^\pi U_{h+1})(x^a, \xi) - W_h(x^a, \xi)\big|,$$

$$\varepsilon_{\mathrm{BE}}^{\max} := \max_{h \in [H]} \sup_{\xi \in \Xi} \sup_{U_{h+1} \in \mathcal{F}_{h+1}} \inf_{W_h \in \mathcal{F}_h} \sup_{x^a} \big|(\mathcal{T} U_{h+1})(x^a, \xi) - W_h(x^a, \xi)\big|.$$

We will use $\varepsilon_{\mathrm{BE}} := \max\{\varepsilon_{\mathrm{BE}}^{\pi^\star}, \varepsilon_{\mathrm{BE}}^{\max}\}$.

**Theorem 4.** Assume Assumption 1 holds. Fix $\delta \in (0, 1)$. Then with probability at least $1 - \delta$,

$$\mathrm{CR}\,(\texttt{LSVI-PE}, K) \;\leq\; \tilde{\mathcal{O}}\left(\Big(H + \sqrt{\tfrac{N}{\lambda_0}}\Big)|\Xi|H\sqrt{K} \;+\; \frac{H}{\sqrt{\lambda_0}}\,K\,\varepsilon_{\mathrm{BE}}\right).$$

Compared to the realizable case, the regret bound now includes an additional bias term, linear in $K$, that scales with the inherent Bellman error $\varepsilon_{\mathrm{BE}}$. This term is unavoidable in general agnostic settings: if $\varepsilon_{\mathrm{BE}} > 0$ is fixed, even an oracle learner suffers an $O(K\varepsilon_{\mathrm{BE}})$ cumulative bias (Zanette et al., 2020).

# F   PROOFS OF REGRET BOUNDS IN SECTION 4

## F.1   EXO-BANDITS

**Proposition 1.** Assume rewards are $\sigma^2$-sub-Gaussian. Then the expected per-round simple regret of $\texttt{FTL}$ satisfies $\mathrm{SR}\,(\texttt{FTL}, k) \leq \sqrt{\frac{2\sigma^2 \log A}{k-1}}$, and consequently the cumulative regret obeys $\mathrm{CR}\,(\texttt{FTL}, K) \leq 2\sigma\sqrt{(K-1)\log A}$.

To show the result we start with the following lemma.

**Lemma 1** (Maxima of sub-Gaussian random variables). Let $X_1, \ldots, X_n$ be independent $\sigma^2$-sub-Gaussian random variables. Then

$$\mathbb{E}\left[\max_{1 \leq i \leq n} X_i\right] \leq \sqrt{2\sigma^2 \log n}$$

and, for every $t > 0$,

$$\mathbb{P}\left\{\max_{1 \leq i \leq n} X_i \geq \sqrt{2\sigma^2(\log n + t)}\right\} \leq e^{-t},$$

or equivalently

$$\mathbb{P}\left\{\max_{1 \leq i \leq n} X_i \geq \sqrt{2\sigma^2 \log(n/\delta)}\right\} \leq \delta,$$

*Proof.* The first part is quite standard: by Jensen's inequality, monotonicity of exp, and $\sigma^2$-subgaussianity, we have, for every $\lambda > 0$,

$$e^{\lambda \mathbb{E}\left[\max_{1 \leq i \leq n} X_i\right]} \leq \mathbb{E}e^{\lambda \max_{1 \leq i \leq n} X_i} = \max_{1 \leq i \leq n} \mathbb{E}e^{\lambda X_i} \leq \sum_{i=1}^n \mathbb{E}e^{\lambda X_i} \leq n e^{\frac{\sigma^2 \lambda^2}{2}}$$

so, taking logarithms and reorganizing, we have

$$\mathbb{E}\left[\max_{1\leq i\leq n} X_i\right] \leq \frac{1}{\lambda}\ln n + \frac{\lambda\sigma^2}{2}.$$

Choosing $\lambda := \sqrt{\frac{2\ln n}{\sigma^2}}$ proves the first inequality. Turning to the second inequality, let $u := \sqrt{2\sigma^2(\log n + t)}$. We have

$$\mathbb{P}\left\{\max_{1\leq i\leq n} X_i \geq u\right\} = \mathbb{P}\left\{\exists i, X_i \geq u\right\} \leq \sum_{i=1}^{n}\mathbb{P}\left\{X_i \geq u\right\} \leq ne^{-\frac{u^2}{2\sigma^2}} = e^{-t}$$

the last equality recalling our setting of $u$. $\square$

Now we provide the proof of Proposition 1.

*Proof.* Observe that the empirical mean is unbiased for each arm at each round,

$$\mathrm{SR}\left(\mathrm{FTL}, k\right) = \mu^\star - \mathbb{E}[\mu_{a_k}] = \max_a \mathbb{E}[\mu_a - \mu_{a_k}] = \max_a \mathbb{E}[\hat{\mu}_a(k) - \mu_{a_k}] \leq \mathbb{E}[\max_a \hat{\mu}_a(k) - \mu_{a_k}]$$

$$= \mathbb{E}[\hat{\mu}_{a_k}(k) - \mu_{a_k}]$$

$$\leq \mathbb{E}[\max_{a\in[A]} \hat{\mu}_a(k) - \mu_a]$$

$$\leq \sqrt{2\sigma^2 \log A/(k-1)},$$

where the last inequality is due to Lemma 1. Therefore, we have

$$\mathrm{CR}\left(\mathrm{FTL}, K\right) = \sum_{k=1}^{K}\mathrm{SR}\left(\mathrm{FTL}, k\right) \leq \sum_{t=2}^{K}\sqrt{2\sigma^2 \log A/(k-1)} \leq 2\sigma\sqrt{(K-1)\log A}.$$

$\square$

### F.2 TABULAR EXO-MDP

**Proposition 3.** [FTL guarantee, Theorem 7 in Sinclair et al. (2023b)] For any $\delta \in (0,1)$, with probability at least $1 - \delta$,

$$\mathrm{SR}\left(\mathrm{FTL}, K\right) \leq H\sqrt{\frac{2\log(2|\Pi|/\delta)}{K}}.$$

In the tabular case this gives the stated dependence $|\Pi| \leq A^{H|\mathcal{X}||\Xi|}$.

*Proof.* Observe that $V_1^\pi\left(s_1, \boldsymbol{\xi}^k\right)$ are iid r.v.s, each of which has mean $V_1^\pi\left(s_1\right)$. Using Hoeffding's inequality and a union bound over all policies shows that the event

$$\mathcal{E} = \left\{\forall \pi \in \Pi : \left|V_1^\pi\left(s_1\right) - \overline{\mathbb{E}}\left[V_1^\pi\left(s_1\right)\right]\right| \leq \sqrt{\frac{H^2\log(2|\Pi|/\delta)}{2K}}\right\}$$

occurs with probability at least $1 - \delta$. Under $\mathcal{E}$ we then have

$$\mathrm{SR}\left(\mathrm{FTL}, K\right) = V_1^{\pi^\star}\left(s_1\right) - V_1^{\hat{\pi}^k}\left(s_1\right)$$

$$= V_1^{\pi^\star}\left(s_1\right) - \overline{\mathbb{E}}\left[V_1^{\pi^\star}\left(s_1, \boldsymbol{\xi}\right)\right] + \overline{\mathbb{E}}\left[V_1^{\pi^\star}\left(s_1, \boldsymbol{\xi}\right)\right] - \overline{\mathbb{E}}\left[V_1^{\hat{\pi}^k}\left(s_1, \boldsymbol{\xi}\right)\right]$$

$$+ \overline{\mathbb{E}}\left[V_1^{\hat{\pi}^k}\left(s_1, \boldsymbol{\xi}\right)\right] - V_1^{\hat{\pi}^k}\left(s_1\right)$$

$$\leq 2\sqrt{\frac{H^2\log(2|\Pi|/\delta)}{2K}}.$$

$\square$

**Proposition 4.** The expected regret of FTL can be bounded as

$$\mathbb{E}[\text{SR}(\text{FTL}, K)] \leq \sqrt{\frac{H^2 \log |\Pi|}{K}}.$$

In the tabular case this gives the stated dependence $|\Pi| \leq A^{H|\mathcal{X}||\Xi|}$.

*Proof.* It holds that

$$
\begin{aligned}
\mathbb{E}[\text{SR}(\text{FTL}, K)] &= V_1^{\pi^\star}(s_1) - \mathbb{E}[V_1^{\hat{\pi}^k}(s_1)] = \max_\pi \mathbb{E}[\overline{\mathbb{E}}[V_1^\pi(s_1, \boldsymbol{\xi})]] - \mathbb{E}[V_1^{\hat{\pi}^k}(s_1)] \\
&\leq \mathbb{E}[\max_\pi \overline{\mathbb{E}}[V_1^\pi(s_1, \boldsymbol{\xi})]] - \mathbb{E}[V_1^{\hat{\pi}^k}(s_1)] \\
&= \mathbb{E}[\widetilde{V}_1^{\hat{\pi}^k}(s_1) - V_1^{\hat{\pi}^k}(s_1)] \\
&\leq \mathbb{E}[\max_\pi \widetilde{V}_1^\pi(s_1) - V_1^\pi(s_1)] \\
&\leq \sqrt{\frac{H^2 \log |\Pi|}{K}},
\end{aligned}
$$

where the last inequality is due to Lemma 1. $\qquad\square$

### F.3    PROOF OF THEOREM 1

**Lemma 2** (Data processing inequality, TV distance). Let $\mu, \nu$ be two probability measures on a discrete set $X$ and $f : X \to Y$ be a mapping. Let $f_{\#,\mu}$ and $f_{\#,\nu}$ be the resulting push-forward measures on the space $Y$. Then

$$\|f_{\#,\mu} - f_{\#,\nu}\|_1 \leq \|\mu - \nu\|_1.$$

*Proof.*

$$
\begin{aligned}
\|f_{\#,\mu} - f_{\#,\nu}\|_1 &= \sum_{y \in Y} |f_{\#,\mu}(y) - f_{\#,\nu}(y)| = \sum_{y \in Y} |\mu(f^{-1}(y)) - \nu(f^{-1}(y))| \\
&= \sum_{y \in Y} \Big| \sum_{x \in f^{-1}(y)} \mu(x) - \sum_{x \in f^{-1}(y)} \nu(x) \Big| \\
&\leq \sum_{y \in Y} \sum_{x \in f^{-1}(y)} |\mu(x) - \nu(x)| \leq \sum_{x \in X} |\mu(x) - \nu(x)| = \|\mu - \nu\|_1,
\end{aligned}
$$

where the second inequality is due to the triangle inequality. $\qquad\square$

#### F.3.1    PROOF USING EXPECTED SIMULATION LEMMA

**Lemma 3** (Simulation lemma, expected version). Let $\mathcal{M} = (P, r)$ and $\mathcal{M} = (P', r)$. Define

$$\epsilon_h(s, a) := \|P_h(s, a) - P'_h(s, a)\|_1 \leq \sqrt{\frac{2S \log}{C_h(s, a)}}.$$

For any fixed policy $\pi$ and $s_1 \sim \rho$,

$$|V^{\pi, \mathcal{M}} - V^{\pi, \mathcal{M}'}| \leq \mathbb{E}\left[\sum_{h=1}^{H-1} (H - h)\epsilon_h(s_h, a_h)|\pi, P, \rho\right].$$

It also holds that for any $s_1$

$$|V^{\pi, \mathcal{M}}(s_1) - V^{\pi, \mathcal{M}'}(s_1)| \leq \mathbb{E}\left[\sum_{h=1}^{H-1} (H - h)\epsilon_h(s_h, a_h)|\pi, P, s_1\right].$$

*Proof.* For two different MDPs, their values are defined for the same initial distribution $\rho(s_1)$

$$
\begin{aligned}
|V^{\pi,\mathcal{M}} - V^{\pi,\mathcal{M}'}| &= |\mathbb{E}[V_1^{\pi,\mathcal{M}}(s_1)] - \mathbb{E}[V_1^{\pi,\mathcal{M}'}(s_1)]| \\
&= |\mathbb{E}[r_1(s_1,\pi_1(s_1)) + [P_1 V_2^{\pi,\mathcal{M}}](s_1,\pi_1(s_1)) - r_1(s_1,\pi_1(s_1)) - [P_1' V_2^{\pi,\mathcal{M}'}](s_1,\pi_1(s_1))]| \\
&= |\rho[P_1(V_2^{\pi,\mathcal{M}} - V_2^{\pi,\mathcal{M}'})](s_1,\pi_1(s_1)) + \rho[(P_1 - P_1') V_2^{\pi,\mathcal{M}'}](s_1,\pi_1(s_1))| \\
&\leq |\mathbb{E}[V_2^{\pi,\mathcal{M}}(s_2) - V_2^{\pi,\mathcal{M}'}(s_2)|s_2 \sim \rho P_1^\pi]| + (H-1)\cdot \mathbb{E}[\epsilon_1(s_1,a_1)|s_1 \sim \rho, a_1 = \pi_1(s_1)] \\
&= |V_2^{\pi,\mathcal{M}} - V_2^{\pi,\mathcal{M}'}| + (H-1)\cdot \mathbb{E}[\epsilon_1(s_1,a_1)|s_1 \sim \rho, a_1 = \pi_1(s_1)] \\
&\leq |V_3^{\pi,\mathcal{M}} - V_3^{\pi,\mathcal{M}'}|] + \mathbb{E}[(H-1)\epsilon_1(s_1,a_1) + (H-2)\epsilon_1(s_2,a_2)|\pi,P,\rho] \\
&\cdots \\
&\leq \mathbb{E}\left[\sum_{h=1}^{H-1}(H-h)\epsilon_h(s_h,a_h)|\pi,P,\rho\right].
\end{aligned}
$$

$\square$

Note that the expectation is taken w.r.t.

$$
s_1 \sim \rho_1, \cdots, a_h = \pi_h(s_h), s_{h+1} \sim P_h(s_h,a_h), \cdots.
$$

The policy $\pi$ and transitions $P, P'$ are considered fixed, which implies that $\epsilon_h(s,a)$ is NOT random for fixed $(s,a)$.

For $k \in [K]$, $h \in [H]$, define the filtration as

$$
\mathcal{F}_h^k := \sigma((s_h^m, a_h^m)_{m\in[k-1],h\in[H]}, (s_{h'}^k, a_{h'}^k)_{h'\in[h-1]}).
$$

The policy $\hat{\pi}^k$ is measurable w.r.t. $\mathcal{F}_0^n$, hence

$$
\hat{\pi}^k \perp \boldsymbol{\xi}^k | \mathcal{F}_k,
$$

but

$$
\hat{\pi}^k \not\perp (s_h^k, a_h^k)_{h\in[H]} | \mathcal{F}_k.
$$

Observe that

$$
\begin{aligned}
V_1^\star(s_1) - V_1^{\hat{\pi}^k}(s_1) &= V_1^\star(s_1) - \widehat{V}_1^{k,\pi^\star}(s_1) + \widehat{V}_1^{k,\pi^\star}(s_1) - \widehat{V}_1^k(s_1) + \widehat{V}_1^k(s_1) - V_1^{\hat{\pi}^k}(s_1) \\
&\leq \left|V_1^{\pi^\star}(s_1) - \widehat{V}_1^{k,\pi^\star}(s_1)\right| + \left|V_1^{\hat{\pi}^k}(s_1) - \widehat{V}_1^{k,\hat{\pi}^k}(s_1)\right|.
\end{aligned}
$$

Define

$$
\epsilon_h^k(\xi_{h-1}) := \left\|P_h(\xi_h \in \cdot|\xi_{h-1}) - \widehat{P}_h^k(\xi_h \in \cdot|\xi_{h-1})\right\|_1
$$

$$
C_h^k(\xi) := \sum_{m=1}^{k-1} \mathbb{I}\left\{\xi_h^k = \xi\right\},
$$

where $C_h^k(\xi)$ is defined by $\mathcal{F}_0^k$.

**Key observation** Since $s_{h+1} = (f(x_h, a_h, \xi_h), \xi_h)$ is a mapping of $\xi_h$ given $x_h$ and $a_h$, **for any (deterministic) policy/action sequence** and any $s_h$, it follows from Lemma 2

$$
\epsilon_h^k(s_h,a_h) := \left\|P_h(s_{h+1} \in \cdot|s_h,a_h) - \widehat{P}_h^k(s_{h+1} \in \cdot|s_h,a_h)\right\|_1 \leq \epsilon_h^k(\xi_{h-1}) \leq \mathcal{O}\left(\sqrt{\frac{|\Xi|\iota}{C_h^k(\xi_{h-1})}}\right),
$$

which bounds the model estimation error by a *policy/action-independent* error term. This will lead to tighter regret bound than directly bounding the model error

$$
\epsilon_h^k(s_h,a_h) \leq \mathcal{O}\left(\sqrt{\frac{|S|\iota}{C_h^k(s_h,a_h)}}\right).
$$

Furthermore, we will see that the use of Exo-state $\xi_{h-1}$ overcomes the *misalignment* issue since the sequence $\boldsymbol{\xi}^{k-1}$ is always $\mathcal{F}^k$-measurable. Note that $C^k, \hat{P}^k, \hat{\pi}^k$ are all $\mathcal{F}^k$-measurable, then $\epsilon^k(\cdot)$ is also $\mathcal{F}^k$-measurable.

**The failure of using state-action count.** Denote by $(s_h^k, a_h^k)_{h \in [T]}$ and $(\tilde{s}_h^k, \tilde{a}_h^k)_{h \in [T]}$ the sequence generated by $(\hat{\pi}^k, P)$ and $(\pi^\star, P)$ at the $n$-th episode. In particular,

$$\tilde{s}_1^k = s_1^k = x_1^k, \tilde{a}_1^k = \pi_1^\star(s_1^k), \tilde{s}_2^k = (f(\tilde{s}_1^k, \tilde{a}_1^k, \xi_1^k), \xi_1^k), \cdots, \tilde{s}_{h+1}^k = (f(\tilde{s}_h^k, \tilde{a}_h^k, \xi_h^k), \xi_h^k), \cdots$$

Note that $(\tilde{s}_h^k, \tilde{a}_h^k)_{h \in [H]}$ is fixed conditional on $\boldsymbol{\xi}^k$, so its randomness only comes from $\boldsymbol{\xi}^k$. We bound the **random** regret as

$$\sum_{k=1}^K V_1^\star - V_1^{\hat{\pi}^k} \leq \sum_{k=1}^K V_1^\star - \widehat{V}_1^{k,\pi^\star} + \widehat{V}_1^k - V_1^{\hat{\pi}^k} \leq \sum_{k=1}^K \left| V_1^{\pi^\star} - \widehat{V}_1^{k,\pi^\star} \right| + \sum_{k=1}^K \left| V_1^{\hat{\pi}^k} - \widehat{V}_1^{k,\hat{\pi}^k} \right|$$

$$\leq \sum_{h=1}^{H-1}(H-h)\sum_{k=1}^K \mathbb{E}\left[\epsilon_h^k(\tilde{s}_h^k, \tilde{a}_h^k)|\mathcal{F}_k\right] + \sum_{h=1}^{H-1}(H-h)\sum_{k=1}^K \mathbb{E}\left[\epsilon_h^k(s_h^k, a_h^k)|\mathcal{F}_k\right]$$

$$\leq \sum_{h=1}^{H-1}(H-h)\mathbb{E}\left[\sum_{k=1}^K \sqrt{\frac{2S\iota}{C_h^k(\tilde{s}_h^k, \tilde{a}_h^k)}}|\mathcal{F}_k\right] + \sum_{h=1}^{H-1}(H-h)\mathbb{E}\left[\sqrt{\frac{2S\iota}{C_h^k(s_h^k, a_h^k)}}|\mathcal{F}_k\right],$$

where the third inequality is due to Lemma 3 and the last inequality is due to Lemma 2. However, the key is that the visiting count

$$C_h^k(s, a) = \sum_{m=1}^{k-1} \mathbb{I}\{(s_h^m, a_h^m) = (s, a)\}$$

is defined by $\mathcal{F}^k$ generated by $(\hat{\pi}, P)$. Although we can bound the second term via standard proof, we cannot obtain an upper bound on the first term. Specifically,

$$\sum_{k=1}^K \sqrt{\frac{2S\iota}{C_h^k(\tilde{s}_h^k, \tilde{a}_h^k)}} = \sum_{k=1}^K \sum_{s,a} \mathbb{I}\left\{(\tilde{s}_h^k, \tilde{a}_h^k) = (s, a)\right\}\sqrt{\frac{2S\iota}{C_h^k(\tilde{s}_h^k, \tilde{a}_h^k)}}$$

$$= \sum_{s,a} \sum_{k=1}^K \mathbb{I}\left\{(\tilde{s}_h^k, \tilde{a}_h^k) = (s, a)\right\}\sqrt{\frac{2S\iota}{C_h^k(s, a)}}$$

$$\neq \sum_{s,a} \sum_{c=1}^{C_h^k(s,a)} \sqrt{\frac{2S\iota}{c}}.$$

The last inequality is due to the fact that $C_h^k(s, a)$ does not increase by 1 if $(\tilde{s}_h^k, \tilde{a}_h^k) = (s, a)$ since $C_h^k$ counts based on $\mathcal{F}^k$ or $(s_h^k, a_h^k)$.

**The solution: bounding via exogenous state count.** Using Lemma 3 we can get

$$\sum_{k=1}^K V_1^\star - V_1^{\hat{\pi}^k} \leq \sum_{k=1}^K V_1^\star - \widehat{V}_1^{k,\pi^\star} + \widehat{V}_1^k - V_1^{\hat{\pi}^k} \leq \sum_{k=1}^K \left| \widehat{V}_1^{k,\pi^\star} - V_1^\star \right| + \sum_{k=1}^K \left| \widehat{V}_1^{k,\hat{\pi}^k} - V_1^{\hat{\pi}^k} \right|$$

$$\leq \sum_{h=1}^{H-1}(H-h)\sum_{k=1}^K \mathbb{E}\left[\epsilon_h^k(\tilde{s}_h^k, \tilde{a}_h^k)|\mathcal{F}_k\right] + \sum_{h=1}^{H-1}(H-h)\sum_{k=1}^K \mathbb{E}\left[\epsilon_h^k(s_h^k, a_h^k)|\mathcal{F}_k\right]$$

$$\leq 2\sum_{h=1}^{H-1}(H-h)\sum_{k=1}^K \mathbb{E}\left[\sqrt{\frac{2|\Xi|\log(KH/\delta)}{C_h^k(\xi_{h-1}^k)}}|\mathcal{F}_k\right],$$

where the expectation in the second line is taken w.r.t. the $(\tilde{s}_h^k, \tilde{a}_h^k)_{h \in [H]} \sim P^{\pi^\star}$ and $(s_h^k, a_h^k)_{h \in [H]} \sim P^{\hat{\pi}^k}$. Taking expectation on both sides, we can get

$$\mathbb{E}\left[\sum_{k=1}^K V_1^\star - V_1^{\hat{\pi}^k}\right] \leq 2\sum_{h=1}^{H-1}(H-h)\mathbb{E}\left[\sum_{k=1}^K \sqrt{\frac{2|\Xi|\log(KH/\delta)}{C_h^k(\xi_{h-1}^k)}}\right] \leq 4H^2|\Xi|\sqrt{2N\log(KH/\delta)}.$$

### F.3.2 PROOF VIA MDS SIMULATION LEMMA

**Lemma 4** (Simulation lemma, martingale difference). Let $\mathcal{M} = (P, r)$ and $\mathcal{M}' = (P', r)$. Fix an arbitrary policy $\pi$. Define

$$\epsilon_h := \|P_h(s_h, \pi_h(s_h)) - P'_h(s_h, \pi_h(s_h))\|_1 \leq \sqrt{\frac{2S\log}{C_h(s_h, \pi_h(s_h))}}$$

$$e_h := [P_h|V_{h+1}^{\pi,\mathcal{M}} - V_{h+1}^{\pi,\mathcal{M}'}|](s_h, \pi_h(s_h)) - |V_{h+1}^{\pi,\mathcal{M}} - V_{h+1}^{\pi,\mathcal{M}'}|(s_{h+1}),$$

where $e_h$ is a martingale difference sequence w.r.t. the filtration $\mathcal{H}_h := \sigma(s_1, \pi_1(s_1), \cdots, s_{h-1}, \pi_{h-1}(s_{h-1}))$. Then

$$|V^{\pi,\mathcal{M}}(s_1) - V^{\pi,\mathcal{M}'}(s_1)| \leq \sum_{h=1}^{H-1} (e_h + (H-h)\epsilon_h).$$

Lemma 4 bounds a deterministic term by the sum of two random variables.

*Proof.*

$$
\begin{aligned}
|V_1^{\pi,\mathcal{M}}(s_1) - V_1^{\pi,\mathcal{M}'}(s_1)| &= |r_1(s_1, \pi_1(s_1)) + [P_1 V_2^{\pi,\mathcal{M}}](s_1, \pi_1(s_1)) - r_1(s_1, \pi_1(s_1)) - [P'_1 V_2^{\pi,\mathcal{M}'}](s_1, \pi_1(s_1))| \\
&= |[P_1(V_2^{\pi,\mathcal{M}} - V_2^{\pi,\mathcal{M}'})](s_1, \pi_1(s_1)) + [(P_1 - P'_1)V_2^{\pi,\mathcal{M}'}](s_1, \pi_1(s_1))| \\
&\leq [P_1|V_2^{\pi,\mathcal{M}} - V_2^{\pi,\mathcal{M}'}|](s_1, \pi_1(s_1)) + |[(P_1 - P'_1)V_2^{\pi,\mathcal{M}'}](s_1, \pi_1(s_1))| \\
&= |V_2^{\pi,\mathcal{M}} - V_2^{\pi,\mathcal{M}'}|(s_2) + e_1 + |[(P_1 - P'_1)V_2^{\pi,\mathcal{M}'}](s_1, \pi_1(s_1))| \\
&\leq |V_2^{\pi,\mathcal{M}} - V_2^{\pi,\mathcal{M}'}|(s_2) + e_1 + \epsilon_1 \cdot (H-1) \\
&\leq |V_3^{\pi,\mathcal{M}} - V_3^{\pi,\mathcal{M}'}|(s_3) + e_2 + \epsilon_2 \cdot (H-2) + e_1 + \epsilon_1 \cdot (H-1) \\
&\leq \cdots \\
&\leq |V_h^{\pi,\mathcal{M}} - V_h^{\pi,\mathcal{M}'}|(s_h) + \sum_{h=1}^{H-1} (e_h + (H-h)\epsilon_h) \\
&= \sum_{h=1}^{H-1} (e_h + (H-h)\epsilon_h).
\end{aligned}
$$

$\square$

Define

$$\epsilon_h^k(s_h, a_h) := \left\|P_h(s_h, a_h) - \hat{P}_h^k(s_h, a_h)\right\|_1 \leq \sqrt{\frac{2S\iota}{C_h^k(s_h, a_h)}}$$

$$e_h^k(s_h, a_h|\pi) := [P_h|V_{h+1}^{\pi} - \hat{V}_{h+1}^{k,\pi}|](s_h, a_h) - |V_{h+1}^{\pi} - \hat{V}_{h+1}^{k,\pi}|(s_{h+1}),$$

where $e_h^k$ is a martingale difference sequence that depends on $\pi$ through $\hat{V}_{h+1}^{k,\pi}$. Recall that $(s_h^k, a_h^k)_{h \in [H]}$ and $(\tilde{s}_h^k, \tilde{a}_h^k)_{h \in [H]}$ are the sequence generated by $(\hat{\pi}^k, P)$ and $(\pi^\star, P)$ at the $k$−th episode, which satisfy $\tilde{s}_h^k = s_1^k = x_1^k$. Using Lemma 4 we can get

$$
\begin{aligned}
\sum_{k=1}^{K} V_1^\star - V_1^{\hat{\pi}^k} &\leq \sum_{k=1}^{K} \left|V_1^{\pi^\star}(s_1) - \hat{V}_1^{n,\pi^\star}(s_1)\right| + \left|V_1^{\hat{\pi}}(s_1) - \hat{V}_1^{n,\hat{\pi}}(s_1)\right| \\
&\leq \sum_{h=1}^{H-1} \sum_{k=1}^{K} e_h^k(\tilde{s}_h^k, \tilde{a}_h^k|\pi^\star) + (H-h)\epsilon_h^k(\tilde{s}_h^k, \tilde{a}_h^k) + \sum_{h=1}^{H-1} \sum_{k=1}^{K} e_h^k(s_h^k, a_h^k|\hat{\pi}^k) + (H-h)\epsilon_h^k(s_h^k, a_h^k)
\end{aligned}
$$

The key observation is to verify MDS by considering the essential filtration $\sigma((\boldsymbol{\xi}^k)_n)$ instead of the full (standard) filtration $\sigma((s_h^k, a_h^k)_{k,h})$. Formally, we define the **exogenous filtration** $(s_1^k = x_1^k)$

$$\mathcal{G}_h^k := \sigma((s_1^m, \boldsymbol{\xi}^m)_{m \in [k-1]}, s_1^k, (\xi_{h'}^k)_{h' \in [h-1]}),$$

which is only generated by the exogenous process. This is different from the full filtration

$$\mathcal{F}_h^k := \sigma((s_h^m, a_h^m)_{m \in [k-1], h \in [H]}, (s_{h'}^k, a_{h'}^k)_{h' \in [h-1]}, s_h^k).$$

For any $k$ and $h$, we can recover/simulate $(\tilde{s}_{h'}^k, \tilde{a}_{h'}^k)_{h' \leq t}$ from $s_1^k$, $\pi^\star$ and $\xi_{h-1}^k$ as follows

$$\tilde{s}_{h'}^k = (\tilde{x}_{h'}^k, \xi_{h'-1}^k), \tilde{a}_{h'}^k = \pi_{h'}^\star(\tilde{s}_{h'}^k), \tilde{x}_{h'+1}^k = f(\tilde{x}_{h'}^k, \tilde{a}_{h'}^k, \xi_{h'}^k),$$

which implies that $(\tilde{s}_{h'}^k, \tilde{a}_{h'}^k)_{h' \leq t}$ is measurable w.r.t. $\mathcal{G}_h^k$. Furthermore, $\hat{P}_\Xi^k$ is measurable w.r.t. $\mathcal{G}^k$ implies $\hat{V}^{k,\pi^\star}$ is measurable w.r.t. $\mathcal{G}^k$. Then

$$e_h^k(\tilde{s}_h^k, \tilde{a}_h^k | \pi^\star) = [P_h | V_{h+1}^{\pi^\star} - \hat{V}_{h+1}^{k,\pi^\star}|](\tilde{s}_h^k, \tilde{a}_h^k) - |V_{h+1}^{\pi^\star} - \hat{V}_{h+1}^{k,\pi^\star}|(\tilde{s}_{h+1}^k)$$

is an MDS w.r.t. $\mathcal{G}_h^k$ since

$$\begin{aligned}
\mathbb{E}\left[e_h^k(\tilde{s}_h^k, \tilde{a}_h^k | \pi^\star) | \mathcal{G}_h^k\right] &= \mathbb{E}\left[[P_h | V_{h+1}^{\pi^\star} - \hat{V}_{h+1}^{k,\pi^\star}|](\tilde{s}_h^k, \tilde{a}_h^k) - |V_{h+1}^{\pi^\star} - \hat{V}_{h+1}^{k,\pi^\star}|(\tilde{s}_{h+1}^k) | \mathcal{G}_h^k\right] \\
&= [P_h | V_{h+1}^{\pi^\star} - \hat{V}_{h+1}^{k,\pi^\star}|](\tilde{s}_h^k, \tilde{a}_h^k) - [P_h | V_{h+1}^{\pi^\star} - \hat{V}_{h+1}^{k,\pi^\star}|](\tilde{s}_h^k, \tilde{a}_h^k) \\
&= 0,
\end{aligned}$$

where the second equality is due to that the only non-measurable variable is $\tilde{s}_{h+1}^k$, and $e_h^k(\tilde{s}_h^k, \tilde{a}_h^k | \pi^\star) \in \mathcal{G}_{h+1}^k$ since $\tilde{s}_{h+1}^k$ is measurable w.r.t. $\mathcal{G}_{h+1}^k$.

Since $\hat{\pi}^k$ is measurable w.r.t. $\mathcal{G}^k$, $\hat{V}^{k,\hat{\pi}^k}$ and $(s_{h'}^k, a_{h'}^k)_{h' \leq t}$ are measurable w.r.t. $\mathcal{G}_h^k$. Then

$$\begin{aligned}
\mathbb{E}\left[e_h^k(s_h^k, a_h^k | \hat{\pi}^k) | \mathcal{G}_h^k\right] &= \mathbb{E}\left[[P_h | V_{h+1}^{\hat{\pi}^k} - \hat{V}_{h+1}^{k,\hat{\pi}^k}|](s_h^k, a_h^k) - |V_{h+1}^{\hat{\pi}^k} - \hat{V}_{h+1}^{k,\hat{\pi}^k}|(s_{h+1}^k) | \mathcal{G}_h^k\right] \\
&= [P_h | V_{h+1}^{\hat{\pi}^k} - \hat{V}_{h+1}^{k,\hat{\pi}^k}|](s_h^k, a_h^k) - [P_h | V_{h+1}^{\hat{\pi}^k} - \hat{V}_{h+1}^{k,\hat{\pi}^k}|](s_h^k, a_h^k) \\
&= 0,
\end{aligned}$$

and $e_h^k(s_h^k, a_h^k | \hat{\pi}^k)$ is measurable w.r.t. $\mathcal{G}_{h+1}^k$. Thus $e_h^k(s_h^k, a_h^k | \hat{\pi}^k)$ is also an MDS w.r.t $\mathcal{G}_h^k$. Using the Azuma-Hoeffding inequality, we obtain w.p. $1 - \delta'$

$$\sum_{h=1}^{H-1} \sum_{k=1}^{K} e_h^k(\tilde{s}_h^k, \tilde{a}_h^k | \pi^\star) + e_h^k(s_h^k, a_h^k | \hat{\pi}^k) \leq \mathcal{O}(H\sqrt{KH \log 1/\delta'}).$$

We can bound the error terms as

$$\begin{aligned}
\sum_{h=1}^{H-1} (H - h) \sum_{k=1}^{K} \epsilon_h^k(\tilde{s}_h^k, \tilde{a}_h^k) + \epsilon_h^k(s_h^k, a_h^k) &\leq 2 \sum_{h=1}^{H-1} (H - h) \sum_{k=1}^{K} \sqrt{\frac{2|\Xi| \log(KH/\delta)}{C_h^k(\xi_h^k)}} \\
&\leq 4H^2 |\Xi| \sqrt{2K \log(KH/\delta)}.
\end{aligned}$$

*Remark* 4. We cannot obtain a bound on the expected regret that is independent of $\delta$ as the full information MAB setting since

$$V^{*,\mathcal{M}} = \max_\pi V^{\pi,\mathcal{M}} \neq \max_\pi \mathbb{E}[V^{\pi,\widehat{\mathcal{M}}}] \leq \mathbb{E}[\max_\pi V^{\pi,\widehat{\mathcal{M}}}] = \mathbb{E}[V^{\hat{\pi},\widehat{\mathcal{M}}}].$$

*Remark* 5. We may obtain a tighter regret bound of $\mathcal{O}(H\sqrt{|\Xi|KH\iota})$ by a finer analysis.

*Remark* 6. The simulation lemma MDS leads to a high prob. regret bound, while the simulation lemma expected version leads to a expected regret bound. They are the same order, but the latter one is weaker.

### F.4 PROOFS OF IMPOSSIBILITY RESULTS

**Definition 4** (Pure-Exploitation Greedy (PEG) after a finite warm-start). Fix an integer $L \geq 1$ (not growing with $K$). *Warm-start:* pull each arm exactly $L$ times (in any order). *Greedy phase:* for all subsequent rounds $K > AL$, play

$$a_k \in \arg \max_{a \in [K]} \hat{\mu}_a(k),$$

where $\hat{\mu}_a(k)$ is the empirical mean of arm $a$ over the learner's own past pulls of $a$. Ties are broken by any deterministic rule that is independent of future rewards.

**Lemma 5** (Monotonicity barrier). Consider PEG. Suppose at the start of the greedy phase there exist arms $i, j$ with $\widehat{\mu}_i(KL) = 0$ and $\widehat{\mu}_j(KL) > 0$. Then PEG never pulls arm $i$ again.

*Proof.* At any time $t \geq KL$, the empirical mean of arm $i$ remains exactly $0$ unless $i$ is pulled; conversely, any arm with at least one observed success retains an empirical mean $> 0$ forever, because the count of successes for that arm can never drop to zero. Since PEG selects an arm with maximal empirical mean and $\widehat{\mu}_j(k) \geq \widehat{\mu}_j(KL) > 0 > \widehat{\mu}_i(k)$ for all $t \geq KL$, arm $i$ is never selected. $\qquad\square$

**Theorem 5** (Linear regret for $K$-armed PEG with $L = 1$). Fix any $K \geq 2$ and any gap $\Delta \in (0, \frac{1}{4}]$. Consider Bernoulli arms with means

$$\mu_1 = \tfrac{1}{2} + \Delta, \qquad \mu_2 = \cdots = \mu_K = \tfrac{1}{2}.$$

Run PEG with warm-start $L = 1$ (each arm pulled once) and then act greedily. For all $T \geq K$,

$$\mathbb{E}[\text{Regret}(T)] \ \geq \ \left(\tfrac{1}{2} - \Delta\right)\left(1 - 2^{-(K-1)}\right)\Delta\,(T - K) \ = \ \Omega(T).$$

*Proof.* Let $X_{a,1} \in \{0, 1\}$ be the first Bernoulli sample from arm $a$. Consider the warm-start event

$$E := \{X_{1,1} = 0\} \ \cap \ \Big\{\exists b \in \{2, \ldots, K\}\colon \ X_{b,1} = 1\Big\}.$$

Independence gives

$$\mathbb{P}(E) = (1 - \mu_1)\Big(1 - \prod_{b=2}^{K}(1 - \mu_b)\Big) = \left(\tfrac{1}{2} - \Delta\right)\left(1 - (\tfrac{1}{2})^{K-1}\right).$$

On $E$, after the $K$-round warm-start we have $\widehat{\mu}_1(K) = 0$ and (at least) one suboptimal arm $b$ with $\widehat{\mu}_b(K) = 1$. By Lemma 5, PEG never pulls arm $1$ again. Hence from round $K+1$ onward PEG plays a suboptimal arm every round, incurring per-round regret $\mu_1 - \max_{a \neq 1} \mu_a = \Delta$. Therefore,

$$\text{Regret}(k) \ \geq \ \Delta\,(T - K) \quad \text{on } E,$$

and taking expectations yields the stated lower bound. $\qquad\square$

**Theorem 6** (Linear regret for any fixed warm-start $L$). Fix $K \geq 2$, any integer $L \geq 1$ that does not grow with $T$, and any $\Delta \in (0, \frac{1}{4}]$. Consider the same Bernoulli instance as in Theorem 5. If PEG is run with warm-start size $L$ and then acts greedily, then for all $T \geq KL$,

$$\mathbb{E}[\text{Regret}(k)] \ \geq \ \underbrace{\left(\tfrac{1}{2} - \Delta\right)^L\left(1 - \left(1 - 2^{-L}\right)^{K-1}\right)}_{\text{a positive constant independent of } T} \cdot \Delta\,(T - KL) \ = \ \Omega(k).$$

*Proof.* Let $S_{a,L}$ be the number of successes observed from arm $a$ during the $L$ warm-start pulls of that arm. Consider

$$E_L := \{S_{1,L} = 0\} \ \cap \ \Big\{\exists b \in \{2, \ldots, K\}\colon \ S_{b,L} = L\Big\}.$$

By independence across arms during the warm-start,

$$\mathbb{P}(S_{1,L} = 0) = (1 - \mu_1)^L = \left(\tfrac{1}{2} - \Delta\right)^L, \qquad \mathbb{P}(S_{b,L} = L) = \mu_b^L = (\tfrac{1}{2})^L,$$

and therefore

$$\mathbb{P}(E_L) = \left(\tfrac{1}{2} - \Delta\right)^L\left(1 - \left(1 - 2^{-L}\right)^{K-1}\right).$$

On $E_L$, after the $KL$-round warm-start we have $\widehat{\mu}_1(KL) = 0$ and at least one suboptimal arm $b$ with $\widehat{\mu}_b(KL) = 1$. By Lemma 5, PEG never returns to arm $1$; consequently it plays a suboptimal arm in every round $t > KL$, suffering per-round regret $\Delta$. Taking expectations yields the claimed bound. $\qquad\square$

**Corollary 5** (Any finite exploration budget). Let an algorithm perform any deterministic, data-independent exploration schedule of finite length $N < \infty$ (not growing with $T$), after which it always selects an arm with maximal current empirical mean (deterministic tie-breaking independent of future rewards). Then there exists a Bernoulli $K$-armed instance on which the algorithm has $\mathbb{E}[\text{Regret}(k)] = \Omega(k)$.

*Proof.* Map the schedule to some $L_a \geq 1$ pulls per arm $a$ during the exploration phase, with $\sum_a L_a = N$. Choose means as in Theorem 5 and define the event that the optimal arm produces only zeros in its $L_1$ pulls while at least one suboptimal arm produces only ones in its $L_b$ pulls. This event has strictly positive probability $\prod$-factor bounded away from 0 (independent of $T$). Conditioned on this event, the post-exploration empirical means create a strict separation (optimal arm at 0, a suboptimal arm at 1), and Lemma 5 applies verbatim to force perpetual suboptimal play thereafter, yielding linear regret in $T$. $\square$

*Remark* 7 (Beyond Bernoulli, bounded rewards). The same conclusion holds for any rewards supported on $[0, 1]$ when there exists a gap $\Delta = \mu^\star - \max_{a \neq a^\star} > 0$. By Hoeffding's inequality, for any fixed $L$ there are constants $p_1, p_2 > 0$ (depending on $L$ and the arm means) such that with probability at least $p_1$ the optimal arm's warm-start average is $\leq \mu^\star - \frac{\Delta}{2}$ and with probability at least $p_2$ some suboptimal arm's warm-start average is $\geq \mu^\star - \frac{\Delta}{4}$. The intersection has constant probability $p_1 p_2 > 0$, producing a strict empirical mean misranking after the warm-start and thus linear regret by Lemma 5.

# G   PROOFS OF REGRET BOUNDS IN SECTION 5

## G.1   PROOF OF THEOREM 2

Define $\delta_h^k(\pi) := (V_h^{k,\pi} - V_h^\pi)(s_h^k)$. We have

$$
\begin{aligned}
\delta_h^k(\pi) &= (V_h^{k,\pi} - V_h^\pi)(s_h^k) = (V_h^{k,\pi} - V_h^\pi)(x_h^k, \xi_{h-1}^k) \\
&= r(x_h^k, \pi, \xi_{h-1}^k) + V_h^{k,\pi,a}(f^a(x_h^k, \pi), \xi_{h-1}^k) - r(x_h^k, \pi, \xi_{h-1}^k) - V_h^{\pi,a}(f^a(x_h^k, \pi), \xi_{h-1}^k) \\
&= \phi(f^a(x_h^k, \pi))^\top (w_h^{k,\pi}(\xi_{h-1}^k) - w_h^\pi(\xi_{h-1}^k)) \\
&=: \phi(x_h^{k,\pi})^\top (w_h^{k,\pi}(\xi_{h-1}^k) - w_h^\pi(\xi_{h-1}^k)) \\
&= \phi(x_h^{k,\pi})^\top \Sigma_h^{-1} \Phi_h(\mathbf{v}_h^{k,\pi}(\xi_{h-1}^k) - \mathbf{v}_h^\pi(\xi_{h-1}^k)),
\end{aligned}
$$

where

$$
\begin{aligned}
\mathbf{v}_h^{k,\pi}(\xi_{h-1}^k, n) &= \sum_{\xi_h^k} \hat{P}_h^k(\xi_h^k | \xi_{h-1}^k) \left[ r(g(x_h^a(n), \xi_h^k), \pi, \xi_h^k) + \phi(f^a(g(x_h^a(n), \xi_h^k), \pi))^\top w_{h+1}^{k,\pi}(\xi_h^k) \right] \\
&=: \sum_{\xi_h^k} \hat{P}_h^k(\xi_h^k | \xi_{h-1}^k) \left[ r(x_{h+1}^k(n), \pi, \xi_h^k) + \phi(f^a(x_{h+1}^k(n), \pi))^\top w_{h+1}^{k,\pi}(\xi_h^k) \right] \\
\mathbf{v}_h^\pi(\xi_{h-1}^k, n) &= \sum_{\xi_h^k} P_h(\xi_h^k | \xi_{h-1}^k) \left[ r(g(x_h^a(n), \xi_h^k), \pi, \xi_h^k) + \phi(f^a(g(x_h^a(n), \xi_h^k), \pi))^\top w_{h+1}^\pi(\xi_h^k) \right] \\
&=: \sum_{\xi_h^k} P_h(\xi_h^k | \xi_{h-1}^k) \left[ r(x_{h+1}^k(n), \pi, \xi_h^k) + \phi(f^a(x_{h+1}^k(n), \pi))^\top w_{h+1}^\pi(\xi_h^k) \right].
\end{aligned}
$$

Note that we denote $x_h^{k,\pi} := f^a(x_h^k, \pi(x_h^k, \xi_{h-1}^k))$ which implicitly depends on $\xi_{h-1}^k$ and $x_{h+1}^k(n) := g(x_h^a(n), \xi_h^k))$ which implicitly depends on $\xi_h^k$. We have

$$
\begin{aligned}
\delta_h^k(\pi) &= \phi(x_h^{k,\pi})^\top \Sigma_h^{-1} \Phi_h^\top (\mathbf{v}_h^{k,\pi}(\xi_{h-1}^k) - \mathbf{v}_h^\pi(\xi_{h-1}^k)) \\
&= \phi(x_h^{k,\pi})^\top \Sigma_h^{-1} \sum_k \phi(x_h^a(n)) \left[ \sum_{\xi_h^k} (\hat{P}_h^k(\xi_h^k | \xi_{h-1}^k) - P_h(\xi_h^k | \xi_{h-1}^k)) r(x_{h+1}^k(n), \pi, \xi_h^k) \right] \\
&\quad + \phi(x_h^{k,\pi})^\top \Sigma_h^{-1} \sum_k \phi(x_h^a(n)) \cdot \\
&\qquad \left[ \sum_{\xi_h^k} \hat{P}_h^k(\xi_h^k | \xi_{h-1}^k) \phi(f^a(x_{h+1}^k(n), \pi))^\top w_{h+1}^{k,\pi}(\xi_h^k) - P_h(\xi_h^k | \xi_{h-1}^k) \phi(f^a(x_{h+1}^k(n), \pi))^\top w_{h+1}^\pi(\xi_h^k) \right].
\end{aligned}
$$

Under Assumption 2, we have

$$
\begin{aligned}
w_h^{k,\pi}(\xi_{h-1}^k) &= \Sigma_h^{-1}\Phi_h \sum_{\xi_h^k} \hat{P}_h^k(\xi_h^k|\xi_{h-1}^k) \left[ r(g(x_h^a(\cdot),\xi_h^k),\pi,\xi_h^k) + \phi(f^a(g(x_h^a(\cdot),\xi_h^k),\pi))^\top w_{h+1}^{k,\pi}(\xi_h^k) \right] \\
&= \Sigma_h^{-1}\Phi_h[\hat{P}_h^k\mathbf{r}](\xi_{h-1}^k) + \Sigma_h^{-1}\sum_k \phi_h(k)\sum_{\xi_h^k}\hat{P}_h^k(\xi_h^k|\xi_{h-1}^k)(M_h^\pi(\xi_h^k)\phi_h(k))^\top w_{h+1}^{k,\pi}(\xi_h^k) \\
&= \Sigma_h^{-1}\Phi_h[\hat{P}_h^k\mathbf{r}](\xi_{h-1}^k) + \Sigma_h^{-1}\sum_k \phi_h(k)\phi_h(k)^\top\sum_{\xi_h^k}\hat{P}_h^k(\xi_h^k|\xi_{h-1}^k)(M_h^\pi(\xi_h^k))^\top w_{h+1}^{k,\pi}(\xi_h^k) \\
&= \Sigma_h^{-1}\Phi_h[\hat{P}_h^k\mathbf{r}](\xi_{h-1}^k) + \sum_{\xi_h^k}\hat{P}_h^k(\xi_h^k|\xi_{h-1}^k)(M_h^\pi(\xi_h^k))^\top w_{h+1}^{k,\pi}(\xi_h^k) \\
&= \Sigma_h^{-1}\Phi_h[\hat{P}_h^k\mathbf{r}](\xi_{h-1}^k) + [\hat{P}_h^k((M_h^\pi)^\top w_{h+1}^{k,\pi})](\xi_{h-1}^k).
\end{aligned}
$$

Similarly, we can get

$$
w_h^\pi(\xi_{h-1}^k) = \Sigma_h^{-1}\Phi_h[P_h\mathbf{r}](\xi_{h-1}^k) + [P_h((M_h^\pi)^\top w_{h+1}^\pi)](\xi_{h-1}^k).
$$

Thus

$$
\begin{aligned}
w_h^{k,\pi} - w_h^\pi &= \Sigma_h^{-1}\Phi_h[(\hat{P}_h^k - P_h)\mathbf{r}](\xi_{h-1}^k) + [\hat{P}_h^k((M_h^\pi)^\top w_{h+1}^{k,\pi})](\xi_{h-1}^k) - [P_h((M_h^\pi)^\top w_{h+1}^\pi)](\xi_{h-1}^k) \\
&= \Sigma_h^{-1}\Phi_h[(\hat{P}_h^k - P_h)\mathbf{r}](\xi_{h-1}^k) + [(\hat{P}_h^k - P_h)((M_h^\pi)^\top w_{h+1}^{k,\pi})](\xi_{h-1}^k) + [P_h((M_h^\pi)^\top(w_{h+1}^{k,\pi} - w_{h+1}^\pi))](\xi_{h-1}^k) \\
&= \Sigma_h^{-1}\Phi_h[(\hat{P}_h^k - P_h)\mathbf{r}](\xi_{h-1}^k) + [(\hat{P}_h^k - P_h)((M_h^\pi)^\top w_{h+1}^{k,\pi})](\xi_{h-1}^k) \\
&\quad + [P_h((M_h^\pi)^\top(w_{h+1}^{k,\pi} - w_{h+1}^\pi))](\xi_{h-1}^k) - (M_h^\pi(\xi_h^k))^\top(w_{h+1}^{k,\pi} - w_{h+1}^\pi)(\xi_h^k) + (M_h^\pi(\xi_h^k))^\top(w_{h+1}^{k,\pi} - w_{h+1}^\pi)(\xi_h^k) \\
&=: \epsilon_h^k(\pi) + e_h^k(\pi) + (M_h^\pi(\xi_h^k))^\top(w_{h+1}^{k,\pi} - w_{h+1}^\pi)(\xi_h^k),
\end{aligned}
$$

where we define

$$
\begin{aligned}
e_h^k(\pi) &:= [P_h((M_h^\pi)^\top(w_{h+1}^{k,\pi} - w_{h+1}^\pi))](\xi_{h-1}^k) - (M_h^\pi(\xi_h^k))^\top(w_{h+1}^{k,\pi} - w_{h+1}^\pi)(\xi_h^k), \\
\epsilon_h^k(\pi) &:= \Sigma_h^{-1}\Phi_h[(\hat{P}_h^k - P_h)\mathbf{r}](\xi_{h-1}^k) + [(\hat{P}_h^k - P_h)((M_h^\pi)^\top w_{h+1}^{k,\pi})](\xi_{h-1}^k).
\end{aligned}
$$

**Lemma 6.** Let $\{\phi_i\}_{i=1}^K \subset \mathbb{R}^d$, and define

$$
A = \sum_{i=1}^K \phi_i\phi_i^\top \in \mathbb{R}^{d\times d},
$$

which is assumed to be full rank. For $\epsilon_i \in \mathbb{R}$ and $u \in \mathbb{R}^d$, set

$$
\varepsilon = (\epsilon_1,\ldots,\epsilon_K)^\top, \quad \Phi = [\phi_1 \cdots \phi_K] \in \mathbb{R}^{d\times K}.
$$

Then the following bound holds:

$$
\left| u^\top A^{-1}\sum_{i=1}^K \phi_i\epsilon_i \right| \leq \|u\|_{A^{-1}}\|\varepsilon\|_2,
$$

where $\|u\|_{A^{-1}} = \sqrt{u^\top A^{-1}u}$.

*Proof.* Observe that

$$
u^\top A^{-1}\sum_{i=1}^K \phi_i\epsilon_i = u^\top A^{-1}\Phi\varepsilon.
$$

Let $A^{-1/2}$ denote the symmetric square root of $A^{-1}$, and define

$$
B := A^{-1/2}\Phi \in \mathbb{R}^{d\times K}.
$$

Then

$$
u^\top A^{-1}\Phi\varepsilon = (A^{-1/2}u)^\top(A^{-1/2}\Phi)\varepsilon = (A^{-1/2}u)^\top B\varepsilon.
$$

Note that

$$BB^\top = A^{-1/2}\Phi\Phi^\top A^{-1/2} = A^{-1/2}AA^{-1/2} = I_d,$$

hence $\|B\|_2 = 1$. By the Cauchy–Schwarz inequality,

$$\left|(A^{-1/2}u)^\top B\varepsilon\right| \leq \|A^{-1/2}u\|_2 \|B\varepsilon\|_2 \leq \|A^{-1/2}u\|_2 \|\varepsilon\|_2.$$

Finally, $\|A^{-1/2}u\|_2 = \sqrt{u^\top A^{-1}u} = \|u\|_{A^{-1}}$, proving the claim. $\qquad\square$

We obtain the recursion for $d_h^k(\pi) := w_h^{k,\pi} - w_h^\pi$ as

$$\begin{aligned}
d_h^k(\pi) &= \epsilon_h^k(\pi) + e_h^k(\pi) + (M_h^\pi(\xi_h^k))^\top d_{h+1}^k \\
&= \sum_{s=h}^{H}(\prod_{h'=h}^{s-1} M_{h'}^\pi(\xi_{h'}^k))^\top(\epsilon_s^k(\pi) + e_s^k(\pi)) \\
&=: \sum_{s=h}^{H} \tilde{\epsilon}_s^k(\pi) + \tilde{e}_s^k(\pi).
\end{aligned}$$

Note that $e_h^k(\pi)$ is an vector-valued MDS w.r.t. $\mathcal{G}_h^k$ since

$$\mathbb{E}\left[e_h^k(\pi)|\mathcal{G}_h^k\right] = \mathbb{E}\left[[P_h((M_h^\pi)^\top(w_{h+1}^{k,\pi} - w_{h+1}^\pi))](\xi_{h-1}^k) - (M_h^\pi(\xi_h^k))^\top(w_{h+1}^{k,\pi} - w_{h+1}^\pi)(\xi_h^k)|\mathcal{G}_h^k\right] = \mathbf{0}.$$

Since $M_{h'}^\pi(\xi_{h'}^k)$ are $\mathcal{G}_h^k$-measurable for $h' \leq h-1$, we have

$$\mathbb{E}\left[\tilde{e}_h^k(\pi)|\mathcal{G}_h^k\right] = \mathbb{E}\left[(\prod_{h'=1}^{h-1} M_{h'}^\pi(\xi_{h'}^k))^\top e_h^k|\mathcal{G}_h^k\right] = (\prod_{h'=1}^{h-1} M_{h'}^\pi(\xi_{h'}^k))^\top \mathbb{E}\left[e_h^k|\mathcal{G}_h^k\right] = \mathbf{0}.$$

Thus $\tilde{e}_s^k(\pi)$ is also a vector-valued MDS w.r.t. $\mathcal{G}_h^k$.

Note that $\Phi$ is full rank, so $\phi(x^a)$ can be represented as $\phi(x^a) = \Phi\alpha$ for some $\alpha \in \mathbb{R}^K$. Under Assumption 3, we can prove the following lemma.

**Lemma 7.** For any $(n, t, \pi, x^a, \xi)$, it holds that $V_h^{k,\pi^k,a}(x^a, \xi) \geq V_h^{k,\pi,a}(x^a, \xi)$.

We have

$$\begin{aligned}
\mathrm{Regret}(K) &= \sum_{k=1}^{K}\left(V_1^{\pi^\star}(s_1^k) - V_1^{\hat{\pi}^k}(s_1^k)\right) \\
&= \sum_{k=1}^{K}\left(V_1^{\pi^\star} - V_1^{k,\pi^\star}\right)(s_1^k) + \left(V_1^{k,\pi^\star} - V_1^{k,\pi^k}\right)(s_1^k) + \left(V_1^{k,\pi^k} - V_1^{\pi^k}\right)(s_1^k) \\
&\leq \sum_{k=1}^{K}\left(V_1^{\pi^\star} - V_1^{k,\pi^\star}\right)(s_1^k) + \left(V_1^{k,\pi^k} - V_1^{\pi^k}\right)(s_1^k) \\
&= \sum_{k=1}^{K} -\phi(x_1^{k,a})^\top \delta_1^k(\pi^\star) + \phi(x_1^{k,a})^\top \delta_1^k(\pi^k) \\
&= \sum_{k=1}^{K}\sum_{h=1}^{H-1} -\phi(x_1^{k,a})^\top(\tilde{\epsilon}_h^k(\pi^\star) + \tilde{e}_h^k(\pi^\star)) + \phi(x_1^{k,a})^\top(\tilde{\epsilon}_h^k(\pi^k) + \tilde{e}_h^k(\pi^k)).
\end{aligned}$$

Note that for any $\mathcal{G}_h^k$-measurable policy $\pi$, the sequence $\phi(x_1^{k,a})^\top \tilde{e}_h^k(\pi)$ is an MDS w.r.t. $\mathcal{G}_h^k$. Moreover,

$$\left|\phi(x_1^{k,a})^\top \tilde{e}_h^k(\pi)\right| \leq 4\sqrt{d}.$$

Next, we bound

$$
\phi(x_1^{k,a})^\top \tilde{\epsilon}_h^k(\pi^k) = \phi(x_1^{k,a})^\top \left( \prod_{h'=1}^{h-1} M_{h'}^\pi(\xi_{h'}^k) \right)^\top \epsilon_h^k
$$

$$
= \left( \prod_{h'=1}^{h-1} M_{h'}^\pi(\xi_{h'}^k) \phi(x_1^{k,a}) \right)^\top \left( \Sigma_h^{-1} \Phi_h[(\hat{P}_h^k - P_h)\mathbf{r}](\xi_{h-1}^k) + [(\hat{P}_h^k - P_h)((M_h^\pi)^\top w_{h+1}^{k,\pi})](\xi_{h-1}^k) \right)
$$

$$
= \left( \prod_{h'=1}^{h-1} M_{h'}^\pi(\xi_{h'}^k) \phi(x_1^{k,a}) \right)^\top \Sigma_h^{-1} \Phi_h[(\hat{P}_h^k - P_h)\mathbf{r}](\xi_{h-1}^k)
$$

$$
+ \left( \prod_{h'=1}^{h-1} M_{h'}^\pi(\xi_{h'}^k) \phi(x_1^{k,a}) \right)^\top [(\hat{P}_h^k - P_h)((M_h^\pi)^\top w_{h+1}^{k,\pi})](\xi_{h-1}^k)
$$

$$
= \left( \prod_{h'=1}^{h-1} M_{h'}^\pi(\xi_{h'}^k) \phi(x_1^{k,a}) \right)^\top \Sigma_h^{-1} \Phi_h[(\hat{P}_h^k - P_h)\mathbf{r}](\xi_{h-1}^k)
$$

$$
+ \left[ (\hat{P}_h^k - P_h) \left( \prod_{h'=1}^{h} M_{h'}^\pi(\xi_{h'}^k) \phi(x_1^{k,a}) \right)^\top w_{h+1}^{k,\pi} \right] (\xi_{h-1}^k).
$$

We can bound the first term as

$$
\left( \prod_{h'=1}^{h-1} M_{h'}^\pi(\xi_{h'}^k) \phi(x_1^{k,a}) \right)^\top \Sigma_h^{-1} \Phi_h[(\hat{P}_h^k - P_h)\mathbf{r}](\xi_{h-1}^k) = (\tilde{M}_{h-1}^\pi \phi(x_1^{k,a}))^\top \Sigma_h^{-1} \Phi_h[(\hat{P}_h^k - P_h)\mathbf{r}
$$

$$
\leq \left\| \tilde{M}_{h-1}^\pi \phi(x_1^{k,a}) \right\|_{\Sigma_h^{-1}} \left\| (\hat{P}_h^k - P_h)\mathbf{r} \right\|_2
$$

$$
\leq \left\| \tilde{M}_{h-1}^\pi \phi(x_1^{k,a}) \right\|_{\Sigma_h^{-1}} \sqrt{N} \left\| (\hat{P}_h^k - P_h)(\xi_{h-1}^k) \right\|_1
$$

The second term can be bounded as

$$
\left[ (\hat{P}_h^k - P_h) \left( \prod_{h'=1}^{h} M_{h'}^\pi(\xi_{h'}^k) \phi(x_1^{k,a}) \right)^\top w_{h+1}^{k,\pi} \right] (\xi_{h-1}^k)
$$

$$
\leq \left\| (\hat{P}_h^k - P_h)(\xi_{h-1}^k) \right\|_1 \cdot \max_{\xi'} \left| \left( \prod_{h'=1}^{h} M_{h'}^\pi(\xi_{h'}^k) \phi(x_1^{k,a}) \right)^\top w_{h+1}^{k,\pi}(\xi') \right|
$$

$$
\leq \left\| (\hat{P}_h^k - P_h)(\xi_{h-1}^k) \right\|_1 \left\| \prod_{h'=1}^{h} M_{h'}^\pi(\xi_{h'}^k) \phi(x_1^{k,a}) \right\| \left\| w_{h+1}^{k,\pi} \right\|
$$

$$
\leq \left\| (\hat{P}_h^k - P_h)(\xi_{h-1}^k) \right\|_1 \left\| \prod_{h'=1}^{h} M_{h'}^\pi(\xi_{h'}^k) \right\| \left\| \phi(x_1^{k,a}) \right\| \left\| w_{h+1}^{k,\pi} \right\|
$$

$$
\leq \left\| (\hat{P}_h^k - P_h)(\xi_{h-1}^k) \right\|_1 \left\| \prod_{h'=1}^{h} M_{h'}^\pi(\xi_{h'}^k) \right\| \sqrt{d}
$$

$$
\leq \left\| (\hat{P}_h^k - P_h)(\xi_{h-1}^k) \right\|_1 \prod_{h'=1}^{h} \left\| M_{h'}^\pi(\xi_{h'}^k) \right\| \sqrt{d}
$$

$$
\leq \sqrt{d} \left\| (\hat{P}_h^k - P_h)(\xi_{h-1}^k) \right\|_1,
$$

where the last inequality is due to the fact that $\sup_{\pi,\xi,h} \|M_h^\pi(\xi)\| \leq 1$. We can bound the regret as

$$
\begin{aligned}
\text{Regret}(K) &\leq \sum_{k=1}^{K} \sum_{h=1}^{H-1} -\phi(x_1^{k,a})^\top (\tilde{\epsilon}_h^k(\pi^\star) + \tilde{e}_h^k(\pi^\star)) + \phi(x_1^{k,a})^\top (\tilde{\epsilon}_h^k(\pi^k) + \tilde{e}_h^k(\pi^k)) \\
&\leq \mathcal{O}(\sqrt{dKH \log 1/\delta'}) + 2 \sum_{k=1}^{K} \sum_{h=1}^{H-1} \left\| \tilde{M}_{h-1}^\pi \phi(x_1^{k,a}) \right\|_{\Sigma_h^{-1}} \sqrt{N} \left\| (\hat{P}_h^k - P_h)(\xi_{h-1}^k) \right\|_1 \\
&\quad + \sqrt{d} \left\| (\hat{P}_h^k - P_h)(\xi_{h-1}^k) \right\|_1 \\
&\leq \mathcal{O}(\sqrt{dKH \log 1/\delta'}) + 2 \sum_h (\sqrt{N/\lambda_0} + \sqrt{d}) \sum_k \sqrt{\frac{|\Xi|\iota}{C_h^k(\xi_{h-1}^k)}} \\
&\leq \mathcal{O}(\sqrt{dKH \log 1/\delta'}) + 2 \sum_h (\sqrt{N/\lambda_0} + \sqrt{d})|\Xi|\sqrt{K\iota} \\
&\leq \mathcal{O}(\sqrt{N/\lambda_0} + \sqrt{d})|\Xi|H\sqrt{K\iota}.
\end{aligned}
$$

### G.2 PROOF OF THEOREM 3

Recall that

$$
\begin{aligned}
\delta_h^k(\pi) &= \phi(x_h^{k,\pi})^\top (w_h^{k,\pi}(\xi_{h-1}^k) - w_h^\pi(\xi_{h-1}^k)) \\
&= \phi(x_h^{k,\pi})^\top d_h^k(\pi) = \phi(x_h^{k,\pi})^\top (\epsilon_h^k(\pi) + e_h^k(\pi) + (M_h^\pi(\xi_h^k))^\top d_{h+1}^k(\pi)) \\
&= \phi(x_h^{k,\pi})^\top \left[ \Sigma_h^{-1} \Phi_h[(\hat{P}_h^k - P_h)\mathbf{r}](\xi_{h-1}^k) + [(\hat{P}_h^k - P_h)((M_h^\pi)^\top w_{h+1}^{k,\pi})](\xi_{h-1}^k) \right] \\
&\quad + \phi(x_h^{k,\pi})^\top \left[ P_h((M_h^\pi)^\top (w_{h+1}^{k,\pi} - w_{h+1}^\pi)))](\xi_{h-1}^k) - (M_h^\pi(\xi_h^k))^\top (w_{h+1}^{k,\pi} - w_{h+1}^\pi)(\xi_h^k) \right] \\
&\quad + \phi(x_h^{k,\pi})^\top (M_h^\pi(\xi_h^k))^\top (w_{h+1}^{k,\pi} - w_{h+1}^\pi)(\xi_h^k) \\
&= \phi(x_h^{k,\pi})^\top \Sigma_h^{-1} \Phi_h[(\hat{P}_h^k - P_h)\mathbf{r}](\xi_{h-1}^k) + \phi(x_h^{k,\pi})^\top \Sigma_h^{-1} \Phi_h[(\hat{P}_h^k - P_h)((M_h^\pi)^\top w_{h+1}^{k,\pi})](\xi_{h-1}^k) \\
&\quad + [P_h(\phi(x_{h+1}^{k,\pi})^\top (w_{h+1}^{k,\pi} - w_{h+1}^\pi)))](\xi_{h-1}^k) \\
&\quad - \phi(x_{h+1}^{k,\pi})^\top (w_{h+1}^{k,\pi} - w_{h+1}^\pi)(\xi_h^k) + \phi(x_{h+1}^{k,\pi})^\top (w_{h+1}^{k,\pi} - w_{h+1}^\pi)(\xi_h^k) \\
&=: \bar{\epsilon}_h^k(\pi) + \bar{e}_h^k(\pi) + \delta_{h+1}^k(\pi),
\end{aligned}
$$

where we used $M_h^\pi(\xi_h^k)\phi(x_h^{k,\pi}) = \phi(x_{h+1}^{k,\pi})$ under Assumption 4. Note that $\bar{e}_s^k(\pi)$ is an MDS w.r.t. $\mathcal{G}_h^k$ since

$$
\mathbb{E}\left[\bar{e}_h^k(\pi)|\mathcal{G}_h^k\right] = \mathbb{E}\left[[P_h(\phi(x_{h+1}^{k,\pi})^\top (w_{h+1}^{k,\pi} - w_{h+1}^\pi)))](\xi_{h-1}^k) - \phi(x_{h+1}^{k,\pi})^\top (w_{h+1}^{k,\pi} - w_{h+1}^\pi)(\xi_h^k)|\mathcal{G}_h^k\right] = 0.
$$

In addition, the following holds almost surely

$$
\begin{aligned}
\left|\bar{e}_h^k(\pi)\right| &= \left|[P_h(\phi(x_{h+1}^{k,\pi})^\top (w_{h+1}^{k,\pi} - w_{h+1}^\pi)))](\xi_{h-1}^k) - \phi(x_{h+1}^{k,\pi})^\top (w_{h+1}^{k,\pi} - w_{h+1}^\pi)(x_h^{k,\pi}, \xi_h^k)\right| \\
&= \left|[P_h(V_{h+1}^{k,\pi,a} - V_{h+1}^{k,\pi,a})](\xi_{h-1}^k) - (V_{h+1}^{k,\pi,a} - V_{h+1}^{k,\pi,a})(x_{h+1}^{k,\pi}, \xi_h^k)\right| \\
&\leq 2(H - 1 - h).
\end{aligned}
$$

We can bound $\bar{\epsilon}_h^k(\pi)$ as

$$
\begin{aligned}
\bar{\epsilon}_h^k(\pi) &= \phi(x_h^{k,\pi})^\top \Sigma_h^{-1} \Phi_h [(\hat{P}_h^k - P_h)\mathbf{r}](\xi_{h-1}^k) + \phi(x_h^{k,\pi})^\top [(\hat{P}_h^k - P_h)((M_h^\pi)^\top w_{h+1}^{k,\pi})](\xi_{h-1}^k) \\
&= \phi(x_h^{k,\pi})^\top \Sigma_h^{-1} \Phi_h [(\hat{P}_h^k - P_h)\mathbf{r}](\xi_{h-1}^k) + [(\hat{P}_h^k - P_h)\phi(x_{h+1}^{k,\pi})^\top w_{h+1}^{k,\pi}](\xi_{h-1}^k) \\
&= \phi(x_h^{k,\pi})^\top \Sigma_h^{-1} \Phi_h [(\hat{P}_h^k - P_h)\mathbf{r}](\xi_{h-1}^k) + [(\hat{P}_h^k - P_h)V_{h+1}^{k,\pi,a}](x_h^{k,\pi}, \xi_{h-1}^k) \\
&\le \left\| \phi(x_h^{k,\pi}) \right\|_{\Sigma_h^{-1}} \left\| [(\hat{P}_h^k - P_h)\mathbf{r}](\xi_{h-1}^k) \right\| + \left\| (\hat{P}_h^k - P_h)(\xi_{h-1}^k) \right\|_1 (H - h) \\
&\le \left\| \phi(x_h^{k,\pi}) \right\|_{\Sigma_h^{-1}} \sqrt{N} \left\| (\hat{P}_h^k - P_h)(\xi_{h-1}^k) \right\|_1 + \left\| (\hat{P}_h^k - P_h)(\xi_{h-1}^k) \right\|_1 (H - h) \\
&= \left\| (\hat{P}_h^k - P_h)(\xi_{h-1}^k) \right\|_1 \left( \sqrt{N} \left\| \phi(x_h^{k,\pi}) \right\|_{\Sigma_h^{-1}} + H - h \right) \\
&\le \sqrt{\frac{|\Xi|\iota}{C_h^k(\xi_{h-1}^k)}} \left( \sqrt{N} \left\| \phi(x_h^{k,\pi}) \right\|_{\Sigma_h^{-1}} + H - h \right)
\end{aligned}
$$

Unrolling the recursion of $\delta_h^k(\pi)$, we have

$$
\delta_1^k(\pi) = \sum_{s=1}^{H-1} \bar{\epsilon}_s^k(\pi) + \bar{e}_s^k(\pi).
$$

**Lemma 8.** For any $(n, t, \pi, x^a, \xi)$, it holds that $V_h^{k,\pi^k,a}(x^a, \xi) \ge V_h^{k,\pi,a}(x^a, \xi)$.

*Proof.* The proof follows from induction. Observe that holds when $h = H - 1$. For any $(x^a, \xi)$, using the definition of $\pi^k$, we have

$$
\begin{aligned}
V_h^{k,\pi,a}(x^a, \xi) &= \phi(x^a)^\top w_h^{k,\pi}(\xi) \\
&= \phi(x^a)^\top \left( \Sigma_h^{-1} \Phi_h [\hat{P}_h^k \mathbf{r}](\xi) + [\hat{P}_h^k((M_h^\pi)^\top w_{h+1}^{k,\pi})](\xi) \right) \\
&= \phi(x^a)^\top \Sigma_h^{-1} \Phi_h [\hat{P}_h^k \mathbf{r}](\xi) + [\hat{P}_h^k \phi(x_{h+1}^a)^\top w_{h+1}^{k,\pi}](\xi) \\
&= \phi(x^a)^\top \Sigma_h^{-1} \Phi_h [\hat{P}_h^k \mathbf{r}](\xi) + [\hat{P}_h^k V_{h+1}^{k,\pi,a}](x^a, \xi) \\
&\ge \phi(x^a)^\top \Sigma_h^{-1} \Phi_h [\hat{P}_h^k \mathbf{r}](\xi) + [\hat{P}_h^k V_{h+1}^{k,\pi',a}](x^a, \xi) \\
&= V_h^{k,\pi',a}(x^a, \xi).
\end{aligned}
$$

$\square$

Now we bound the regret

$$
\begin{aligned}
\text{Regret}(K) &= \sum_{k=1}^K \left( V_1^{\pi^\star}(s_1^k) - V_1^{\hat{\pi}^k}(s_1^k) \right) \\
&= \sum_{k=1}^K \left( V_1^{\pi^\star} - V_1^{k,\pi^\star} \right)(s_1^k) + \left( V_1^{k,\pi^\star} - V_1^{k,\pi^k} \right)(s_1^k) + \left( V_1^{k,\pi^k} - V_1^{\pi^k} \right)(s_1^k) \\
&\le \sum_{k=1}^K \left( V_1^{\pi^\star} - V_1^{k,\pi^\star} \right)(s_1^k) + \left( V_1^{k,\pi^k} - V_1^{\pi^k} \right)(s_1^k) \\
&= \sum_{k=1}^K -\delta_1^k(\pi^\star) + \delta_1^k(\pi^k) \\
&= \sum_{k=1}^K \sum_{h=1}^{H-1} -(\bar{\epsilon}_s^k(\pi^\star) + \bar{e}_s^k(\pi^\star)) + \bar{\epsilon}_s^k(\pi^k) + \bar{e}_s^k(\pi^k) \\
&\le \mathcal{O}(H\sqrt{KH\log 1/\delta'}) + 2\sum_{h=1}^{H-1} \left( \sqrt{N} \left\| \phi(x_h^{k,\pi}) \right\|_{\Sigma_h^{-1}} + H - h \right) \sum_{k=1}^K \sqrt{\frac{2|\Xi|\log(KH/\delta)}{C_h^k(\xi_h^k)}} \\
&\le \mathcal{O}(H\sqrt{KH\log 1/\delta'}) + 4(H^2 + H\sqrt{N/\lambda_0})|\Xi|\sqrt{2K\log(KH/\delta)}.
\end{aligned}
$$

### G.3 PROOF OF THEOREM 4

We start with two simple geometric and statistical facts.

**Lemma 9** (Anchor LS predictor stability). For any $t, \xi$, any anchor vectors $y, u \in \mathbb{R}^K$, and any $x^a$,

$$\left|\phi(x^a)^\top \Sigma_h^{-1} \Phi_h(y - u)\right| \leq \lambda_0^{-1/2} \|y - u\|_2.$$

*Proof.* By Cauchy-Schwarz, $\left|\phi^\top A^{-1}\Phi(y - u)\right| \leq \|\phi\|_2 \|A^{-1}\Phi\| \|y - u\|_2$. Since $\|\phi\| \leq 1$ and $\|A^{-1}\Phi\| = \sigma_{\min}(\Phi)^{-1} = \lambda_0^{-1/2}$, the claim follows. $\square$

**Lemma 10** (Row-wise empirical transition concentration). Fix $t$ and $\xi$. Let $g : \Xi \to [0, H]$ and suppose $\widehat{P}^n(\cdot \mid \xi)$ is the empirical distribution from $m = n_h^k(\xi) \geq 1$ i.i.d. samples of $\xi'$ drawn from $P(\cdot \mid \xi)$ (across episodes). Then for any $\delta \in (0, 1)$,

$$\Pr\left(\left|(\widehat{P}^n - P)g\right| \leq H\sqrt{\frac{\log(2/\delta)}{2m}}\right) \geq 1 - \delta.$$

*Proof.* $(\widehat{P}^n - P)g = \frac{1}{m}\sum_{i=1}^m Z_i - \mathbb{E}[Z_i]$ where $Z_i := g(\xi_i') \in [0, H]$ with $\xi_i' \sim P(\cdot \mid \xi)$ i.i.d. Apply Hoeffding's inequality. $\square$

The concentration will be lifted to uniform (over $n, t, \xi$) events via a union bound and the standard summation $\sum_{j=1}^M (m_j + 1)^{-1/2} \leq 2\sqrt{M}$.

*Proof.* We follow a the one-step decomposition as in the proof of Theorem 3, carefully adding the misspecification term.

Fix any reference policy $\pi$ (we will take $\pi = \pi^\star$ at the end). Let $s_h^k = (x_h^k, \xi_h^k)$ be the state visited in episode $n$ by the coupling argument used in LSVI analyses (or simply the realized trajectory under the deployed policy at episode $n$). Denote the value error

$$\delta_h^k(\pi) := \left(V_h^{k,\pi} - V_h^\pi\right)(s_h^k),$$

where $V^{k,\pi}$ is the value when Bellman backups use $\widehat{P}^n$ and parameters $w^n$, while $V^\pi$ uses the true model and the ideal parameters $w_\cdot^\pi$ that linearly represent the values of $\pi$ as well as possible (defined below).

Let $v_h^{k,\pi}(\xi) \in \mathbb{R}^N$ and $v_h^\pi(\xi) \in \mathbb{R}^N$ be the anchor target vectors under (empirical) greedy backup and (true) $\pi$-backup, respectively:

$$\left[v_h^{k,\pi}(\xi)\right]_n = \sum_{\xi'} \widehat{P}^n(\xi'|\xi)\left[r\big(x_h(n), a_h^k(n, \xi), \xi'\big) + \phi\big(f^a(x_h(n), a_h^k(n, \xi))\big)^\top w_{h+1}^n(\xi')\right],$$

$$\left[v_h^\pi(\xi)\right]_n = \sum_{\xi'} P(\xi'|\xi)\left[r\big(x_h(n), \pi, \xi'\big) + \phi\big(f^a(x_h(n), \pi)\big)^\top w_{h+1}^\pi(\xi')\right].$$

The LS predictor at $x_h^{a,k} := f^a(x_h^k, \Sigma_h^k)$ is $\phi(x_h^{a,n})^\top \Sigma_h^{-1} \Phi_h(\cdot)$. Hence

$$\delta_h^k(\pi) = \phi(x_h^{a,n})^\top \Sigma_h^{-1} \Phi_h\left(v_h^{k,\pi}(\xi_{k-1}^n) - v_h^\pi(\xi_{k-1}^n)\right).$$

Write, with $g_{h+1}^{k,\pi}(n, \xi') := r(\cdot) + \phi(\cdot)^\top w_{h+1}^k(\xi')$ and $g_{h+1}^\pi$ defined analogously with $w_{h+1}^\pi$,

$$v_h^{k,\pi} - v_h^\pi = \underbrace{(\widehat{P}^n - P)\, g_{h+1}^{k,\pi}}_{\text{(A) transition error}} + \underbrace{P\big(g_{h+1}^{k,\pi} - g_{h+1}^\pi\big)}_{\text{(B) propagation}} + \underbrace{\rho_h^\pi}_{\text{(C) misspecification}},$$

where $\rho_h^\pi := v_h^\pi - u_h^\pi$ and $u_h^\pi$ is the anchor vector of

$$W_h^\pi \in \arg\min_{W \in \mathcal{F}_h} \sup_{x^a} \left|(T^\pi V_{h+1}^\pi)(x^a, \xi_h^k) - W(x^a, \xi_h^k)\right|.$$

By Definition 3, $\|\rho_h^\pi\|_\infty \leq \varepsilon_{\text{BE}}$.

Apply Lemma 9 to equation G.3:

$$\left|\delta_h^k(\pi)\right| \leq \lambda_0^{-1/2}\left(\left\|(\widehat{P}^n - P)g_{h+1}^{k,\pi}\right\|_2 + \left\|P\big(g_{h+1}^{k,\pi} - g_{h+1}^\pi\big)\right\|_2 + \|\rho_h^\pi\|_2\right)$$
$$\leq \lambda_0^{-1/2}\left(\left\|(\widehat{P}^n - P)g_{h+1}^{k,\pi}\right\|_2 + \left\|g_{h+1}^{k,\pi} - g_{h+1}^\pi\right\|_2 + \sqrt{N}\,\varepsilon_{\mathrm{BE}}\right),$$

since $P$ is a contraction in $\ell_2$ and rewards/values are in $[0, H]$ so $g \in [0, H]$ coordinate-wise.

Fix $t, \xi$. Lemma 10 with a union bound over $k \leq K$, $t \leq H$, $\xi \in \Xi$ yields with probability $1 - \delta/2$ that

$$\left\|(\widehat{P}^n - P)g_{h+1}^{k,\pi}\right\|_2 \leq H\sqrt{|\Xi|}\sqrt{\frac{\log\big(2HK|\Xi|/\delta\big)}{2\,n_h^k(\xi)}}$$

uniformly. Summing these martingale-like increments along the sample path and using $\sum_{j=1}^M(n_j + 1)^{-1/2} \leq 2\sqrt{M}$ gives the contribution

$$\tilde{C}_2\,|\Xi|\,H\,\sqrt{K\log\frac{HK|\Xi|}{\delta}}$$

per stage, which after accounting for the LS geometry (the $\Sigma_h^{-1}\Phi_h$ factor) and the greedy-vs-policy coupling yields

$$C_2\,|\Xi|\Big(H^2 + H\sqrt{\tfrac{N}{\lambda_0}}\Big)\sqrt{K\log\frac{HK|\Xi|}{\delta}}.$$

Here the $H^2$ and $H\sqrt{N/\lambda_0}$ arise from $H$-step propagation/telescoping and the LS projection norm as in standard LSVI analyses; constants are absorbed.

The term $\left\|g_{h+1}^{k,\pi} - g_{h+1}^\pi\right\|_2$ is linear in $|V_{h+1}^{k,\pi} - V_{h+1}^\pi|$, hence in $|\delta_{h+1}^n(\pi)|$. Unfolding over $t = 1, \ldots, H$ and using Freedman/Bernstein-type arguments for the resulting martingale differences (and rewards bounded by 1) gives

$$C_1\,H\,\sqrt{K\log\frac{HK|\Xi|}{\delta}}.$$

By Lemma 9 and $\|\rho_h^\pi\|_2 \leq \sqrt{N}\,\varepsilon_{\mathrm{BE}}$,

$$\left|\phi(x_h^{a,n})^\top\Sigma_h^{-1}\Phi_h\,\rho_h^\pi\right| \leq \lambda_0^{-1/2}\sqrt{N}\,\varepsilon_{\mathrm{BE}}.$$

Summing over $t = 1, \ldots, H$ gives $H\lambda_0^{-1/2}\sqrt{N}\,\varepsilon_{\mathrm{BE}}$ per episode. The standard comparison of $\hat{\pi}_n$ with $\pi^\star$ doubles this constant but stays of the same order; summing over $n = 1, \ldots, K$ yields

$$C_3\,\frac{H}{\sqrt{\lambda_0}}\,K\,\varepsilon_{\mathrm{BE}}.$$

Combining (4)–(6) with a union bound over the high-probability events gives the claimed inequality with probability at least $1 - \delta$. $\qquad\square$

## H  DETAILED NUMERICAL EXPERIMENTS

### H.1  TABULAR MDP

We conduct numerical experiments using tabular Exo-MDPs, and display the model estimation error over episodes and the regret comparison of `PTO`, `PTO-Opt` and `PTO-Lite` in Figure 3. We provide the implementation details below.

- **Model estimation.** `PTO` or `LSVI-PE` estimates the model $\widehat{P}_t(y' \mid y)$ from past episodes (counts per time-step) and solves backward DP using $\widehat{P}_t$.
- **Optimistic model.** At each Bellman backup the `PTO-Opt` solves $\max_{Q:\|Q-\widehat{P}\|_1 \leq \mathrm{bonus}}\sum_{y'} Q(y')V(y')$ by mass transfer to obtain an optimistic expectation.
- **Policy evaluation.** All algorithms are evaluated by exact backward induction on the true $P_y$ to obtain stage-1 value functions $V(\cdot, \cdot, 1)$.

- **Regret and model error.** Per episode we measure instantaneous regret as $\sum_{x,y}\left(V^\star_{x,y,1} - V^\pi_{x,y,1}\right)$ and report cumulative regret $\sum_{k\leq K}$ (averaged across runs). Model error is measured by the average Frobenius norm $\frac{1}{T}\sum_t \|\widehat{P}_t - P_y\|_F$.

**Baseline methods.** We compare the `PTO` to `PTO-Opt` (Section D.3.2) that solves a constrained $\ell_1$- subproblem for optimistic model with confidence radius bonus $= c\sqrt{2Y\log(KY/0.01)/N_{t,y}}$ (default $c = 0.3$). We also implement three `PTO-Lite` baselines with subsampling ratios of 0.2, 0.5, and 0.8 for comparison, serving as the natural intermediate points between PEL algorithms and exploration-heavy methods. Instead of constructing the full empirical exogenous model from all episodes, Lite subsamples the historical exogenous transitions at each stage. The resulting subsampled dataset is used to compute a lightweight estimate, reducing computation while keeping the model statistically representative.

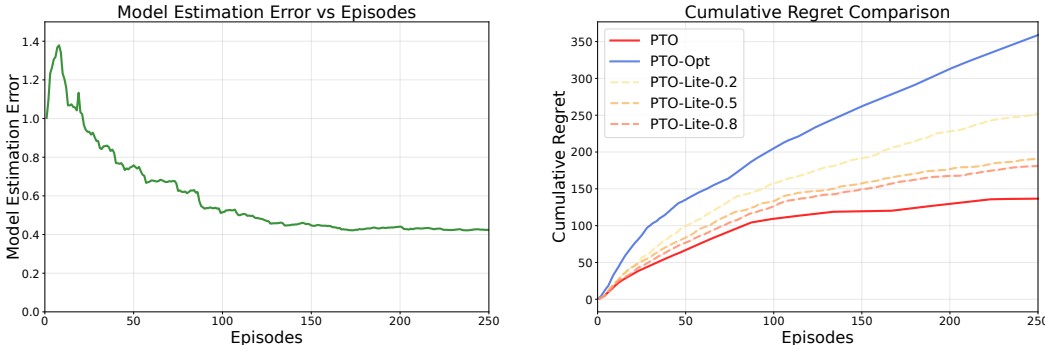

Figure 3: Comparison of `PTO`, `PTO-Opt` and `PTO-Lite`.

## H.2    STORAGE CONTROL

We display the model estimation error over episodes and the regret comparison between `LSVI-PE` and `LSVI-Opt` in Figure 4 across three Exo-MDPs with different horizon lengths. Across $H \in \{6, 8, 10\}$, `LSVI-PE` consistently outperforms `LSVI-Opt` in cumulative regret.

We also analyze three expanded Exo-MDPs with state/action spaces scaled and planning horizons increased, and the results are presented in Figure 5. `LSVI-Opt` and three `LSVI-Lite` variants are implemented as baselines. Across all these enlarged benchmarks, `LSVI-PE` maintains the best overall performance, and Lite achieves performance close to `LSVI-PE`, validating that PEL remains effective even under aggressive subsampling of exogenous traces.

We provide the pseudo-code of `LSVI-PE` for storage control in Algorithm 2.

**Baseline method.** `LSVI-Opt` differs from `LSVI-PE` in Line 12 of Algorithm 2. Specifically, `LSVI-Opt` computes the optimistic target

$$y_h^k(n) \leftarrow \sum_{\xi'\in\Xi} \tilde{P}_h^k(\xi' \mid \xi) \cdot \max_{a'\in[-a_{\max}, a_{\max}]} \left\{ r\big(g(\rho_n, \xi'), a', \xi'\big) + \phi\big(f^a(g(\rho_n, \xi'), a')\big)^\top w_{h+1}^k(\xi')\right\},$$

where $\tilde{P}_h^k$ is the optimistic model obtained by solving the $\ell_1$ constrained subproblem around $\hat{P}_h^k$ with confidence radius bonus $= c\sqrt{2Y\log(KY/0.01)/N_{t,y}}$ (default $c = 0.5$).

**Detailed setup for Case I.** We numerically analyze a storage control problem with continuous endogenous state space $\mathcal{X} = [0, C]$, discrete exogenous state space $\Xi = [Y]$, and continuous action space $\mathcal{A} = [-a_{max}, a_{max}]$. $\mathcal{X}$ is discretized by $N$ anchors. The default parameters are $C = 10$, $Y = 10$, $a_{max} = 2$, $N = 10$, and $K = 100$ epsiodes. Three time horizon lengths $H \in \{6, 8, 10\}$ are evaluated for comparison. The exogenous variable is the discrete power price with the following transition rules applied: a 70 % probability exists of either remaining in the original state or transitioning to an adjacent state, with the remaining 30 % assigned to uniform selection among all feasible states.

**Detailed setup for Case II.** We analyze three expanded storage control problem cases, where both the state spaces and planning horizons are scaled by factors of 2x to 4x relative to the original settings. To observe the long-term performance and convergence characteristics of each algorithm, we increase the number of running episodes to 200. The subsample factors of the three Lite baselines are 0.2, 0.5, and 0.8, respectively. All other experimental configurations remain consistent with Case I.

**Computational efficiency.** The major computational overhead of Algorithm 2 is to solve the optimal action for a given state $s_h^k$ at each time-step $h$

$$\hat{\pi}_h^k(x_h^k, \xi_h^k) = \arg \max_{a \in [-a_{\max}, a_{\max}]} \left\{ r(x_h^k, a, \xi_h^k) + \phi\big(f^a(x_h^k, a)\big)^\top w_{h+1}^k(\xi_h^k) \right\}.$$

We emphasize this step is computationally efficient via anchor enumeration due to the LP structure of the subproblem.

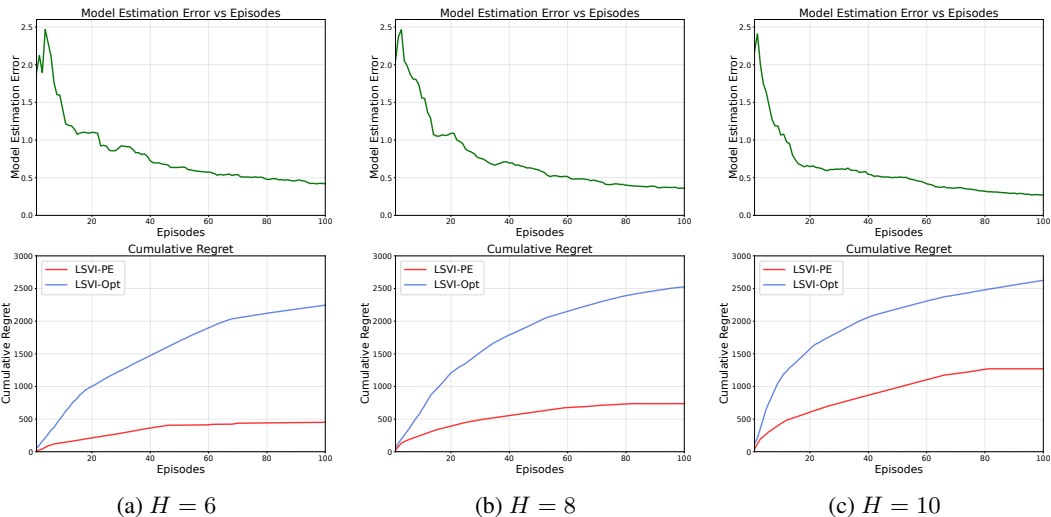

(a) $H = 6$      (b) $H = 8$      (c) $H = 10$

Figure 4: Comparison of `LSVI-PE` and `LSVI-Opt` across three different time horizon lengths in Case I.

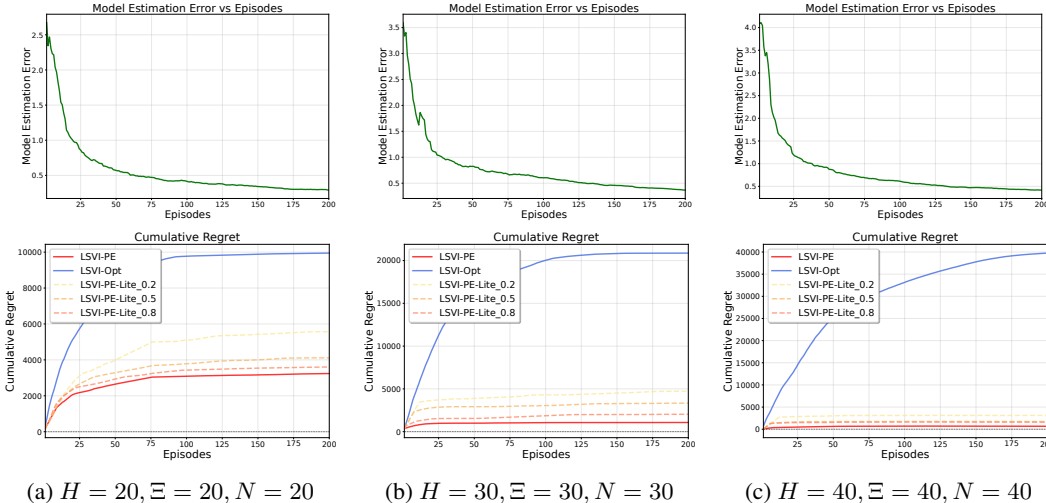

(a) $H = 20, \Xi = 20, N = 20$  (b) $H = 30, \Xi = 30, N = 30$  (c) $H = 40, \Xi = 40, N = 40$

Figure 5: Comparison of `LSVI-PE`, `LSVI-Opt` and `LSVI-PE-Lite` under three different experimental scales in Case II.

---

**Algorithm 2** `LSVI-PE` for storage control

---

**Require:** Horizon $H$; capacity $C$; anchors $\rho_n = \frac{n-1}{N-1}C$, $n = 1...N$; hat features $\phi : [0, C] \to \mathbb{R}^N$ with $\phi(\rho_n) = e_n$

**Require:** Action set $\mathcal{A} = [-a_{\max}, a_{\max}]$; efficiencies $\eta^+, \eta^- > 0$; leakage $\alpha \in (0, 1]$

**Require:** Reward $r(s, a, \xi) = \xi a - \alpha_c|a| - \beta_h s$; post-decision $f^a(s, a) = \mathrm{clip}\big(s + \eta^+ a^+ - \frac{1}{\eta^-}a^-, 0, C\big)$; pre-decision update $g(s^a, \xi') = \alpha s^a$

**Require:** Price codebook $\Xi = \{\zeta_1, \ldots, \zeta_R\}$; dataset of $k$ price trajectories $\{\xi_h^\ell\}_{\ell=1, h=1}^{k, H}$ with $\xi_h^\ell \in \Xi$

1: **Update** $\hat{P}_h^k(\cdot \mid \xi)$: for each $(h, \xi)$,

$$\hat{P}_h^k(\xi' \mid \xi) = \begin{cases} \frac{N_h^k(\xi, \xi')}{\sum_z N_h^k(\xi, z)}, & \sum_z N_h^k(\xi, z) > 0 \\ \frac{1}{R}, & \text{otherwise (unvisited row)} \end{cases}$$

where $N_h^k(\xi, \xi') = \sum_{\ell=1}^k \mathbf{1}\{\xi_h^\ell = \xi, \ \xi_{h+1}^\ell = \xi'\}$.

2: **Backward Value Iteration:**

3: **for** $h = H$ down to 1 **do**

4:     **for** each $\xi \in \Xi$ **do**

5:                      // Design at post-decision anchors (identity under hat basis)

6:         $\Phi_h \leftarrow [\phi(\rho_1), \ldots, \phi(\rho_n)]$; $a_h \leftarrow \Phi_h \Phi_h^\top$          // $\Phi_h = I_K$, $a_h = I_K$

7:         $b_h^k(\xi) \leftarrow \mathbf{0} \in \mathbb{R}^K$

8:         **for** $n = 1$ to $N$ **do**

9:             **if** $h = H$ **then**

10:                 $y_h^k(n) \leftarrow 0$

11:             **else**

12:                 $y_h^k(n) \leftarrow \sum_{\xi' \in \Xi} \hat{P}_h^k(\xi' \mid \xi) \cdot \max_{a' \in [-a_{\max}, a_{\max}]} \Big\{ r\big(g(\rho_n, \xi'), a', \xi'\big) + \phi\big(f^a(g(\rho_n, \xi'), a')\big)^\top w_{h+1}^k(\xi') \Big\}$

13:                  // Inner max is 1-D LP (piecewise linear); solve via breakpoint enumeration

14:             **end if**

15:             $b_h^k(\xi) \leftarrow b_h^k(\xi) + \phi(\rho_n)\, y_h^k(n)$               // writes $y_h^k(n)$ into entry $k$

16:         **end for**

17:         $w_h^k(\xi) \leftarrow \Sigma_h^{-1} b_h^k(\xi)$          // with $\Sigma_h = I_N$: $w_h^k(\xi) = [y_h^k(1), \ldots, y_h^k(N)]^\top$

18:     **end for**

19: **end for**

20: **Output:**

21: $\hat{V}_h^{k,a}(x^a, \xi) = \phi(x^a)^\top w_h^k(\xi)$,    for all $(s^a, \xi, h)$

22: **return** $\hat{V}_h^{k,a}$

---

# I  DECLARATION OF THE USE OF LARGE LANGUAGE MODELS

We used large language models (LLMs) to assist in proofreading and improving the language, grammar, and clarity of this manuscript. The authors retain full responsibility for all intellectual content, results, and claims presented in this paper.

