# OpenReview forum: "Is Pure Exploitation Sufficient in Exogenous MDPs with Linear Function Approximation?"
_ICLR.cc/2026/Conference — ICLR 2026 Poster_

### Official Review · Reviewer_4oif · 2025-10-23

**Soundness:** 3
**Presentation:** 3
**Contribution:** 3
**Rating:** 6
**Confidence:** 2

**Summary:**

The paper shows that pure exploration—model estimation with greedy action execution—is sufficient for rate-optimal regret in exogenous MDPs when the exogenous process is the only unknown. The paper covers both the tabular MDP case and the case of MDPs with linear function approximation. In both cases, they show a regret of order $O(\sqrt{K})$.

**Strengths:**

- The paper is well written and easy to follow. Overall, the authors do a good job of providing intuition about the results and explaining why the algorithm works.
- I am not up to date with the literature on exogenous MDPs, but from what I can gather, the results seem novel and interesting (I did not have time to check all the proofs).

**Weaknesses:**

- The main weakness of the paper is that the setting is quite specific. In particular, the assumption that the only unknown component is the exogenous process seems quite strong to me. The authors justify this choice as being common in the literature, but I do not have enough knowledge of the literature to judge this claim. However, I can understand that in certain operational research applications, this assumption might be reasonable.

**Questions:**

- Could you provide some intuition as to why pure exploration is sufficient to achieve rate-optimal regret in this setting? Is it because the endogenous process is fully known, and the exogenous process is independent of the actions, so that it can be learned at a "fast" rate (approximately $1/\sqrt{K}$) without impacting the regret too much?
- In the linear function approximation setting, could you comment on how the regret depends on the problem parameters (such as d, anchor points, etc.)? Is this dependence optimal, or could it be improved?
- The assumption that anchor points are known seems strong. Could you comment on this assumption? Is it possible to relax it?

Overall, I think the paper provides interesting and novel results on exogenous MDPs, even though the setting is quite specific.

---

> ### Author Response · Authors · 2025-11-22
> **Response**
>
> Thanks for your feedback! We address your comments below and have posted a revised version. During the response we refer to the global response for more details.
>
> ## W1
> *The main weakness of the paper is that the setting is quite specific. In particular, the assumption that the only unknown component is the exogenous process seems quite strong to me. The authors justify this choice as being common in the literature, but I do not have enough knowledge of the literature to judge this claim. However, I can understand that in certain operational research applications, this assumption might be reasonable.*
>
> We appreciate this comment and address it extensively in **GR3 A)** in Global Response.
>
>
> ## Q1 intuition on why pure exploration is sufficient to achieve rate-optimal regret
>
> Thank you for this insightful question. Yes, the key intuition is exactly as you describe. Because the endogenous dynamics $f$ are fully known and the exogenous process evolves independently of the learner’s actions, each trajectory provides an unbiased sample of the exogenous trace regardless of the policy used to generate it. Conditioned on this trace, the endogenous evolution is deterministic, allowing us to compute counterfactual value functions for any policy. We provide a detailed response in **GR2** in Global Response.
>
>
>
> ## Q2
> *In the linear function approximation setting, could you comment on how the regret depends on the problem parameters (such as d, anchor points, etc.)? Is this dependence optimal, or could it be improved?*
>
> We thank the reviewer for this question and are happy to clarify the dependence on the problem parameters in the linear function approximation (LFA) setting.
>
> Our main regret bound in the LFA setting (Theorem 2) is
>
> $$
> \widetilde{O}\Big(
> \big(\sqrt{N/\lambda_0} + \sqrt{d}\big)|\Xi|H\sqrt{K}
> \Big),
> $$
>
> where
> - $d$ the feature dimension of the post-decision value approximation,
> - $N$ the number of anchor points,
> - $\lambda_0 := \min_{h \in [H]} \lambda_{\min}(\Sigma_h)$ is the minimum eigenvalue of the anchor covariance matrices $\Sigma_h = \Phi_h \Phi_h^\top$,
> - $|\Xi|$ is the size of the exogenous state space,
> - $H$  the horizon,
> - $K$  the number of episodes.
>
> Thus, the regret is controlled by
> - Function class complexity via the term $\sqrt{d}$,
> - Anchor design and conditioning via the term $\sqrt{N/\lambda_0}$,
> - Exogenous component and horizon via the factor $|\Xi| H$,
>
> while remaining independent of the cardinality of the endogenous state and action spaces.
>
> The anchor-related terms reflect standard conditioning phenomena. The number of anchors $N$ captures how many distinct anchor post-decision states we use per stage, while $\lambda_0$ measures how well these anchors span the feature space. In well-designed systems we expect $N = \Theta(d)$ and $\lambda_0 = \Theta(1)$, in which case
> $$
> \sqrt{N/\lambda_0} = \Theta(\sqrt{d})
> \quad\text{and}\quad
> \mathrm{CR}(\text{LSVI-PE}, K)
> = \widetilde{O}\big(|\Xi|H\sqrt{dK}\big).
> $$
> Under the stronger Assumption 4, Theorem 3 further improves the $d$-dependence to
> $$
> \widetilde{O}\Big((H + \sqrt{N/\lambda_0})|\Xi|H\sqrt{K}\Big),
> $$
> which can be strictly better when $H \ll d$.
>
> Regarding optimality, in closely related settings such as linear bandits and linear MDPs, lower bounds show that a $\sqrt{dK}$ dependence is information-theoretically unavoidable (up to logarithmic and horizon factors). In the well-conditioned regime $N \asymp d$, $\lambda_0 = \Theta(1)$, our bound matches this $\sqrt{dK}$ scaling (modulo the problem-specific factors $|\Xi|$ and $H$), so we view the $\sqrt{d}$ dependence as essentially optimal for generic linear function classes. The additional factor $\sqrt{N/\lambda_0}$ captures the statistical and numerical cost of using a finite, potentially ill-conditioned set of anchors, and is standard for anchor-/coreset-based approximations.
>
> In the revision, we  add a short discussion after Theorems 2–3 that  decomposes the regret into its dependence on $d$, $N$, $\lambda_0$, $|\Xi|$, and $H$, and (ii) explains that in the well-conditioned regime the bound simplifies to $\widetilde{O}(|\Xi| H \sqrt{dK})$, which is consistent with known optimal rates in linear settings.
>
> ## Q3
> *The assumption that anchor points are known seems strong. Could you comment on this assumption? Is it possible to relax it?*
>
> Please refer to **GR4 A)** and **GR4 D)** in Global Response.

---

### Official Review · Reviewer_8gd3 · 2025-10-25

**Soundness:** 2
**Presentation:** 2
**Contribution:** 3
**Rating:** 4
**Confidence:** 3

**Summary:**

This paper studies exogenous MDPs where the noise in the transitions is independent on the action taken by the learner.
Under this setting the authors show that no exploration mechanism is needed in Bandits Problem and Tabular MDPs where the transition dynamic is known.
Moreover, the result is extended to the linear case.

**Strengths:**

I think that the question tackled here is interesting, as several problems of practical interest are usually solved by algorithms that do not use any exploration mechanism.

**Weaknesses:**

I think that the assumption about the existence of the anchor set is not strong; however, I do think that it is a strong assumption that the learning algorithm knows this set.
Why is it not possible to guarantee the invertibility of $\Sigma$ by defining $\Sigma = \sum^N_{n=1} \phi_n \phi_n^T + \beta I$ where $\beta$ is a small scalar and $I$ is the identity matrix ? In this way, switching from Least Square to Ridge regression it should be possible to avoid the assumption.

Why do you use the anchor points to define the matrix $\Sigma$ instead of using the data collected during the policy rollout phase? It seems weird to me that the rollout phase only uses the encountered exogenous states to estimate the exogenous to exogenous transition matrix. In contrast, the encountered states and actions are discarded.

**Questions:**

1) How is the initial exogenous state chosen? Is it sampled from a fixed distribution, or can it be chosen adversarially?

2) If the transition dynamics (the $f$) in tabular MDPs were not known, would pure exploitation still suffice?

3) In the tabular guarantees, there is no dependence in the number of endogenous states and actions ($S$ and $A$). However, the feature dimension $d$ shows up in Theorem 2? If I represent a tabular MDP by choosing the features that are one one-hot encoded vector of dimension $d = SA$, I would get a dependence on $\sqrt{SA}$  if I apply your Theorem 2 in this setting, while Theorem 1 avoids this dependence.

---

> ### Author Response · Authors · 2025-11-22
> **Response 1/2**
>
> Thanks for your feedback! We address your comments below and have posted a revised version. During the response we refer to the global response for more details.
>
> For expository purposes we first address W2:
>
> ## W2 why use the anchor points to define the matrix $\Sigma$ instead of using the rollout data; the encountered states and actions are discarded
>
> We are studing the effectiveness of Pure Expoitation in Exo-MDP with LFA, which has fundamental difference to optimism-based algorithm/analysis in general RL with LFA. Note that under LFA, we have
> $$ V_h^{\pi,a}(x^{a},\xi) = \phi(x^{a})^{\top} w_h^{\pi}(\xi). $$
>
> Traditional design choice of $\Sigma$ via rollout states fails for PEL, due  to failures. We deliberately define $\Sigma$ from fixed anchors to resolve this challenge.
>
> - **Failure 1: Rollout states in Pure exploitation fail in coverage.** In pure exploitation, the visited states concentrate on those from the  greedy policies $\pi^k$. Without exploration, the rollout distribution fail to cover regions of the feature space that are crucial for $V^\star(x^a,\xi)$. This problem is solved in optimism-based algrithm for standard linear MDP by enforcing optimism (e.g., LSVI-UCB [1], UCRL-VTR [2]), which encourage the algorithm to cover the rarely visited states. However, in PEL, this prevents the algorithm to effecitively construct a well-covered states. We resolve this by directly working with fixed states that provide sufficient coverage of feature directions, e.g., a grid in the storage level in our resource-management examples (Section 6.2).
>
>
> - **Failure 2:  optimism-based analysis fails.** This is related to the issue clarified for tabular case. We need to bound two value gaps, one of which corresponds to the policy $\pi^*$ rather than $\pi^k$. If instead we constructed $\Sigma$ from the rollout states, then $\Sigma$, the targets, and the exogenous model would all be jointly *data-dependent*, and in particular adapted to the same exogenous filtration driving the martingale differences. This breaks the clean separation we exploit in Appendix G: the self-normalized martingale tools used in optimism-based linear MDP analyses would be needed, but those rely on explicit exploration to guarantee sufficient coverage of the feature space, which we deliberately avoid in pure exploitation. In contrast, the counterfactual analysis in Appendix G uses the fact that $\Sigma_h$ is fixed and data-independent, so that we can (i) first bound the anchor-level Bellman regression errors using standard concentration for the exogenous model; (ii) then propagate these errors to arbitrary states with constants depending only on $\lambda_0$.
>
>
> **Why it is harmless to *discard* on-policy endogenous states.**
> - The action chosen does not affect the estimation of exogenous process. Fixed states that sufficiently covers the endogenous space is OK.
> - While we do not use on-policy endogenous states, we use historical exogeneous states to estimate the exogenous transition kernel $\hat{P}$, which is the only unknown component affecting the Bellman targets.
>
>
> ***
> References
>
> [1] Jin, Chi, et al. "Provably efficient reinforcement learning with linear function approximation." Conference on learning theory. 2020.
>
> [2] Ayoub, Alex, et al. "Model-based reinforcement learning with value-targeted regression." International Conference on Machine Learning. PMLR, 2020.
>
> ## W1
>
> **strong assumption that the learning algorithm knows this set**
>
> Please refer to **GR4 A)** in Global Response.
>
>
> **why not ridge regression**
>
> Please refer to **GR4 C)** in Global Response.
>
>
>
> ## Q1
> > How is the initial exogenous state chosen? Is it sampled from a fixed distribution, or can it be chosen adversarially?
>
> Our regret guarantees hold in the worst case and do not rely on any assumptions about how the initial exogenous state is chosen. The initial exogenous state can be sampled from any fixed starting distribution. In particular, the initial exogenous state may be arbitrary or even adversarial.  We've updated Section 3 to clarify this point.
>
> ## Q2
> > If the transition dynamics in tabular MDPs were not known, would pure exploitation still suffice?
>
> Please refer to **GR3 A)** in Global Response.

---

> > ### Author Response · Authors · 2025-11-22
> > **Response 2/2**
> >
> > ## Q3
> > > In the tabular guarantees, there is no dependence in the number of endogenous states and actions. However, the feature dimension shows up in Theorem 2? If I represent a tabular MDP by choosing the features that are one one-hot encoded vector of dimension $d=SA$, I would get a dependence on $\sqrt{SA}$ if I apply your Theorem 2 in this setting, while Theorem 1 avoids this dependence.
> >
> > The difference comes from the fact that Theorem 1 and Theorem 2 analyze different regimes and different algorithms, and we do not intend Theorem 2 as a tighter rederivation of the tabular result.
> >
> > The key reason is that Theorem 1 is a tabular result for PTO, where we explicitly work with finite endogenous state–action spaces and can run *exact* dynamic programming. This is why the regret bound avoids any dependence on $|X|$ and $|A|$. In contrast, Theorem 2 is a linear function approximation  result for LSVI-PE, intended for continuous endogenous state–action spaces, where the natural complexity parameter is the feature dimension $d$.
> >
> > We would like to clarify that if we force the tabular MDP into the LFA setting by using one-hot features, then the feature dimension would be $d = |X|$ rather than $d=|S||A|=|X||\Xi||A|$. The reason is that we work with the post-decison state space, which is the same as the endo. state space $X$. Then applying Theorem 2 indeed yields a $\sqrt{X}$ dependence. When we use the most over-parameterized representation (one-hot over all tabular states/actions), we necessarily inherit a $\sqrt{d}$ dependence, as is standard in linear-MDP/LFA regret bounds.
> >
> > By contrast, Theorem 1 exploits that in the tabular case we do not need to go through a generic least-squares approximation: we can write exact Bellman equations on the finite state space and analyze PTO directly, which is why $|X|$ and $|A|$ drop out of the regret scaling even though they still enter the computational complexity.
> >
> > Overall, we add a remark after Theorem 2 stating that:
> > - Theorem 2 is not intended to improve upon Theorem 1 in the tabular regime. The two results are complementary.
> > - If one instantiates LSVI-PE with one-hot tabular features, Theorem 2 indeed yields a $\widetilde{O}(|\Xi|H\sqrt{SAK})$ bound, while Theorem 1 gives the sharper $\widetilde O(|\Xi|H^2\sqrt{K})$ bound by exploiting the *exact tabular structure*.
> > - In the intended LFA regime, $d$ is the appropriate complexity measure and independence from $|X|,|A|$ in Theorem 2 is meaningful and stronger than any tabular dependence.

---

> ### Comment · Reviewer_8gd3 · 2025-11-23
>
> Thank you for your long answer.
>
> I think that my understanding is that pure exploitation suffices for sublinear regret only if strong assumptions are met: 1) Known dynamics 2) Known anchor states set. Please correct me if I am wrong.
>
> If my understanding of the contribution is correct I find that it operates under excessively strong assumptions under which is not very surprising that no exploration is needed.
>
> Given that I currently keep my original score.
>
> Please let me know if I am missing any important part of your contribution.
>
> Best,
>
> Reviewer

---

> > ### Author Response · Authors · 2025-11-24
> >
> > Thank you for your prompt follow-up and for clarifying your understanding. We would like provide clarifications on assumptions and highlight our key contributions as follows.
> > - **Known dynamics**. Our setting does NOT assume a known transition kernel. What is known is only the deterministic endogenous map $x'=f(x,a,\xi')$. All randomness comes from the exogenous Markov chain $\xi' \sim P_{\Xi}(\cdot|\xi)$, whose kernel is **unknown** and must be learned. Equivalently, the endogenous transition kernel is stochastic and unknown
> > $$ x'\sim P_{X}(\cdot|x,a,\xi):=\sum I\\{x'=f(x,a,\xi')\\} \cdot P_{\Xi}(\xi' \mid \xi). $$
> > The Markovian exogeneous transition $P_{\Xi}(\cdot|\xi)$ is also stochastic and unknown.  The full transition kernel from state $s = (x, \xi)$ to $s' = (x', \xi')$ is stochastic and **unknown**
> > $$P(s' | s,a) = 1[f(x,a,\xi') = x'] \cdot P_{\Xi}(\xi' \mid \xi).$$
> > Deterministic map $f$ + unknown noise $P_{\Xi}$ is still an **unknown** kernel. Thus we are NOT in a *trivial and degenerated*  regime of a known MDP, and the regret bounds are obtained while learning purely from greedy trajectories.
> >
> > - Both operation research and ADP in practice and theoretic RL community work with the assumption of known endogeous map and known anchor set.
> >     - The  known endogeous map is an intrinsic, standard property of Exo-MDP. As a structured sub-class of MDPs, Exo-MDPs with property of known endogeneous map  have been studied in a growing line of work [1-11]. In particular, [4,8] considers exactly the same assumption, but in a less general setting with **discrete** endogenous dynamics and **i.i.d. (non-Markovian)** exogenous processes and focus on exploration/optimism methods.
> >     - The known anchor set is not a hidden property of the environment, but an **algorithmic design choice** that is standard in ADP and matches how Exo-MDPs are implemented in practice. Anchor states are standard in ADP for control and operation research applications [1,9-11]. In inventory control and storage systems [9-10], practitioners exploit *domain knowledge* and *problem structure*, e.g.  piecewise-linearity or convexity of the value functions [1]. With piecewise-linear  hat features, anchors are just the breakpoints of the basis functions [9-10].  This is also shown in our storage control example in Section 6.2.
> > - **Our key contribution**. We provide a unified framework for the design and analysis of Pure Exploitation Learning for both tabular and LFA Exo-MDPs,  bridging the gap between ADP/OR practice and the theoretical RL literature on Exo-MDPs.
> >     - On the ADP side, several works shows that greedy can succeed, but  offer only **asymptotic convergence** results in **highly structured** (piecewise-linear/convex) models [9-11]. In particular, [9] provides the first asymptotic convergence of pure exploitation for a special case of Exo-MDP where the optimal value functions are piecewise linear, with integer break points and concave in the exogeneous variable.
> >     - On the RL theory side, recent Exo-MDP analysis [4,8] assume **discrete** endogenous dynamics and **i.i.d. (non-Markovian)** exogenous processes and focus on exploration/optimism, which is a less general setting.
> >
> > In contrast, we study Exo-MDPs with **continuous** endogenous states and **Markovian** exogenous processes, and provide the first near-optimal **finite-sample** regret guarantees for pure exploitation in this more general setting.  Furthermore, our results strengthen the traditional theory and practice that pure exploitation can converge asymptotically in structured Exo-MDP. Our sublinear regret guarantees ensure that the PE is statistically efficient for **any sample size** and **general structure**, making it reliably effective in practice and safe for practitioners to use.
> >
> > - **Our main technical contribution**. Even under these assumptions, establishing sublinear regret gaurantee is nontrivial. Classical optimism-based analyses fails for PEL. We introduce a new regret decomposition with two value gaps and a **counterfactual analysis** to control the optimal-policy gap using only exogenous counts. We exploit the policy-independence of the exogenous process and obtain regret bounds that are independent of the endogenous state and action space sizes. This analysis is fundamentally different from standard tabualr/linear-MDP optimism proofs and is of independent interest.
> >
> > We hope this clarifies that (i) the transition kernel is stochastic and unknown, only the endogenous map is specified; (ii) the structural assumptions are standard and aligned with existing Exo-MDP, ADP, and OR practice rather than our artificial strengths; and (iii) our main contribution lies in providing the first finite-sample theory and unified algorithmic framework for pure exploitation in a more general Exo-MDP setting, together with a new analytic approach. If you feel that any of these points are still under-emphasized in the paper, we would be very grateful for further suggestions.

---

> > > ### Author Response · Authors · 2025-11-24
> > >
> > > ***
> > > References
> > >
> > > [1] W. Powell. Reinforcement Learning and Stochastic Optimization: A Unified Framework for Sequential Decisions, 2022.
> > >
> > > [2] T. Dietterich et al. Discovering and removing exogenous state variables and rewards for reinforcement learning. ICML, 2018.
> > >
> > > [3] Yonathan Efroni et al. Sample-efficient reinforcement learning in the presence of exogenous information. COLT, 2022.
> > >
> > > [4] Sean R Sinclair et al. Hindsight learning for mdps with exogenous inputs. ICML 2023.
> > >
> > > [5] Jiekun Feng et al. Scalable deep reinforcement learning for ride-hailing.
> > > ACC, 2021.
> > >
> > > [6] Matias Alvo et al. Neural inventory control in networks via hindsight
> > > differentiable policy optimization. arXiv preprint, 2023
> > >
> > > [7] Haozhe Chen et al. Qgym: Scalable simulation and benchmarking of queuing network controllers. arXiv preprint, 2024.
> > >
> > > [8] Jia Wan et al. Exploiting exogenous structure for sample-efficient reinforcement learning. arXiv preprint arXiv:2409.14557, 2024.
> > >
> > > [9] J. Nascimento  et al. An optimal approximate dynamic programming algorithm for the lagged asset acquisition problem. Mathematics of Operations Research, 2009.
> > >
> > > [10] J. Nascimento et al. An optimal approximate dynamic programming algorithm for concave, scalar storage problems with vector-valued controls. IEEE Transactions on Automatic Control, 2013.
> > >
> > > [11] D. Jiang et al. An approximate dynamic programming algorithm for monotone value functions. Operations research, 2015.

---

> > > > ### Comment · Reviewer_8gd3 · 2025-11-25
> > > >
> > > > Thanks a lot for your answer.
> > > >
> > > > My understanding is that other works such as [8] already solves this task using exploration schemes and do not need at all a known anchor states set.
> > > >
> > > > Therefore, my opinion is that the contribution of the current work crucially relies on this new assumption that was not imposed in [8].
> > > >
> > > > Sorry, [8] does not seem to require the endogenous map $f$.
> > > >
> > > > Given these limitations, I stand with my original assessment of the paper.
> > > >
> > > > Best,
> > > > Reviewer

---

> > > > > ### Author Response · Authors · 2025-11-25
> > > > >
> > > > > Thank you for the follow-up and highlighting the comparison with Wan et al. (2024) [8]! We aggree that [8] provides important progress in Exo-MDPs, and we are studying the same subclass of MDPs. However, we would like to clarify some important differences in setting and assumptions.
> > > > >
> > > > >
> > > > > First, [8] studies a simpler, fully tabular setting: endogenous state space $\mathcal{X}$, exogenous state space $\Xi$, and action spaces $\mathcal{A}$ are all **finite / discrete**, and the main focus  is on  **iid (non-Markovian)** exogenous processes  $\xi_h \sim P$ that do not depend on $\xi_{h-1}$ and $h$, with a Markovian extension. In contrast, our work treats  **Markovian** and **non-stationary** exogenous processes
> > > > >
> > > > > $$\xi_{h+1} \sim P_h(\cdot \mid \xi_{h}),$$
> > > > >
> > > > > where each $\xi_{h+1}$ depends on $\xi_{h}$, and the exogeneous kernel varies across $h$. Furthermore, we consider **continuous** endogenous states and actions, which is the standard regime in many OR applications (e.g., storage, inventory).
> > > > >
> > > > > Second, [8] also work under the assumption that the endogenous map $f$ and reward map $r$ are **known** (see their Equation (1a)-(1b) in Section 3.1), just as in our formulation.
> > > > >
> > > > > Our contributions extend beyond [8] in two ways:
> > > > >
> > > > > - In the tabular Exo-MDP setting (the regime of [8]), [8] shows that pure exploitation (their Plug-In methods) attains regret $\tilde{\mathcal{O}}(H^{2} \sqrt{|\Xi|K})$ under iid exogeneous input and  $\tilde{\mathcal{O}}(H^{2} |\Xi|^2\sqrt{K})$ under **Markovian** exogeneous input (extension in their Section 6.2). In contrast, we obtain a regret bound of $\tilde{\mathcal{O}}(H^{2} \sqrt{|\Xi|K})$ in Theorem 1, improving by a factor of $|\Xi|^{\frac{3}{2}}$. Here we translate their results to the non-stationary case with a standard $\sqrt{H}$ factor.
> > > > > - More importantly, we provide the first regret guarantees for pure exploitation in Exo-MDPs with continuous endogenous state/action spaces via LFA. This continuous setting lies outside the scope of [8], and here the anchor set appears as a standard algorithmic device (analogous to choosing knots or grid points) rather than an oracle assumption.  In the simpler, tabular setting of [8], we emphasize that the anchor sets can just be taken to be all pairs, and hence this assumption really is only needed for the continuous setting.
> > > > >
> > > > > We have updated the related work (Exo-MDPs in Section 2) to highlight these distinctions. We appreciate your comments and are happy to discuss further if anything remains unclear.

---

### Official Review · Reviewer_4nVr · 2025-10-30

**Soundness:** 3
**Presentation:** 2
**Contribution:** 3
**Rating:** 6
**Confidence:** 4

**Summary:**

The paper makes a contribution to the reinforcement learning literature attempting to characterize when greedy policies are sufficient for provable guarantees, or when exploration is not necessary. The authors show that this is indeed the case for exogenous MDPs, where there is an exogenous Markovian component that is unaffected by the learner's actions. Their variant of LSVI achieving sublinear regret crucially hinges on an assumption that the endogenous transitions are deterministic and known, as well as three technical assumptions on an anchor set that is closed under the Bellman operator with non-negative residuals. A tabular method only requires the former.

**Strengths:**

- The paper makes a solid contribution to the RL literature. Understanding exactly when greedy exploitation is sufficient for provable guarantees is a topic that has gotten more attention lately.
- This tends to occur when there is sufficient environment noise, and Exo-MDPs appear to be one such case. To my knowledge, this is novel.
- The authors tackle bandits and both tabular and linear MDPs, showing that this holds somewhat more broadly than just in an isolated case.
- The paper is largely well written, save for a few issues on clarity.

**Weaknesses:**

1. Not much intuition is provided on exactly why such a positive result is possible. At a very very high level that probably amounts to skimming over a lot of subtleties, it seems to be because the learner can decouple the exogenous transitions, of which no exploration is necessary to learn them, from the endogenous transitions, whom are deterministic and known. Assuming that one can do so, the whole problem then reduces to learning the exogenous transitions for input to learning a Q-function via the standard LSVI procedure.
2. The assumptions are quire strong. Needing the existence of an anchor set that is known to the learner, plus additional assumptions on said set, is quite strong in the RL literature (I am reading this as its alternate name, a coreset). It is understandable that strong assumptions are necessary, but one could also (for instance) assume that the minimum eigenvalue of the design matrix is bounded under any policy. Characterizing exactly why this is needed, how it can be obtained (it is folklore that Frank-Wolfe-like procedures get you one, but it should be stated), and the relevant intuition would go a long way towards helping the unfamiliar reader and justifying these assumptions.
3. The proofs in the appendix, especially for the linear MDP section, are quite poorly written. The authors state that they make a counterfactual analysis. It is unclear to me where this is done, or what exactly this means. This is striking -- I've seen the linear MDP proofs many, many times, but it's not clear to me where it happens.

**Questions:**

1. How much of this relies on the assumption of known deterministic endogenous dynamics?
2. Can the anchor set assumption be weakened? What could replace it?
3. Where is the counterfactual analysis?

---

> ### Author Response · Authors · 2025-11-22
> **Response 1/3**
>
> Thanks for your feedback! We address your comments below and have posted a revised version. During the response we refer to the global response for more details.
>
> ## W1
> *Not much intuition is provided on exactly why such a positive result is possible. At a very very high level that probably amounts to skimming over a lot of subtleties, it seems to be because the learner can decouple the exogenous transitions, of which no exploration is necessary to learn them, from the endogenous transitions, whom are deterministic and known. Assuming that one can do so, the whole problem then reduces to learning the exogenous transitions for input to learning a Q-function via the standard LSVI procedure.*
>
> Thank you for this excellent observation, the intuition you describe is precisely the structural reason pure exploitation succeeds in Exo-MDPs. We address this point in **GR2** in Global response.
>
>
> ## W2/Q2
> Thank you for the careful comments about the strength and interpretation of our anchor assumptions (Assumptions 1–4) and their relation to coresets and Frank–Wolfe constructions. We provide point-by-point clarification below.
>
> **A) "known anchor set" "construction of anchors"**
> Please refer to **GR5 A)** in Global Response.
>
>
> **B) "connections to coreset and Frank-Wolfe procedures"**
>
> Thank you for pointing out the connection between our anchor set and coreset constructions. Indeed, the anchors in Assumption 1 act as a small, well-conditioned spanning set of feature vectors, analogous in spirit to coresets used in linear RL and ADP. We clarify the connections and differences below.
>
> Anchor sets serve the similar purpose as coresets in linear RL and ADP. In our paper, anchor sets provide a well-conditioned spanning set of feature vectors enabling stable value regression and Bellman transport. This is similar to the concept of coresets or representative set in linear bandits and RL, e.g. Mahalanobis-distance representative sets in [1] with well-conditioned feature coverage, optimal design in [2] with minimal set of points that well-condition the Gram matrix, coreset with well-conditioned feature coverage in [3].
>
> Regarding the connections between coreset and Frank-Wolfe (FW) methods, which arises in convex optimization. Specifically, the coreset there represents small subset of points that approximately represents a much larger dataset for the purpose of convex optimization. FW constructs such coresets by iteratively selecting “anchor” points via the linear minimization oracle. This idea has been used in problems such as learning convex bodies [4], sparse convex optimization [5], and practical large-scale machine learning [6]. However, FW-based coreset construction has largely remained within the convex optimization literature rather than RL settings involving linear function approximation.
>
> In contrast, in our Exo-MDP setting, the anchor set is designed a priori using domain knowledge and problem structure. In many structured control problems they can be constructed a priori, without observing any data. For example, in inventory or resource-storage problems, anchors can be naturally chosen as a grid of storage levels, e.g., extreme low / high and  intermediate points, which are natural design points for value approximation.
>
> Extension from domain-driven anchors to algorithmic construction. Our currunt method leverages domain knowlege or problem struture. When such problem-dependent design choice is difficult, it is a promising direction to adapts anchor selection online. We first generate large pool of candidate post-decision states, and adaptively construct the anchors by FW-like methods.
>
>
> **C) "assume that the minimum eigenvalue of the design matrix is bounded under any policy"**
>
> We note that this assumption resembles the Covariate Diversity condition used in exploration-free linear bandits [7–8] and linear RL [9], where ensuring $\lambda_{\min}(\Sigma^\pi) > 0$ is the key structural requirement enabling sample-efficient learning without explicit exploration. However, in our Exo-MDP setting this assumption plays a very different role, and in fact becomes insufficient and incompatible with the structure of the problem. Requiring the learner to guarantee $\lambda_{\min}(\Sigma^\pi) > 0$ under every policy contradicts the nature of Exo-MDPs: in general MDPs, active exploration is necessary precisely to keep $\lambda_{\min}(\Sigma^\pi)$ well-behaved; but in Exo-MDPs the policy does not influence the exogenous trajectory, so the learner cannot manipulate the state distribution to enforce diversity. As a result, $\Sigma^\pi may$ be poorly conditioned for perfectly valid policies, and the learner has no mechanism to rectify this. We have added a clarification of this point in the revised paper.
>
> ## W3/Q3
> **A) proofs for the linear MDP section**
> Thank you for the helpful comments. In the revision we have substantially rewritten and reorganized the proofs for clarity and consistency.

---

> ### Author Response · Authors · 2025-11-22
> **Response 2/3**
>
> In particular, all proofs in the appendix, especially the linear MDP case, have been revised for consistency, clarity, and completeness, ensuring that the logical flow and notation align with the main text.
>
> **B) where counterfactual analysis is done, or what exactly this means**
> Thank you for pushing us to clarify (i) why standard optimism-based analyses are not sufficient in our setting, and (ii) what we mean by *counterfactual trajectories* and where exactly they enter the proofs. We now provide discussion on why classical analysis fails and clarification of counterfactual analysis. The counterfactual analysis is one of our core technical contributions, as opposed to classical optimism-based analysis for exploration-heavy methods.
>
> ### B.1) why (classical) optimism-based analysis would fail for Pure Exploitation Learning $\Longrightarrow$ counterfactual analysis
> We use tabular Exo-MDPs to illustrate why optimism-based analysis fails for Pure Exploitation Learning.
>
> In the tabular Exo-MDP, the only unknown is the exogenous Markov chain $P_{\Xi}$. A natural first attempt is to apply a classical optimism-based analysis, leading to the decomposition of regret as
>
> $$\text{Regret} = \sum_{k=1}^K V^{\star}_1-V^{\pi^k}_1 = \sum\_{k=1}^K V_1^{\pi^{\star}}-\widehat{V}_1^{k,\pi^{k}}+\widehat{V}^{k,\pi^{k}}_1-V_1^{\pi^k}. $$
>
> Optimism-based analysis starts with the fact that the first term is nonpositive due to optimism, and bound the regret by the value gap $\widehat{V}^{k,\pi^{k}}_1-V_1^{\hat{\pi}^k}$.
>
> The value gap is  bounded by applying a simulation lemma.  This is exactly the step where the classical optimism-based proofs use a clean telescoping argument, and it fails for Pure Exploitation here: the optimism doe not hold. In the tabular Exo-MDP proof for PEL (Section 4 and Appendix F.2) we do not rely on optimism to control regret. Instead, we consider a **new regret decomposition**
>
> $$
> \begin{aligned}
>  \text{Regret} = \sum_{k=1}^K V^{\star}_1-V^{\pi^k}_1 &= \sum\_{k=1}^K  (V_1^{\pi^{\star}}-\widehat{V}_1^{k,\pi^{\star}}) +(\widehat{V}_1^{k,\pi^{\star}} - \widehat{V}^{k,\pi^k}_1)+(\widehat{V}^{k,\pi^k}_1-V_1^{\pi^k}) \\\\
> &\leq  \sum\_{k=1}^K \underbrace{(V_1^{\pi^{\star}}-\widehat{V}_1^{k,\pi^{\star}})}\_{(I)} + \underbrace{(\widehat{V}^{k,\pi^k}_1-V_1^{\pi^k})}\_{(II)}
> \end{aligned}
> $$
>
> We do not use the optimism, but the optimality of $\pi^k$ w.r.t. the estimated model $\hat{P}^k$ to bound the regret by **double value gaps**. Both **(I)** and **(II)** corresponds to the value gap of the same policy (either $\pi^{\star}$ or $\hat{\pi}^k$) evaluated between the true model $P$ and empirical model $\hat{P}^k$，so an native attempt bounds each term by using *simulation lemma*. **(II)** can be bounded using on-policy analysis, similar to optimism-based analysis, where the trajectories in each episode are produced by $\pi^k$. However, the data are collected along the learned policy $\pi^k$, while the value gap in **(I)** corresponds to $\pi^{\star}$. This motivates us to introduce an auxiliary, *counterfactual trajectories* $\tilde{\boldsymbol{\tau}}^k:=(\tilde{s}^k_{h},\tilde{a}^k_{h})_h$,  that evolves under
> - the optimal policy $\pi^{\star}$ (rather than the excuted one $\pi^k$)
> - observed exogeneous process $\boldsymbol{\xi}^k=(\xi^k_1,\cdots,\xi^k_h)$
>
> This is one conceptual novelty, in contrast to the realized/observed one  $\boldsymbol{\tau}^k=(s^k_{h},a^k_{h})\_h$  generated by the excuted policy $\pi^k$ and observed $\boldsymbol{\xi}^k$. Based on the counterfactual trajectories, we can bound **(I)** using simulation lemma over $\tilde{\boldsymbol{\tau}}^k$, w.r.t. the *exogenous filtration* generated by the exogenous trajectory $\boldsymbol{\xi}^k$. However, this introduces an additional challenge of **policy misalignment** that prevents the usual telescoping of Bellman errors along the optimal trajectory. Appendix F.3 formalizes this failure and shows that the model error for $\tilde{\boldsymbol{\tau}}^k$:
>
> $$ \epsilon^k_h(\tilde{s}^k\_h,\tilde{a}^k\_h):= \|| P_h(s_{h+1}\in\cdot|s_h,a_h)-\widehat{P}^k_h(s_{h+1}\in\cdot|s_h,a_h) \||_1 $$
>
> cannot be controlled  using on-policy counts $C^k_h(s^k_h,a^k_h)$, which corresponds to the realized trajectory $\boldsymbol{\xi}^k$. We resolve this by rewriting value gap in terms of exogenous-kernel estimation errors, thus replacing the state–action counts for counterfactual process with exogenous counts $C^k_h(\xi_h)$, which is independent of policy and endogeneous processes. This leads to a sublinear regret bound independent of endogeneus state space and action space.
>
> This departs from the classical optimism-based analysis and is the first place where our *counterfactual analysis* is essential. Because exogenous dynamics are action/policy-independent, each realized/observed exogenous trajectory $\boldsymbol{\xi}^k$ can be used to simulate trajectories under any policy $\pi$ rather than $\pi^k$ for evaluating value $V^{k,\pi}$.

---

> ### Author Response · Authors · 2025-11-22
> **Response 3/3**
>
> This is the **classical analysis fails ⇒ new regret decomposition ⇒ counterfactual analysis** message that we  make more explicit in the revised version, by adding a paragraph at the end of Section 4.2 summarizing the above.
>
> In summary, we identify the failure of applying optimism-based analysis for PEL and proposed a new regret decomposition and counterfactual analysis. Furthermore, our analysis yields regret bound independent of endogeneus state space and action space.
>
>
> **B.2) Where is the counterfactual analysis?**
> Formally, we define the counterfactual trajectory $\tilde{\tau}$ corresponding to policy $\pi\neq \pi^k$,  exogeneous process $\boldsymbol{\xi}$, and initial state $x_1$ as $\tilde{\tau}=T(\pi,\boldsymbol{\xi},x_1)$ s.t.
>
> $$ \tilde{s}\_{h} = (\tilde{x}\_{h},\xi_{h}),\tilde{a}_{h} = \pi_h(\tilde{s}_h),\tilde{x}_h=f(\tilde{x}_h,\tilde{a}_h,\xi_h). $$
>
> **i) It first appears at the analysis for tabular Exo-MDP.**
>
> For each episode $k$, we introduce an auxiliary, counterfactual process $(\tilde{s}^k_{h},\tilde{a}^k_{h})$ that evolves under the optimal policy $\pi^{\star}$ (rather than the excuted one $\pi^k$) and observed exogeneous process $\boldsymbol{\xi}^k$.
>
> Intuitively, these counterfactual trajectories answer the question: “What trajectories would have been produced if the policy was optimal instead of the excuted one $\pi^k$?” We work with the exogenous filtration generated by the exogenous trajectory (rather than the full history), and use the simulation lemma to control the value gap **(I)** in Equation [] by the model error $\epsilon^k_h(\tilde{s}^k_{h},\tilde{a}^k_{h})$ over the counterfactual trajectory. Using data processing inequality, we then replace state–action counts $C^k_h(\tilde{s}^k_{h},\tilde{a}^k_{h})$ with exogenous counts $C^k_h(\xi^k_h)$, so that model-estimation error can be controlled by concentration of $C^k_h(\xi^k_h)$. This leads to sublinear regret bound independent of endogeneus state space and action space.
> The chain is
> $$\text{Regret} ⇒ \text{Double Value Gap} ⇒ \epsilon^k_h(\tilde{s}^k_{h},\tilde{a}^k_{h}) ⇒ \epsilon^k_h(\xi^k_h) ⇒ C^k_h(\xi^k_h) ⇒ \mathcal{O}(H^2|\Xi|\sqrt{K}).$$
>
> **ii) It is further used in the analysis for Exo-MDP with LFA.**
>
> The fundamental intuition is similar to the tabular setting. The optimism-based analysis fails in the regret decomposition, and we introduce a new regret decomposition, which requires to bound two value gaps as **(I)** and **(II)** in Equation []. We bound the value gap **(I)** for the optimal policy by applying simulation lemma to counterfactual process generated by $\pi^*$ rather than $\pi^k$. The proof then:
> - decomposes the gap along this counterfactual trajectory into (i) Bellman regression error at the anchors and (ii) model error terms
> - couples all these errors to the realized exogenous trajectory .
>
> This is different from standard linear-MDP optimism analysis, which rely on *confidence sets* and *self-normalized concentration* in the parameter space. Here the key objects are these counterfactual trajectories and the exogenous martingales.
>
> We have clarified this in the revised version by illustrating the counterfactual trajectories after Theorem 1, summarizing the above discussions, and adding a forward reference to Appendix F and Appendix G.
>
> ## Q1
> *How much of this relies on the assumption of known deterministic endogenous dynamics?*
>
> Please refer to **GR3** in Global Response.
>
>
>
> ***
> References
>
> [1] Yang, Lin, and Mengdi Wang. "Sample-optimal parametric q-learning using linearly additive features." ICML 2019.
>
> [2] Tor Lattimore and Csaba Szepesvári. Bandit Algorithms. Cambridge University Press, 2020
>
> [3] Eaton, Eric, et al. "Replicable Reinforcement Learning with Linear Function Approximation." arXiv preprint arXiv:2509.08660 (2025).
>
> [4] Clarkson, Kenneth L. "Coresets, sparse greedy approximation, and the Frank-Wolfe algorithm." ACM Transactions on Algorithms 2010.
>
> [5] Jaggi, Martin. "Revisiting Frank-Wolfe: Projection-free sparse convex optimization." ICML, 2013.
>
> [6] Bachem, Olivier, Mario Lucic, and Andreas Krause. "Practical coreset constructions for machine learning." arXiv preprint arXiv:1703.06476 (2017).
>
> [7] Hamsa Bastani, Mohsen Bayati, and Khashayar Khosravi. Mostly exploration-free algorithms for contextual bandits. Manag. Sci., 2021
>
> [8] Sampath Kannan, Jamie Morgenstern, Aaron Roth, Bo Waggoner, and Zhiwei Steven Wu. A smoothed analysis of the greedy algorithm for the linear contextual bandit problem. In NeurIPS, 2018.
>
> [9] Civitavecchia, Luca, and Matteo Papini. Exploration-Free Reinforcement Learning with Linear Function Approximation. RLJ 2025.

---

### Official Review · Reviewer_odvK · 2025-11-01

**Soundness:** 3
**Presentation:** 2
**Contribution:** 2
**Rating:** 6
**Confidence:** 3

**Summary:**

The authors propose a pure exploration learning framework in Exo-MDPs, where the state decomposes into exogenous and endogenous components. Starting from a simple Exo-Bandit warm-up, they show that in tabular Exo-MDPs, one can estimate the exogenous transition kernel and computed the policy via dynamic programming, deriving a theoretical regret bound. For linear function approximation, they introduce an algorithm called LSVI-PE, which combines post-decision state and counterfactual trajectory analysis, and provide a corresponding theoretical analysis.

**Strengths:**

They present the first near-optimal regret bound for Exo-MDPs under linear function approximation and theoretically establish that it is independent of the endogenous state and action cardinalities. Furthermore, they rigorously prove that the exogenous process in EXO-MDPs evolves independently of the policy, thereby removing the need for explicit exploration. This is supported by $\tilde{\mathcal{O}} (\sqrt{K})$ regret guarantees in both tabular and linear function approximation settings.

**Weaknesses:**

1. The modelling assumptions are somewhat restrictive, as the theoretical results rely on the exogenous state space being discrete and on the endogenous transition and reward functions being known.
2. The regret bound scales linearly with $|\Xi|$ in both the tabular and linear function approximation settings, which may limit scalability. In particular, LSVI-PE can exhibit degraded performance when the anchor placement is suboptimal and $\lambda_0$ becomes small.

**Questions:**

Given the relatively small experimental scale—in terms of state/action set sizes, horizon length, and the number of episodes—and that the baseline is limited to an optimism-augmented variant, would it be feasible to scale up the experiments and include additional baselines, particularly other pure-exploitation methods?

---

> ### Author Response · Authors · 2025-11-22
> **Response 1/2**
>
> Thanks for your feedback! We address your comments below and  have posted a revised version.. During the response we refer to the global response for more details.
>
> ## W1 restrictive modelling assumptions, discrete exogenous state space and known endogenous transition and reward functions
>
> Thank you for prompting us to clarify the scope and motivation behind the Exo-MDP assumptions. Please refer to **GR3** in Global Response.
>
> ## W2
> > The regret bound scales linearly with $|\Xi|$ in both the tabular and linear function approximation settings, which may limit scalability. In particular, LSVI-PE can exhibit degraded performance when the anchor placement is suboptimal and becomes small.
>
> We highlight several clarifications as follows
>
> **Regret scales linearly with $|\Xi|$.**
> The dependence on $|\Xi|$ arises solely from estimating the exogenous Markov kernel, which is the only unknown component in Exo-MDPs. The minimax rate for estimating a $|\Xi|$-state Markov chain is $\Theta(\sqrt{\frac{|\Xi|}{K}})$, so the statistically optimal regret dependence is $O(\sqrt{|\Xi|})$. Our bounds scale as $O(|\Xi|)$, which is therefore off by a factor of $\sqrt{|\Xi|}$. We believe this gap may be improvable with sharper mixing-time–aware or variance-reduction analyses, and we view closing this factor as an interesting direction for future work.
>
> We note that Wan et al. (2024) obtain regret bounds of order $d^2$ for linear-mixture MDPs; in the special case where the exogenous dimension is $d = |\Xi|$, their dependence is comparable to ours. Our results further remove dependence on the (potentially huge or continuous) endogenous state space, which is the primary motivation for working in the Exo-MDP framework.
>
> **Suboptimal anchor placement and small $\lambda_0$**
> We would like to emphasize that the invertibility and well conditionedness $\Sigma$ can be easily guaranteed by *design* of anchors. As is clarified in **GR A)**, anchors are user-selected, and we can exploit domain knowledge and problem structure to construct well-spread set of anchor feature vectors. In principle, we can select the anchors $(x^a_n)_n$ by precomputing their feature vectors $\phi(n)$ offline,  prior to the learning phase. For example, the storage control example in Section 6.2 is a well-conditioned examples with $\Sigma=I_N$ and well-behaved $\lambda_0=1$. We have added this dicussion on invertibility/well-conditionedness of $\Sigma$ in Section 5.
>
> In addition, we provide a discussion in **GR4 C)** on how to improve the well-conditionedness of $\Sigma$ by ridge regression, which we believe is an promising method to improve the numerical stability while tradeoffing bias.
>
> ## Q1 scale up the experiments and include additional baselines
> Thank you for this excellent suggestion. We fully agree that evaluating pure-exploitation methods at larger scales is important. We  expanded the experimental section by scaling up the problem size and adding new pure-exploitation baselines. All updates are reflected in Section 6 and Appendix H.
>
> **Scaled-up experimental setup**
> To better assess scalability, we increased the problem dimensions in both the synthetic Exo-MDP and the storage control setting:
> - Larger state spaces. We concurrently expanded both the endogenous and exogenous state spaces, and evaluated the robustness and effectiveness of pure-exloitation methods on problems with state spaces scaled to 2x to 4x the original size.
> - Longer horizons. We incrementally scaled up the time steps of a full horizon from 20 to 40, stressing the multi-step compounding of approximation errors.
> - More episodes. For all the expanded large-scale experimental settings mentioned above, we configured 200-250 episodes to observe the long-term performance and convergence characteristics of each algorithm.
>
> **Added additional baselines**
> We implemented two new baselines PTO-Lite and LSVI-PE-Lite. Instead of constructing the full empirical exogenous model from all episodes, Lite subsamples the historical exogenous transitions at each stage. The resulting subsampled dataset is used to compute a lightweight estimate, reducing computation while keeping the model statistically representative. To enhance the analysis of Lite’s performance across different setup scales, we designed three Lite variants with distinct subsample ratios for the two new baselines, specifically with subsampling ratios of 0.2, 0.5, and 0.8. These baselines explicitly trade off statistical efficiency vs. computational efficiency and serve as the natural intermediate points between Pure Exploitation Learning (PEL) algorithms and exploration-heavy methods.

---

> ### Author Response · Authors · 2025-11-22
> **Response 2/2**
>
> **Empirical findings**
>
> Across all enlarged benchmarks, the following consistent trends emerge:
> - PEL retains the best overall performance, even as state/action dimensions and horizons increase.
> - Lite achieves performance close to PEL, while reducing computational cost by 20%-80%, validating that PEL remains effective even under aggressive subsampling of exogenous traces.
> - Lite variants with smaller subsample factors yield more significant reductions in computational overhead, but at the cost of moderately degraded experimental performance. Nevertheless, the two new pure-exploitation baselines still outperform the exploration-driven method.
> - Exploration-driven baselines PTO-Opt, LSVI-UCB underperform because exploration does not help in Exo-MDPs as our theory indicates, and their computational cost grows poorly with dimension.
>
> These results, shown in updated Figure 1 and Figure 3 and the extended comparisons in Figure 5, support the scalability of our approach and its robustness and effectiveness across different scenarios.

---

### Author Response · Authors · 2025-11-22
**Global Response 1/5**

We thank all reviewers for your thoughtful and constructive feedback. We carefully reviewed all comments and have posted a revised version, along with detailed point-by-point responses for each reviewer.  Below we summarize the main updates and clarifications.


## GR1. Framing and presentation
We carefully revised the introduction, contributions, and discussion to better position the contributions of our work and clearly articulate why pure exploitation is viable and efficient in Exo-MDPs. We also streamlined notation and ensured consistency across the main text and appendix. We hope these edits help improve the main story and readability of the paper.

## GR2. Highlight intuition why pure exploitation is sufficient
> Reviewer 4nVr: "intuition why such a positive result is possible"
> Reviewer 4oif: "intuition why pure exploration is sufficient to achieve rate-optimal regret in this setting"

Thank you for pointing this out. In the revision, we substantially expanded Sections 3–5 to make this decoupling explicit and to articulate why it eliminates the need for exploration.

We adjusted the manuscript to explain the core structural reason that exploration is unnecessary in Exo-MDPs. Because the exogenous process evolves independently of the agent's actions, each episode reveals an exogenous trace $\xi_{1:H} = (\xi_1, \ldots, \xi_H)$ that can be reused to evaluate the performance of *any* policy.

Indeed, given a realized exogenous sequence $\xi_{>h}$, the exogenous transition map $f$ and reward function $r$ are known, so the *hindsight* or ("exogenous") value function $V_h^{\pi}(s, \xi_{> h})$ is fully determined, known, and evaluatable for *any* policy (see Section 3). The true value function $V_h^{\pi}(s)$ is simply the expectation of this quantity over the exogenous distribution. Thus, a single observed trace provides an *unbiased estimate of the value of every policy*, not only on the policy that generated the trajectory.

This data-reuse property distinguishes Exo-MDPs from general MDPs and removes the classical exploration-exploitation tradeoff, since any greedy trajectory provides the same statistical information about the exogenous distribution as any exploration one.

As a result, pure exploitation can continually refine value estimates for all policies without requiring optimism or forced exploration. We added this inituition/explanation in the revised Introduction and Section 3.

## GR3. Clarification of (Exo-MDP) modeling assumptions
>Reviewer odvK: "The modelling assumptions are somewhat restrictive, as the theoretical results rely on the exogenous state space being discrete and on the endogenous transition and reward functions being known."
Reviewer 4nVr: "How much of this relies on the assumption of known deterministic endogenous dynamics?"
Reviewer 8gd3: "If the transition dynamics (the $f$) in tabular MDPs were not known, would pure exploitation still suffice?"
Reviewer 4oif: "the setting is quite specific" "the assumption that the only unknown component is the exogenous process seems quite strong"


We thank the reviewers for prompting us to clarify the scope and motivation behind the Exo-MDP assumptions. In the revision we expanded Section 3 with an example from inventory control which naturally satisfies known deterministic endogenous dynamics and reward functions. Detailed discussion and additional examples are now provided in Appendix C.
### A) Known transition function and reward function
First we want to clarify that our model assumes that the endogenous dynamics are **stochastic**, only deterministic as a function of the exogenous state distribution through $f$. The full transition kernel from state $s = (x, \xi)$ to $s' = (x', \xi')$ can be written as:
$$P(s' | s,a) = 1[f(x,a,\xi') = x'] \Pr(\xi' \mid \xi)$$

We want to highlight several aspects of the assumption that the transition and reward function $f$ and $r$ are known.

**i) Sufficiency.** These assumptions are precisely what make pure exploitation viable. As described above under point **GR2**, once $f$ and $r$ are known the only source of uncertainty is the exogenous distribution $\mathbb{P}_h(\cdot \mid \xi)$ which is independent of the learner's actions. This structure enables data reuse and counterfactual value estimation and is the basis for our regret guarantees. No analysis of PE in even fully tabular Exo-MDP (with Markovian exogeneous proceses). We establish the regret bound of PE in tabular Exo-MDP, and more importantly, we are the first to show the effectiveness of PE in the case of continuous endogenous state space and continuous action space.

---

> ### Author Response · Authors · 2025-11-22
> **Global Response 2/5**
>
> **ii) Necessity.** We also added a new discussion and formal result showing that these assumptions are not only realistic, but also necessary for pure exploitation to succeed. If either the endogenous transition function $f$ or the reward function $r$ is unknown, the problem class already contains hard instances equivalent to standard bandits and general finite MDPs. In these settings, purely greedy algorithms are known to incur $\Omega(K)$ regret.
>
> Indeed, in our revision we added a formal statement in Section 4.3 (pure exploitation can fail in general MDPs):
>
> > For any pure-exploitation algorithm, there exists a tabular MDP with unknown $f$ or unknown $r$ on which the algorithm suffers linear regret.
>
> In other words, **without known $f,r$, pure exploitation is not (minimax) sufficient. There exist hard instances with linear regret.** We includes simple 2-armed bandits (a 1-step Exo-MDP) where FTL incurs $\Omega(K)$ regret, and 2-step MDPs where unknown transitions reduce to unknown rewards at the first stage. Thus, without known $f,r$, pure exploitation is not minimax-sufficient, in sharp contrast to the Exo-MDP setting where we show sublinear regret is achievable.
>
> **iii) Reasonableness.** Known deterministic endogenous dynamics and reward maps are standard in operation research/ADP models: e.g., inventory updates, storage evolution, and queueing dynamics are deterministic functions of the previous state, action, and exogenous input (demand, price, arrivals). The uncertainty enters entirely through exogenous drivers, exactly matching the Exo-MDP abstraction. This model has also been studied in other papers, including [1-3].
>
> ### B) On discrete exogeneous state space
> A substantial body of existing work shows that the Exo-MDP formulation is not a restrictive modeling assumption in practice. In many real systems, the endogenous state space can be extremely large or even continuous, while the exogenous driver is low-dimensional. This pattern is well documented in the operations research and inventory-control literature, where the system state (inventory levels, backlog, storage levels, or resource capacities) are continuous, but the exogenous state (demand, prices, arrivals) is captured by a small discrete set of possible inputs. We've included a discussion on this in Section 3, and refer to [4] for an example model of this in inventory control.
>
> Moreover, [5] formally establish an equivalence between Exo-MDPs, tabular MDPs, and discrete linear-mixture MDPs. Their result shows that the defining restriction of Exo-MDPs is not the discreteness of the exogenous space, but rather the assumption that the endogenous transition and reward maps $f$ and $r$ are known.
>
> ***
> Reference
>
> [1] Madeka, Dhruv, et al. "Deep inventory management." arXiv preprint arXiv:2210.03137 (2022).
>
> [2] Che, Ethan, Jing Dong, and Hongseok Namkoong. "Differentiable discrete event simulation for queuing network control." arXiv preprint arXiv:2409.03740 (2024).
>
> [3] Sinclair, Sean R., et al. "Hindsight learning for mdps with exogenous inputs." International Conference on Machine Learning. PMLR, 2023.
>
> [4] Besbes, Omar, and Alp Muharremoglu. "On implications of demand censoring in the newsvendor problem." Management Science 59.6 (2013): 1407-1424.
>
> [5] Wan, Jia, et al. "Exploiting Exogenous Structure for Sample-Efficient Reinforcement Learning." arXiv preprint arXiv:2409.14557 (2024).
>
> ## GR4. Clarification of anchor states
> >Reviewer odvK: "degraded performance when the anchor placement is suboptimal and $\lambda_0$ becomes small."
> Reviewer 4nVr: "Needing the existence of an anchor set that is known to the learner, plus additional assumptions on said set, is quite strong in the RL literature" "Can the anchor set assumption be weakened? What could replace it?"
> Reviewer 8gd3: "the existence of the anchor set is not strong" "it is a strong assumption that the learning algorithm knows this set" "Why use the anchor points to define the matrix instead of using the data collected during the policy rollout phase?" "Why is it not possible to guarantee the invertibility of $\Sigma$ by defining $\Sigma=\sum_{n=1}^N \phi_n\phi_n^\top+\beta I$ where $\beta$ is  a small scalar and $I$ is the identity matrix?"
> Reviewer 4oif: "The assumption that anchor points are known seems strong. Could you comment on this assumption? Is it possible to relax it?"
>
> ### A) Knowledge of anchor set (Reviewer 4nVr/8gd3/4oif)
> Thank you for the valuable comments. We agree this deserves clearer explanation and have revised Section 5 and Appendix C to make the following points explicit. While knowing the anchor set a priori appears strong from a *general RL* perspective, this assumption is well-motivated in our Exo-MDP setting:

---

> ### Author Response · Authors · 2025-11-22
> **Global Response 3/5**
>
> - Anchor states are designed by the learner. Assumption 1 requires that we fix a finite collection of post-decision states $x^a_h(n)$ such that the feature matrix $\Phi_h := [\phi(x_h^{a}(1)),\,\ldots,\,\phi(x_h^{a}(N))]$ has full row rank. These states are the representative grid that the practitioner chooses when constructing the feature map $\phi$, including hat basis, spline knots, tile centers [1]. This mirrors standard RL with LFA, where the learner chooses $\phi$ and implicitly chooses the *basis points* on which $\phi$ is built.
> - This is standard in ADP and matches how Exo-MDPs are implemented in practice. Anchor states are standard in ADP for control and operation research applications [2-4]. In applications like inventory control and storage systems [2-3], practitioners often can exploit *domain knowledge* and *problem structure*, e.g.  piecewise-linearity or convexity of the value functions [4]. For instance, with piecewise-linear hat features, anchors are just the breakpoints of the basis functions [3-4].  This is also shown in our storage control example in Section 6.2.
>
> We emphasize these points in Section 5 (Role of anchor states) and provide a recipe for constructing anchors in common models, including tabular, and hat/spline in Appendix C.2.
> ***
> References
>
> [1] RS Sutton, AG Barto. Reinforcement learning: An introduction.
>
> [2] J. Nascimento and W. Powell. An optimal approximate dynamic programming
>
> algorithm for the lagged asset acquisition problem. Mathematics of Operations Research, 2009.
>
> [3] J. Nascimento and W. Powell. An optimal approximate dynamic programming algorithm for concave, scalar storage problems with vector-valued controls. IEEE Transactions on Automatic Control, 2013.
>
> [4] W. Powell. Reinforcement Learning and Stochastic Optimization: A Unified Framework for Sequential Decisions, 2022.
>
>
> ### B) Why not use rollout states to build $\Sigma$ ⇒ use of anchor states (Reviewer 8gd3)
>
> Our goal is to study Pure Expoitation Learning (PEL) in Exo-MDPs under LFA, a setting with fundamentally different statistical behavior than optimism-based RL with LFA. Recall that under post-decision LFA we have:
> $$ V_h^{\pi,a}(x^{a},\xi) = \phi(x^{a})^{\top} w_h^{\pi}(\xi). $$
>
> Constructing $\Sigma$ from rollout (on-policy) states (which is standard in optimistic RL) breaks down for PEL, for two reasons.
>
> **Failure 1: Rollout states in Pure exploitation fail in coverage.**
>
> In pure exploitation, all trajectories come from the greedy policies $\pi^k$. Consequently, the visited post-decision states **concentrate on a small region of the feature space**.  Without any optimism, these trajectories fail to cover directions of $\phi(x^a)$ needed to accurately approximate $V^\star(x^a,\xi)$. Optimistic algorithms such as LSVI-UCB [1] or UCRL-VTR [2] *manufacture* this converage by forcing exploration.  But PEL deliberately removes these exploration mechanisms, so roll-out based $\Sigma$ becomes too ill-conditioned to support value regression.
>
> To avoid this degeneracy, we construct $\Sigma$ from a fixed, user-chosen anchor set, ensuring that the anchor features span all necessary directions. In Section 6.2 these anchors correspond to a simple uniform grid, yielding $\Sigma = I_N$, and excellent conditioning.
>
> **Failure 2:  optimism-based analysis fails.**
>
> A second, more sublte but crucial issue arises in our analysis. We need to bound two value gaps, one of which corresponds to the policy $\pi^*$ rather than $\pi^k$. If instead we constructed $\Sigma$ from the rollout states, then $\Sigma$, the targets, and the exogenous model would all be jointly *data-dependent*, and in particular adapted to the same exogenous filtration driving the martingale differences. This breaks the clean separation we exploit in Appendix G: the self-normalized martingale tools used in optimism-based linear MDP analyses would be needed, but those rely on explicit exploration to guarantee sufficient coverage of the feature space, which we deliberately avoid in pure exploitation. In contrast, the counterfactual analysis in Appendix G uses the fact that $\Sigma_h$ is fixed and data-independent, so that we can (i) first bound the anchor-level Bellman regression errors using standard concentration for the exogenous model; (ii) then propagate these errors to arbitrary states with constants depending only on $\lambda_0$.
>
> **Why it is harmless to *discard* on-policy endogenous states.**
> - Actions do not affect the exogenous process.  Therefore, ignoring endogenous states for estimating $\Sigma$ does not bias learning.
> - We still use all observed exogenous transitions to estimate $\hat{P}$, which is the only unknown component driving Bellman targets.
> - Anchors provide full feature-space coverage, ensuring that value approximation remains accurate even if the greedy policy visits only a small subset of the state space.

---

> > ### Author Response · Authors · 2025-11-22
> > **Global Response 4/5**
> >
> > ### C) Why not guarantee the invertibility of $\Sigma$ by using ridge? (Reviewer 8gd3) Suboptimal anchor placement and small $\lambda_0$ (Reviewer odvK)
> > We would like to emphasize that the invertibility and well-conditionedness of $\Sigma$ can be guaranteed directly by design of the anchor set. As clarified in **GR A)**, anchors are fully user-selected, and practitioners can exploit domain knowledge or problem structure to construct a well-spread collection of feature vectors. In principle, one can precompute the anchor feature vectors $\phi(x^a_n)$ offline, prior to learning, ensuring that $\Sigma$ is full-rank and well-conditioned. For example, in the storage control experiment (Section 6.2), a simple uniform grid yields $\Sigma = I_N$ and $\lambda_0 = 1$. We have added text in Section 5 emphasizing how anchor selection naturally controls $\lambda_0$.
> >
> >
> > **Thank you for the suggestions on ridge regression/regularization.** Ridge regression is a promising method to improve the invertibility/numerical stability while tradeoffing bias. While we can improve the invertibility of $\Sigma$ by carefully designing the anchors and features, such process can be computationally heavy sometimes. Regularization can replace strict invertibility with a controlled bias term. It trades a small bias controlled by $\beta$ for numerical stability. We now provide a short theoretical sketch showing how a $\beta$-regularizer impacts the regret bound. Recall that the original regret bound in Theorem 2 states
> > $$ (\sqrt{N/\lambda_0}+\sqrt{d}\big)|\Xi|H\sqrt{K}. $$
> > Letting $\lambda_{\beta}:=\lambda_0+\beta$, then reg. reduces the conditioning part of the regret bound to
> > $(\sqrt{N / \lambda_{\beta}}+\sqrt{d})|\Xi|H\sqrt{K}$. However, it introduces the additional bias as it solves a reguralized problem
> >
> > $$ \min_w \|\Sigma_h w - y\|_2^2 + \beta\|w\|_2^2. $$
> >
> > So even if the model is perfectly realizable (no approx. error), we can show that his induces an extra Bellman error term of order $\frac{\beta}{\lambda_{\beta}} \sqrt{d}$, which then propagates through the horizon and across $K$ episodes. Therefore, the total regret bound is worsened as
> > $$ (\sqrt{N/\lambda_{\beta}}+\sqrt{d}\big)|\Xi|H\sqrt{K} + \frac{H}{\sqrt{\lambda_{\beta}}} K \frac{\beta}{\lambda_{\beta}} \sqrt{d}.$$
> >
> > In the realizable setting our main theory takes $\beta=0$,  so no extra bias term appears. If one adds a small ridge term $\beta I$ numerical stability, the analysis can be interpreted as introducing an effective inherent Bellman error of size $\epsilon_{\text{ridge}}=\mathcal{O}(\frac{\beta}{\lambda_{\beta}} \sqrt{d})$, which adds an $\frac{\beta H \sqrt{d}K}{\lambda_{\beta}^{\frac{3}{2}}}$. Thus ridge reduces only constants in the $\mathcal{O}(\sqrt{K})$ term, but introduces an additional linear-in-$K$ contribution, which vanishes as $\beta\rightarrow0$.
> >
> > ### D) Weakening/possible relaxations (of known anchor set) (Reviewer 4nVr/4oif)
> > We suggest several natural directions for further relaxation:
> > - Adaptive anchors selection via coreset/Frank-Wolfe style procedures.
> > The anchors can be viewed as a small, hand-picked coreset for the endogenous space. In principle, we can learn this coreset from data via coreset/Frank-Wolfe style procedures and then running LSVI-PE on the resulting anchors. Analyzing such a scheme requires a second layer of error control between the data-driven anchors and the optimal anchor set and is beyond the scope of the present work, but we now mention this as a promising direction in the Conclusion.
> >
> > - Approximate closure assumptions via inherent Bellman error. The realizable analysis (Theorems 2–3) uses Assumptions 2–3 (or 4) to guarantee exact closure of Bellman updates in the anchor span. However, our agnostic analysis (Theorems 4–5) only requires the basic Assumption 1 plus a finite inherent Bellman error. In other words, even if the post-decision values are only approximately representable by the anchor features and the cone/closure conditions only hold approximately, the regret bound still holds with an additional $O(K \varepsilon_{\mathrm{BE}})$ bias term. This already provides a quantitative weakening: violations of Assumptions 2–3 are absorbed into $\varepsilon_{\mathrm{BE}}$, rather than being ruled out outright. We will make this connection explicit in the text by pointing out that Assumptions 2–3 are mainly used to ensure $\varepsilon_{\mathrm{BE}} = 0$. When they fail, the agnostic bounds describe the resulting bias.

---

> > > ### Author Response · Authors · 2025-11-22
> > > **Global Response 5/5**
> > >
> > > ## GR5. Proofs and technical presentation
> > > > Reviewer 4nVr: "The proofs in the appendix, especially for the linear MDP section, are quite poorly written"
> > >
> > > All proofs in the appendix have been substantially rewritten for consistency, clarity, and completeness, ensuring that the logical flow and notation align with the main text.
> > >
> > > ## GR6. Expanded numerical evaluation
> > > > Reviewer odvK: "scale up the experiments and include additional baselines, particularly other pure-exploitation methods"
> > >
> > > We augmented the simulation section with additional experiments，including new baselines and ablation studies. This illustrates the scalability and robustness of pure exploitation methods under varying conditions, as well as the advantage over other pure-exploitation methods.
> > >
> > > In summary, we are grateful for the reviewers careful reading and believe the revisions substantially improve clarity and scope. We hope these changes address all concerns and help convey the contribution of establishing the first regret guarantees for pure exploitation in Exo-MDPs, and welcome any additional comments or suggestions.

---

### Meta-Review · Area_Chair_gMMe · 2026-01-12

**Summary:**

The paper shows that pure exploration is sufficient for rate-optimal regret in exogenous MDPs with linear function approximation.

The reviewers have concerns regarding (i) the algorithm requires knowing the endogenous map and the anchor set, (ii) restrictive modeling assumptions and (iii) poorly written proofs with little intuition provided.

**Reviewer Concerns:**

The authors did a job in terms of addressing concerns regarding the modeling assumption and the writing quality. On the other hand, requiring knowledge about the endogenous map and the anchor set does seem to be a weakness of their algorithm. However, I believe the techniques and results are interesting enough and deserve to be published at ICLR.

**Reviewer Scores:**

I believe all reviewers will keep their scores.

---

### Decision · Program_Chairs · 2026-01-26

Accept (Poster)